# PLSCR1 is a cell-autonomous defence factor against SARS-CoV-2 infection

Dijin Xu[1,2,3,4,13], Weiqian Jiang[1,2,3,13], Lizhen Wu[3], Ryan G. Gaudet[1,2,3,4], Eui-Soon Park[1,2,3,4], Maohan Su[5], Sudheer Kumar Cheppali[6,7], Nagarjuna R. Cheemarla[3,8], Pradeep Kumar[1,2,3,4], Pradeep D. Uchil[4], Jonathan R. Grover[4], Ellen F. Foxman[3,8], Chelsea M. Brown[9], Phillip J. Stansfeld[9], Joerg Bewersdorf[5], Walther Mothes[4], Erdem Karatekin[6,7,10,11,12], Craig B. Wilen[3,8] & John D. MacMicking[1,2,3,4✉]

Understanding protective immunity to COVID-19 facilitates preparedness for future pandemics and combats new SARS-CoV-2 variants emerging in the human population. Neutralizing antibodies have been widely studied; however, on the basis of large-scale exome sequencing of protected versus severely ill patients with COVID-19, local cell-autonomous defence is also crucial[1–4]. Here we identify phospholipid scramblase 1 (PLSCR1) as a potent cell-autonomous restriction factor against live SARS-CoV-2 infection in parallel genome-wide CRISPR–Cas9 screens of human lung epithelia and hepatocytes before and after stimulation with interferon-γ (IFNγ). IFNγ-induced PLSCR1 not only restricted SARS-CoV-2 USA-WA1/2020, but was also effective against the Delta B.1.617.2 and Omicron BA.1 lineages. Its robust activity extended to other highly pathogenic coronaviruses, was functionally conserved in bats and mice, and interfered with the uptake of SARS-CoV-2 in both the endocytic and the TMPRSS2-dependent fusion routes. Whole-cell 4Pi single-molecule switching nanoscopy together with bipartite nano-reporter assays found that PLSCR1 directly targeted SARS-CoV-2-containing vesicles to prevent spike-mediated fusion and viral escape. A PLSCR1 C-terminal β-barrel domain—but not lipid scramblase activity—was essential for this fusogenic blockade. Our mechanistic studies, together with reports that COVID-associated *PLSCR1* mutations are found in some susceptible people[3,4], identify an anti-coronavirus protein that interferes at a late entry step before viral RNA is released into the host-cell cytosol.

Cell-autonomous immunity is an essential survival strategy used by bacteria, plants and animals to combat infection[5–7]. In people, it safeguards mucosal barriers and target tissues against major human-tropic pathogens including *Mycobacterium tuberculosis*, *Salmonella enterica* serovar Typhi, *Shigella flexneri* and HIV-1[8–11]. Whether cell-autonomous immunity combats SARS-CoV-2 has not been fully investigated, however, because most attention has focused on the role of neutralizing antibodies. This question takes on greater urgency given evidence showing that T cells recognize SARS-CoV-2 vaccines and new viral variants of concern (VOCs) by secreting IFNγ[12,13], a type II cytokine that is known to mobilize human cell-autonomous immunity in most nucleated cells[5]. Indeed, increased production of IFNγ coincides with protection against COVID-19 in young adults and children, along with increased expression of type I (IFNα and IFNβ) and III (IFNλ) interferons (IFNs)[14,15]. Accordingly, genetic lesions in IFN signalling are often associated with severe disease[1–4,16] that, together with type I and II IFN autoantibodies[17–19], could account for up to 20% of critical COVID-19

cases[20]. In addition, IFNγ therapy promoted SARS-CoV-2 clearance and rescued immunodeficient patients with COVID-19 who had not recovered after treatment with convalescent plasma or remdesivir[21]. Collectively, these discoveries suggest that IFNγ could act as a central orchestrator of anti-SARS-CoV-2 defence. Characterizing its activity will provide insights into how cell-autonomous immunity confers frontline resistance during COVID-19 and aid our understanding of both natural and vaccine-induced protection.

## Human PLSCR1 inhibits SARS-CoV-2

We first tested the potency of IFNγ at restricting infection with live SARS-CoV-2 using human Huh7.5 hepatoma cells that naturally express the ACE2 receptor[22–24]. SARS-CoV-2 USA-WA1/2020 proved to be highly sensitive to recombinant human IFNγ; a mean half-maximum inhibitory concentration (IC$_{50}$) of 7.14 pM resembled that of recombinant human IFNα2a (IC$_{50}$, 3.25 pM) in dose–response curves (Fig. 1a).

[1]Howard Hughes Medical Institute, New Haven, CT, USA. [2]Yale Systems Biology Institute, West Haven, CT, USA. [3]Department of Immunobiology, Yale University School of Medicine, New Haven, CT, USA. [4]Department of Microbial Pathogenesis, Yale University School of Medicine, New Haven, CT, USA. [5]Department of Cell Biology, Yale University School of Medicine, New Haven, CT, USA. [6]Yale Nanobiology Institute, West Haven, CT, USA. [7]Department of Cellular and Molecular Physiology, Yale University School of Medicine, New Haven, CT, USA. [8]Department of Laboratory Medicine, Yale University School of Medicine, New Haven, CT, USA. [9]School of Life Sciences and Department of Chemistry, University of Warwick, Coventry, UK. [10]Department of Molecular Biophysics and Biochemistry, Yale University, New Haven, CT, USA. [11]Saints-Pères Paris Institute for the Neurosciences, Université de Paris, Centre National de la Recherche Scientifique UMR 8003, Paris, France. [12]Wu Tsai Institute, Yale University, New Haven, CT, USA. [13]These authors contributed equally: Dijin Xu, Weiqian Jiang. ✉e-mail: john.macmicking@yale.edu

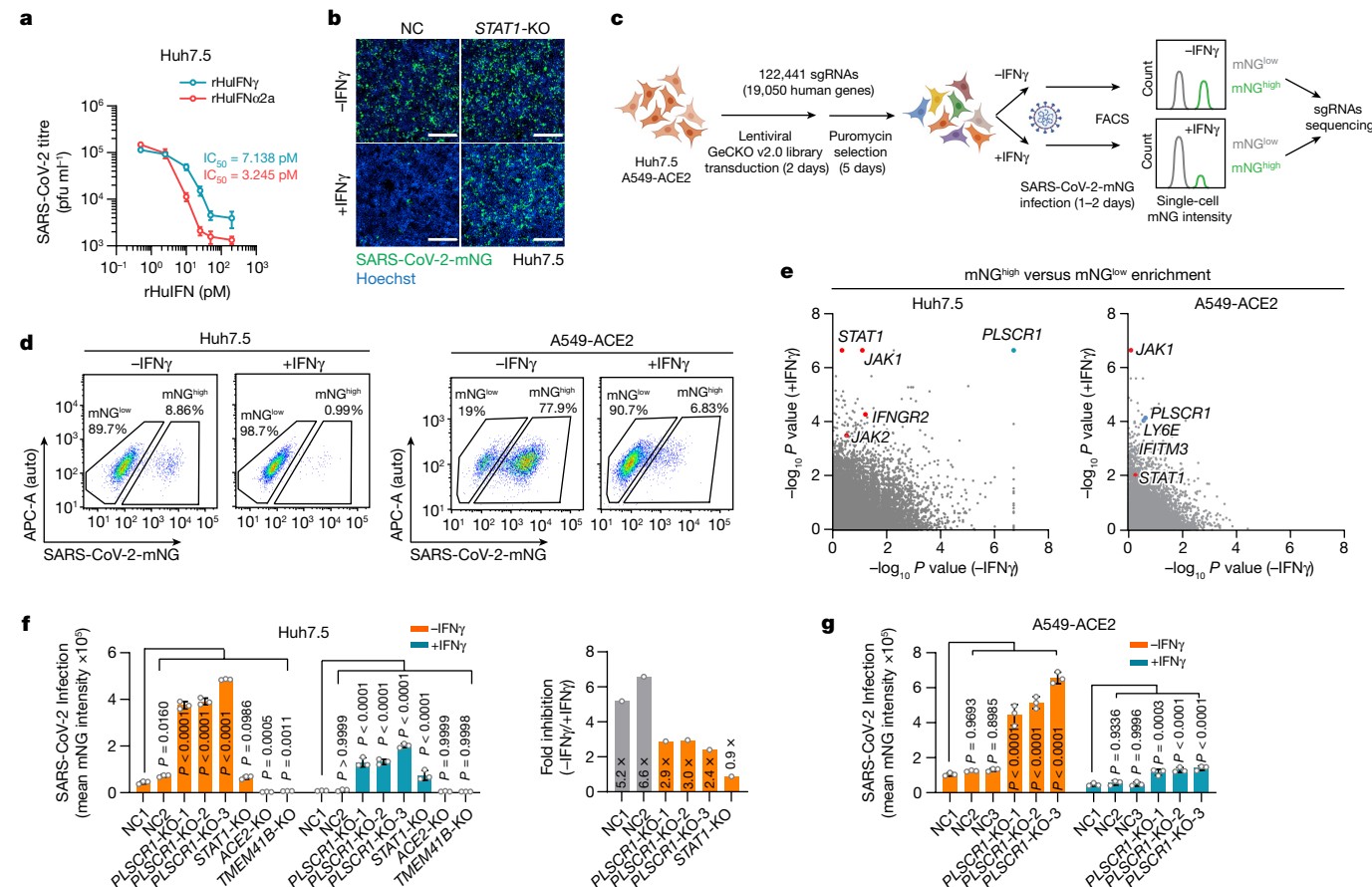

**Fig. 1 | Genome-wide CRISPR–Cas9 screens identify PLSCR1 as a potent anti-SARS-CoV-2 defence factor. a**, Huh7.5 cells were treated with different concentrations of IFNs (recombinant human IFNγ (rHuIFNγ) or recombinant human IFNα2a (rHuIFNα2a)) and then infected with SARS-CoV-2 (isolate USA-WA1/2020) at a multiplicity of infection (MOI) of 1. Virus production (plaque-forming units (PFU) per ml) was quantified by plaque assay at 2 days post-infection (dpi) ($n = 3$). **b**, Representative images showing infection with SARS-CoV-2 expressing mNeonGreen (SARS-CoV-2-mNG, isolate USA-WA1/2020) in negative control (NC) or *STAT1*-KO Huh7.5 cells in resting or IFNγ (8 U ml$^{-1}$)-activated conditions. Green, SARS-CoV-2-mNG; blue, Hoechst. **c**, Schema showing the genome-wide CRISPR screening workflow. **d**, FACS plots of resting or IFNγ-activated Huh7.5 (8 U ml$^{-1}$) (left) and A549-ACE2 (70 U ml$^{-1}$) (right) cells infected with SARS-CoV-2-mNG at an MOI of 1 for 48 h or an MOI of 0.3 for 24 h,

respectively. The percentage of infected cells is shown for populations with a high (mNG$^{high}$) or low (mNG$^{low}$) level of mNG expression. **e**, Comparisons of gene-level enrichment scores (mNG$^{high}$ versus mNG$^{low}$ populations) between untreated and IFNγ-treated conditions in Huh7.5 (left) and A549-ACE2 (right) cells. **f**, SARS-CoV-2-mNG fluorescent intensity (normalized to cell counts) in Huh7.5 (left) and average restriction ratio (−IFNγ/+IFNγ) in Huh7.5 cells of the indicated genotypes (right) ($n = 3$). **g**, SARS-CoV-2-mNG fluorescent intensity in A549-ACE2 cells of the indicated genotypes. The fluorescent intensity of mNeonGreen was quantified at 2 dpi for Huh7.5 cells (**f**) and 1 dpi for A549-ACE2 cells (**g**). Three *PLSCR1*-KO cell lines were generated using different sgRNAs ($n = 3$). Data are mean ± s.d. $P$ values from one-way ANOVA followed by Tukey's multiple comparison test in **f**,**g**. Scale bars (**b**), 500 μm. Experiments performed three times.

The potent anti-SARS-CoV-2 activity of IFNγ was confirmed using CRISPR–Cas9 engineering to delete signal transducer and activator of transcription-1 (STAT1), which is required for IFNγ-induced gene expression. Stable Huh7.5 *STAT1*-knockout (KO) cells failed to control SARS-CoV-2 after exposure to recombinant human IFNγ (Fig. 1b). Thus, human type II IFNγ is a powerful signal reprogramming human cells to restrict SARS-CoV-2 in a STAT1-dependent manner.

Next, we sought to identify which IFNγ-induced proteins conferred this effect. Parallel genome-wide loss-of-function (LoF) screens were used to uncover anti-SARS-CoV-2 restriction factors in Huh7.5 cells or human lung epithelial A549 cells ectopically expressing ACE2, the latter mimicking host-cell targets within the respiratory tract[23] (Fig. 1c). A GeCKO v2.0 single-guide RNA (sgRNA) library of 122,441 sgRNAs across 19,050 genes was transduced into each cell type, and this was followed by puromycin selection for stable integration. sgRNA-integrated cells were then treated with recombinant human IFNγ before being infected with SARS-CoV-2 expressing mNeonGreen (mNG). This strategy allowed us to use fluorescence-activated cell sorting (FACS) to

separate infected cells into permissive mNG$^{high}$ or restrictive mNG$^{low}$ populations, and then perform next-generation sequencing of sgRNA frequencies. sgRNAs that target key host defence factors accumulated in the mNG$^{high}$ group (Fig. 1d).

Human genes enriched in mNG$^{high}$ versus mNG$^{low}$ populations were ranked by the MAGeCK algorithm, with $P$ values in the resting and the IFNγ-treated conditions presented for comparison (Fig. 1e). Key genes in the IFNγ signalling pathway (*IFNGR2*, *STAT1* and *JAK2*) and several IFN-stimulated genes (ISGs) that have been reported[22,25] to restrict SARS-CoV-2 (for example, *LY6E* and *IFITM3*) were identified, verifying the robustness of the LoF screen (Fig. 1e). In addition, a phospholipid scramblase 1 (*PLSCR1*) gene with reported antiviral activities[26–30] but uncharacterized against SARS-CoV-2 was one of the most significant mNG$^{high}$ hits in both Huh7.5 and A549 cells, even under conditions of basal expression (Fig. 1e). These pronounced phenotypes were subsequently validated by three independent *PLSCR1* sgRNAs; here, the percentage of infected Huh7.5 cells increased by up to 5.2-fold and the viral mNG signal by 7.2-fold in cells with disrupted *PLSCR1* alleles

(Fig. 1f and Extended Data Fig. 1a,b). Notably, chromosomal disruption of *ACE2* or *TMEM41B*—which served as positive controls—blocked virus uptake and replication, whereas *STAT1* deficiency rendered Huh7.5 cells more susceptible after treatment with IFNγ, like the loss of *PLSCR1* (Fig. 1f). Disruption of *PLSCR1* also led to a 4.5-fold increase in viral load in A549-ACE2 lung epithelia (Fig. 1g). Hence, both genome-scale and single-clonal LoF analyses show that human PLSCR1 is an important restriction factor against SARS-CoV-2 infection.

*PLSCR1* mRNA is naturally upregulated within the upper respiratory tract of people infected with SARS-CoV-2[31] (Extended Data Fig. 1c), and is strongly induced by IFNγ in human primary tracheal epithelia and multiple cell lines (Extended Data Fig. 1d). Its protein expression was also upregulated in most cell types treated with higher doses of type I IFNα2a, IFNβ1a or type III IFNλ1 (Extended Data Fig. 1d,e), consistent with both interferon-stimulated response element (ISRE) and gamma-interferon activation site (GAS) elements being identified within its promoter (Extended Data Fig. 1f). Notably, the addition of recombinant human IFNα2a, IFNβ1a or IFNλ1 at these doses required PLSCR1 to inhibit coronavirus infection (Extended Data Fig. 1g). Hence, although PLSCR1 responds most robustly to type II IFNγ signalling to protect human cells against SARS-CoV-2, it can respond to type I or III IFNs as well.

## PLSCR1 activity across VOCs and hosts

Antiviral defects in *PLSCR1*-KO cells were rescued by genetic complementation. Stable reintroduction of PLSCR1 into *PLSCR1*-KO Huh7.5 clones completely reversed the LoF phenotype (Fig. 2a and Extended Data Fig. 2a). This was evident not only in the mean viral mNG intensity but also in plaque assays measuring the amount of infectious virus produced (Fig. 2b,c). Furthermore, overexpression of PLSCR1 in the absence of IFNγ priming reduced SARS-CoV-2 infection by as much as 78% (Fig. 2d and Extended Data Fig. 2b). A similar antiviral effect was observed in *STAT1*-KO cells, indicating that its activity was independent of the activation of other ISGs (Fig. 2e and Extended Data Fig. 2c–e). Thus, PLSCR1 itself is sufficient for substantial restriction of SARS-CoV-2, a phenotype observed in previous LoF assays in which basal levels of PLSCR1 were also protective. This restriction extended to SARS-CoV-2 Delta B.1.617.2 and Omicron B.1.1.529, two VOCs with higher transmissibility and immune evasion than the original USA-WA1/2020 strain[32]. Here, measuring the total viral RNA levels using quantitative PCR (qPCR) showed that infection by both VOCs increased significantly by 6.0-fold (Delta) or 7.6-fold (Omicron) in *PLSCR1*-KO cells (Fig. 2f). Complementation with *PLSCR1* cDNA reversed these effects (Fig. 2f). Thus, PLSCR1 exhibits broad antiviral activity against SARS-CoV-2 VOCs.

This broad anti-SARS-CoV-2 profile prompted us to consider whether such activity is evolutionarily conserved. Three epidemic and pandemic coronaviruses (SARS-CoV, MERS-CoV and SARS-CoV-2) possibly emerged through spillover from bats or camels[33], suggesting that PLSCR1 might confer anti-coronavirus activity in these zoonotic hosts and in experimental models such as mice. Notably, cross-species complementation using PLSCR1 orthologues from horseshoe bats (*Rhinolophus sinicus*, a reservoir for SARS-CoV and SARS-CoV-2)[34] or house mice (*Mus musculus*) fully rescued Huh7.5 *PLSCR1*-KO cells, mimicking complementation by human PLSCR1 (Fig. 2g and Extended Data Fig. 2f). Bat, mouse and human PLSCR1 orthologues can thus be functionally exchanged against SARS-CoV-2. Further evidence of evolutionary conservation was found in mouse immortalized type I alveolar cells ectopically expressing human ACE2 (LET1-ACE2) to allow SARS-CoV-2 infection. Chromosomal disruption of the mouse *Plscr1* locus rendered LET1-ACE2 cells susceptible to SARS-CoV-2 challenge (Extended Data Fig. 2g,h). SARS-CoV-2 restriction was also evident in multiple human epithelial and stromal cell populations that serve as major sites of PLSCR1 expression, including the respiratory tract epithelium (Extended Data Fig. 3a). Expressing PLSCR1 in primary human

tracheal epithelial cells (hTEpiCs) potently inhibited SARS-CoV-2 infection (Fig. 2h and Extended Data Fig. 3b), and CRISPR–Cas9 disruption of the *PLSCR1* locus in human lung (Calu-3 and A549-ACE2) or tonsillar epithelium (UT-SCC-60A-ACE2), cervical epithelial cells (HeLa CCL2-ACE2) or skin keratinocytes (HaCaT-ACE2) all resulted in susceptibility to SARS-CoV-2 (Fig. 2i and Extended Data Fig. 3d–f). Thus, the restrictive profile of PLSCR1 across diverse cell types and species reveals its evolutionary importance for host defence against SARS-CoV-2 infection.

To corroborate the strong effect of PLSCR1, we compared it with other ISGs (IFITM3, NCOA7, LY6E and CD74) that have been reported to restrict SARS-CoV-2 (25,35–38). Overexpressing them to similar levels in Huh7.5 cells showed that PLSCR1 exerted a stronger antiviral effect than either IFITM3 or NCOA7 (10.5-fold inhibition versus 1.5–2.0-fold) (Fig. 2i and Extended Data Fig. 3f). Ectopic expression of the CD74 p41 isoform resulted in the strongest activity (55-fold inhibition); however, its natural expression is mostly confined to immune cells, whereas PLSCR1 protects a wider range of cell types against SARS-CoV-2. Examination of PLSCR1 and LY6E showed comparable protection in primary hTEpiCs (Fig. 2h). Given that LY6E was also identified as a top candidate in our LoF screen in A549-ACE2 cells (Fig. 1e), we knocked it out for direct comparison with *PLSCR1*-KO cells. Deletion of *PLSCR1* yielded greater susceptibility under basal conditions, whereas *LY6E* contributed more in IFNγ-activated A549-ACE2 cells (Fig. 2j and Extended Data Fig. 3g,h). Dual *PLSCR1* and *LY6E* deficiency further increased susceptibility versus either alone, revealing independent yet co-operative effects between these two ISGs. Thus, PLSCR1 can act synergistically with other ISGs to enhance anti-SARS-CoV-2 resistance.

## PLSCR1 blocks coronavirus entry

Next, we sought to identify which step of the viral life cycle is targeted by PLSCR1. First, we challenged Huh7.5 cells with a replication-incompetent HIV-1 vector pseudotyped with SARS-CoV-2 spike proteins. Loss of *PLSCR1* markedly increased the infectivity of pseudoviruses containing spike proteins not only from SARS-CoV-2 USA-WA1/2020, but also from Delta B.1.617.2 (by 9.9-fold) or Omicron B1.1.529 (by 11.0-fold), indicating that PLSCR1 is essential for inhibiting virus entry by VOCs (Fig. 3a,b).

PLSCR1 also restricted pseudoviruses containing spike proteins from SARS-CoV, MERS-CoV or BatCoV-WIV-1, all highly pathogenic coronaviruses (Fig. 3c). Likewise, engineered loss of *Plscr1* rendered mouse LET1 cells more permissive to infection by intact mouse coronavirus (MHV) (Fig. 3d), which belongs to the same β-coronavirus genus as the SARS-CoV-2, SARS-CoV, MERS-CoV and BatCoV-WIV-1 strains. Thus, PLSCR1 appears to be crucial for blocking the entry of highly pathogenic β-coronaviruses.

This preferential blockade was further demonstrated by experiments using spike proteins from less virulent seasonal HCoV-229E, HCoV-NL63, HCoV-OC43 or HCoV-HKU1 strains. Here, *PLSCR1* deletion had more modest effects on their entry (Extended Data Fig. 4a). It was also largely dispensable for EBoV (Ebola virus), HCV (hepatitis C virus) and VSV (vesicular stomatitis virus) pseudoviruses as well as live DENV-1 (dengue virus type 1) or HSV-1 (herpes simplex virus 1) (Fig. 3c and Extended Data Fig. 4b,c). Compensation by other restriction factors could partly explain the loss of viral sensitivity (IFITM3 was over 90 times more potent than PLSCR1 against EBoV; Extended Data Fig. 4d). Even so, PLSCR1 appeared most effective in blocking host-cell invasion by highly pathogenic coronaviruses.

## PLSCR1 disrupts virus–membrane fusion

Next we addressed how PLSCR1 blocks coronavirus invasion. SARS-CoV-2 uses two fusion routes to invade host cells: a cathepsin-dependent endosomal fusion pathway and a TMPRSS2-dependent cell-surface fusion pathway[39] (Extended Data Fig. 4e). In ACE2-expressing cells

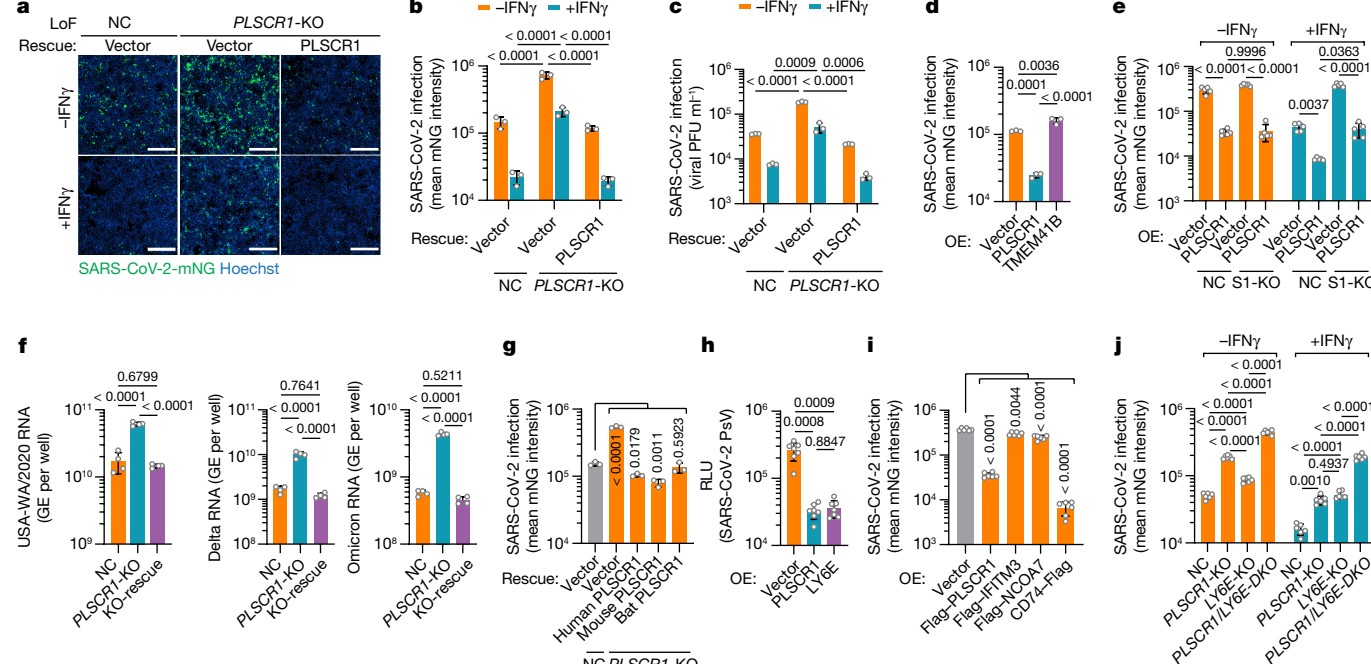

**Fig. 2 | PLSCR1 is an evolutionarily conserved defence protein against coronavirus infection. a**, Representative images showing the infectivity of SARS-CoV-2-mNG in resting or IFNγ-activated NC, *PLSCR1*-KO and *PLSCR1*-KO-complemented Huh7.5 cells at 48 hours post-infection (hpi) (MOI = 1). **b**, Stable genetic complementation. Quantification of SARS-CoV-2-mNG fluorescent intensity in Huh7.5 cells (*n* = 3). **c**, Complementation against virus production. Plaque assay showing the production of infectious viruses in Huh7.5 cells infected for 48 h at an MOI of 0.5 (*n* = 3). **d**, SARS-CoV-2-mNG infection in Huh7.5 cells overexpressing (OE) the indicated proteins (*n* = 3). **e**, SARS-CoV-2-mNG infection in control or *STAT1*-KO (S1-KO) Huh7.5 cells overexpressing PLSCR1 in the presence or absence of IFNγ (*n* = 5). **f**, Intracellular levels of SARS-CoV-2 RNA in Huh7.5 cells at 24 hpi. Primers detecting nucleocapsid were used to amplify viral RNA. GE, genome equivalents. Cells were infected with SARS-CoV-2 USA-WA1 (left), Delta B.1.617.2 (middle) or Omicron BA.1 (right) variants at an MOI of 0.5 (*n* = 4). **g**, SARS-CoV-2-mNG infection in

*PLSCR1*-KO Huh7.5 cells complemented with vector control or human (*Homo sapiens*), mouse (*Mus musculus*) or bat (*Rhinolophus sinicus*) orthologues of PLSCR1 (*n* = 3). **h**, Pseudovirus (PsV) infection in hTEpiCs overexpressing the indicated proteins. An HIV-1-based luciferase-expressing vector pseudotyped with SARS-CoV-2 spike (Omicron) was quantified by luciferase activity at 48 hpi. RLU, relative light units (*n* = 6). **i**, Quantification of SARS-CoV-2 infection in Huh7.5 cells stably overexpressing the indicated ISGs (MOI = 1, 48 hpi) (*n* = 6). **j**, SARS-CoV-2 infection in A549-ACE2 cells of the indicated genotypes (DKO, double knockout) in the presence or absence of IFNγ (100 U ml⁻¹) at 24 hpi. (MOI = 0.2) (*n* = 6). Data are mean ± s.d. *P* values from one-way ANOVA followed by Tukey's multiple comparison test in **b**–**d**,**e** (−IFNγ group) and **f**,**g**,**j** (+IFNγ group) and Brown–Forsythe or Welch ANOVA with Dunnett's post-hoc test in **e** (+IFNγ group) and **h**,**i**,**j** (−IFNγ group). Scale bars (**a**), 500 μm. All experiments performed three times, except **b** (five times).

that lack TMPRSS2 (Huh7.5 and A549-ACE2), viral uptake proceeds entirely through the cathepsin-dependent fusion pathway, in which the spike protein is processed by cysteine proteases to unmask a fusion peptide for exit from the endolysosome into the host cytosol[39]. Inhibiting this pathway with saturating concentrations of E-64d (a cysteine protease inhibitor) reversed the susceptible phenotype of Huh7.5 or A549-ACE2 *PLSCR1*-KO cells, whereas the TMPRSS2 inhibitor camostat had no effect (Extended Data Fig. 4f,g). Thus, PLSCR1 strongly affects the endosomal pathway.

Next, we introduced TMPRSS2 into Huh7.5 cells along with E-64d treatment to test viral entry solely by the cell-surface fusion pathway. Here, PLSCR1 also exerted SARS-CoV-2 restriction (Extended Data Fig. 4h). Similar results were found in human Calu-3 cells, in which cell-surface fusion predominates owing to high endogenous TMPRSS2 expression. Generating *PLSCR1*-KO Calu-3 cells resulted in a 5.5-fold increase in overall SARS-CoV-2 susceptibility. Treating *PLSCR1*-KO Calu-3 cells with E-64d to force entry through the cell-surface route yielded a more modest 3.0-fold increase in susceptibility, whereas silencing TMPRSS2 with camostat led to a 14.0-fold increase in endosomal susceptibility (Extended Data Fig. 4i). Hence, PLSCR1 primarily interferes with endosomal entry of SARS-CoV-2, although it can restrict the TMPRSS2 fusion pathway as well.

Endosomal entry consists of several steps: viral receptor binding, internalization, vesicle trafficking, spike cleavage and virus–endosome

fusion[39]. We first checked the levels of ACE2 on the plasma membrane for viral receptor binding in *PLSCR1*-KO cells. Surface proteins labelled with membrane-impermeable biotin were subjected to streptavidin pulldown and immunoblotting. PLSCR1 deficiency did not alter the amount of ACE2 reaching the surface or the total expression of ACE2 (Fig. 3e). Next, we tested whether PLSCR1 blocks SARS-CoV-2 binding and internalization. Viral attachment at 4 °C for 1 h was almost identical for both wild-type and *PLSCR1*-KO cells (Fig. 3f). Shifting cells to 37 °C for 30 min enabled the internalization of bound SARS-CoV-2 followed by trypsin digestion of uninternalized virus. PLSCR1 deficiency did not affect this parameter (Fig. 3f). The number of SARS-CoV-2 particles entering *PLSCR1*-KO lysosomes similarly resembled wild-type cells because blocking viral–lysosome fusion and the subsequent release of viral RNA with hydroxychloroquine (HCQ) or E-64d resulted in similar levels of spike and nucleocapsid signals inside LAMP1⁺ vesicles (Extended Data Fig. 5a,b). Endosomal acidification and spike cleavage were likewise unaffected in *PLSCR1*-KO cells (Fig. 3g and Extended Data Fig. 5c). Thus, PLSCR1 did not block viral receptor expression, binding, internalization, trafficking or spike cleavage.

Finally, we used a split-NanoLuc-reporter-based assay to test virus–endosome fusion; here, the reporter undergoes self-complementation of luciferase activity upon reaching the cytosol[40]. Huh7.5 or 293T-ACE2 cells expressing a LgBiT fragment were infected with a SARS-CoV-2-spike-expressing pseudovirus bearing the HiBiT fragment.

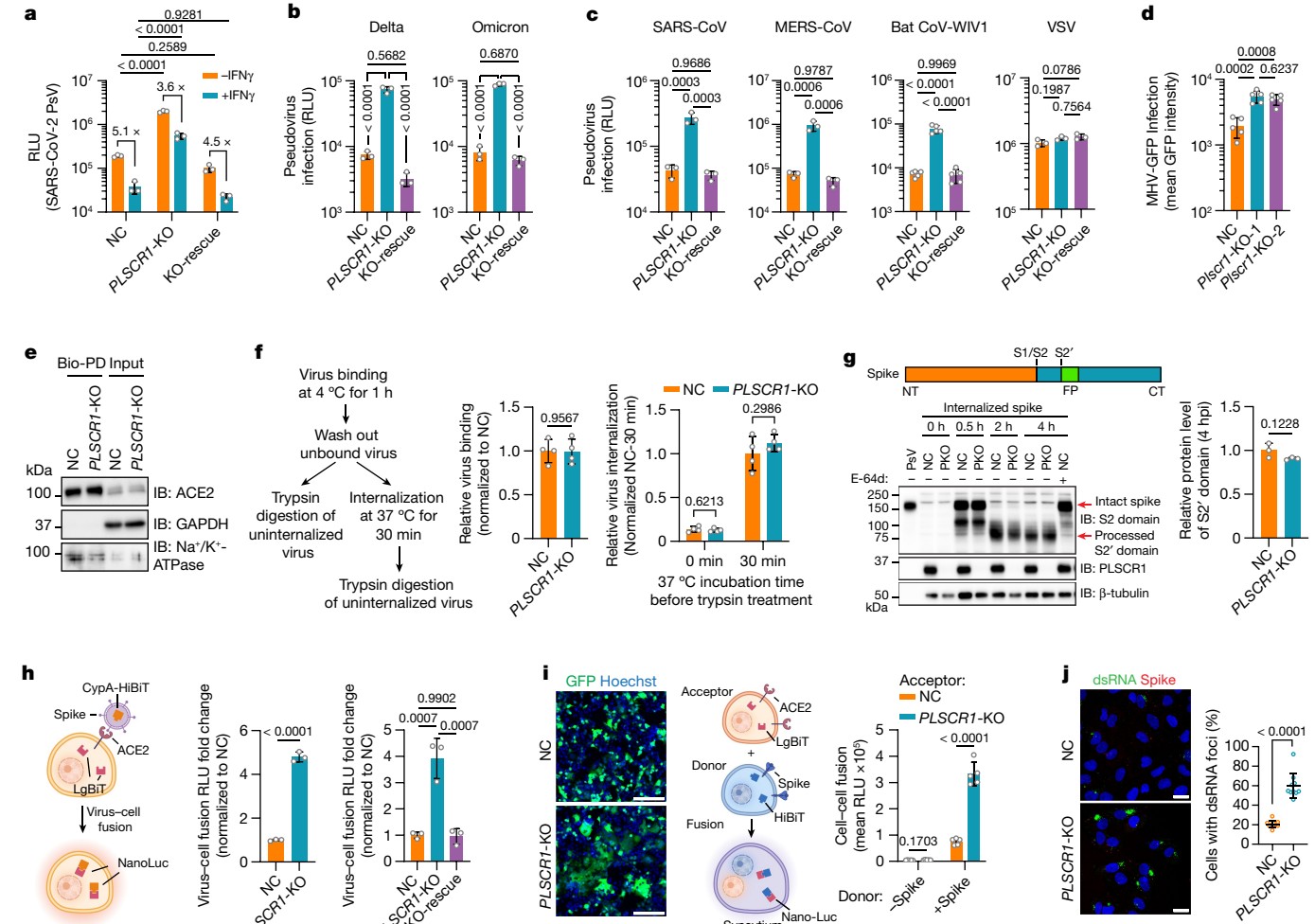

**Fig. 3 | PLSCR1 blocks coronavirus entry and spike-mediated membrane fusion with host cells. a**, Virus entry into Huh7.5 cells of the indicated genotypes. Cells were challenged with an HIV-1-based luciferase-expressing vector pseudotyped with SARS-CoV-2 spike (USA-WA1/2020) and assayed for luciferase activity at 48 hpi. ($n = 3$). **b**, Virus entry into Huh7.5 cells inoculated with pseudovirus bearing spike proteins from SARS-CoV-2 Delta (left) or Omicron (right) ($n = 3$). **c**, Viral entry efficiency into Huh7.5 cells inoculated with PsVs bearing fusion proteins from SARS-CoV ($n = 3$), MERS-CoV ($n = 3$), bat CoV-WIV1 ($n = 5$) or VSV ($n = 3$). **d**, Quantification of MHV-A59-GFP fluorescent intensity in LET1-ACE2 cells (MOI = 0.1, 24 hpi, $n = 5$). **e**, Levels of ACE2 in NC or *PLSCR1*-KO A549-ACE2 cells using surface biotinylation. Membrane-impermeable biotin was used to label cell-surface proteins, followed by streptavidin pulldown. Surface ACE2 was then quantified by western blot. GAPDH was used as a cytosol marker. Bio-PD, biotin pulldown. **f**, Virus binding and internalization in NC or *PLSCR1*-KO Huh7.5 cells. The amount of viral RNA in NC cells was normalized to 1 ($n = 4$). **g**, Cleavage of the SARS-CoV-2 spike protein in NC or *PLSCR1*-KO (PKO) A549-ACE2 cells infected with PsV carrying SARS-CoV-2 spike for the indicated time periods. Cells treated with E-64d (20 μM) served as a negative control ($n = 3$). IB, immunoblot; NT, N-terminus; FP, fusion peptide; CT, C-terminus. **h**, Virus-cell fusion assay in Huh7.5 (middle) and 293T-ACE2 (right) cells ($n = 3$). Fusion was measured by complemented (NanoLuc) activity. **i**, Left, representative images showing syncytia formation after co-culturing Huh7.5 cells of the indicated genotypes and 293T cells expressing SARS-CoV-2 spike and EGFP. Right, quantification of fusion activity in Huh7.5 cells of the indicated genotypes. Cell–cell fusion was measured by complemented NanoLuc activity ($n = 5$). **j**, Left, representative images showing the formation of dsRNA foci in A549-ACE2 cells at 3 hpi. (MOI = 5). dsRNA was detected with a monoclonal rJ2 antibody (MABE1134). Right, percentage of cells with dsRNA foci. $n = 10$ image fields (NC, 209 cells; *PLSCR1*-KO, 210 cells analysed). Data are mean ± s.d. $P$ values from two-sided Student's $t$-test in **f**,**g**,**h** (middle) and **i** (–Spike group), two-sided Student's $t$-test with Welch's correction in **i** (+Spike group), one-way ANOVA followed by Tukey's multiple comparison test in **a**–**d**,**h** (right) and two-sided Mann–Whitney test in **j**. Scale bars, 200 μm (**i**) and 20 μm (**j**). All experiments performed three times, except **a**,**g**,**i** (four times).

Reconstituted LgBiT-HiBiT bioluminescence was greatly increased in *PLSCR1*-KO cells and genetically rescued through *PLSCR1* complementation (Fig. 3h and Extended Data Fig. 5d). To establish whether PLSCR1 blocked membrane fusion itself, we performed a syncytium assay in which donor cells expressing SARS-CoV-2 spike and the HiBiT fragment were co-cultured with acceptor cells expressing ACE2 and the LgBiT fragment (Fig. 3i). When *PLSCR1*-KO cells served as the acceptor population, spike-mediated cell–cell fusion increased significantly (Fig. 3i). PLSCR1 overexpression reduced this response (Extended Data Fig. 5e). Thus, PLSCR1 directly prevents membrane fusion triggered by the SARS-CoV-2 spike protein.

To confirm this function, we monitored the release of viral RNA after the fusion step. Double-stranded RNA (dsRNA), an intermediate product of viral replication, serves as a surrogate marker for the entry of SARS-CoV-2 genomic RNA into the host cytosol. Massive dsRNA foci were observed in 60% of *PLSCR1*-KO cells by 180 min after infection, when viral release has occurred, but the production of new virions has not been completed (Fig. 3j). By contrast, only 21% of control cells had viral dsRNA foci, which were considerably smaller. Nearly 80% of *PLSCR1*-KO cells exhibited intense and widely dispersed nucleocapsid signals 60 min later, indicating completed SARS-CoV-2 viral RNA release and viral protein synthesis (Extended Data Fig. 5f). In 76% of control

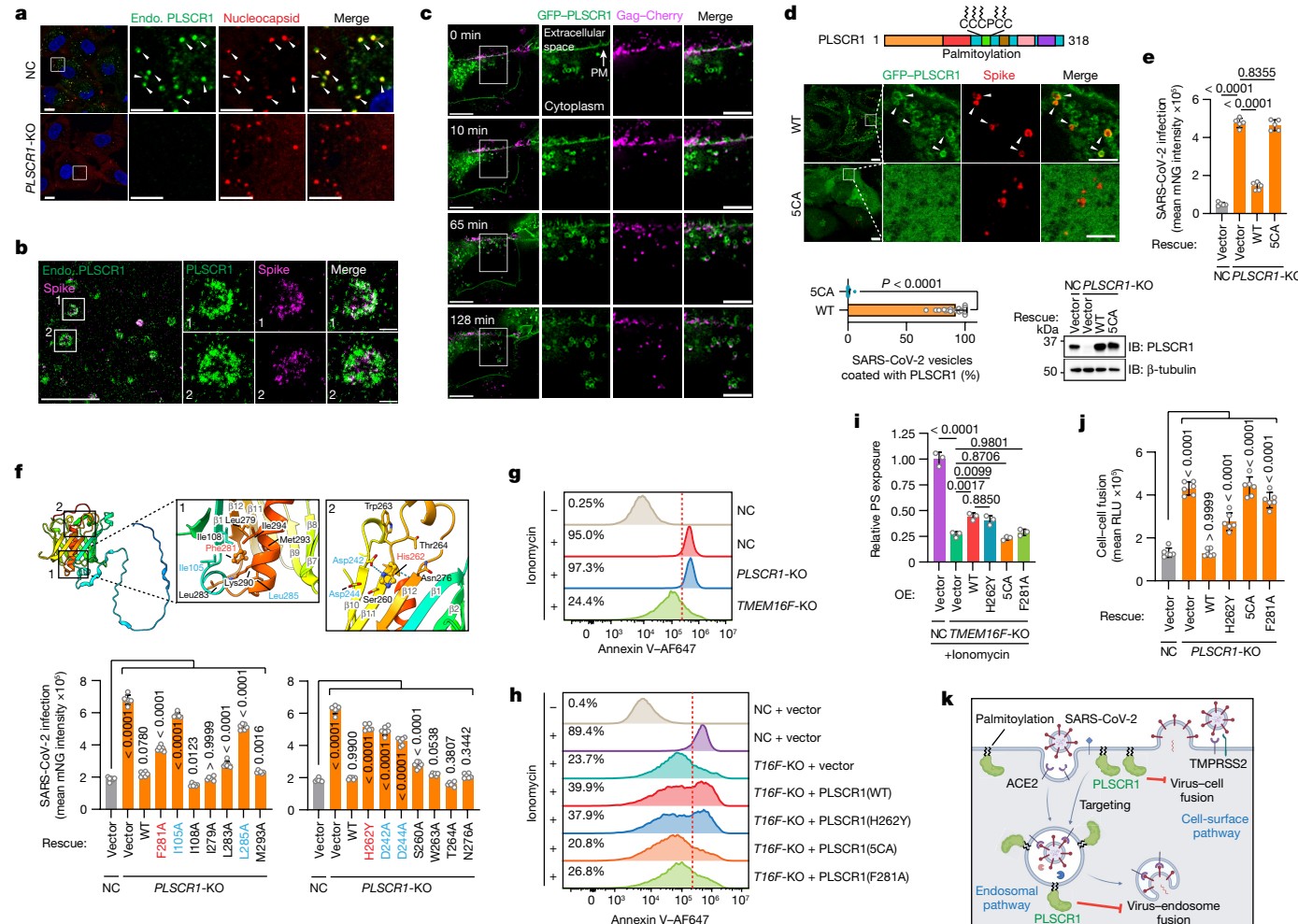

**Fig. 4 | The lipid scramblase and antiviral activities of PLSCR1 are uncoupled. a**, Confocal images showing the localization of endogenous (endo.) PLSCR1 and SARS-CoV-2 nucleocapsid in NC or *PLSCR1*-KO A549-ACE2 cells infected with SARS-CoV-2 (2 hpi, MOI = 25). **b**, W-4Pi-SMS nanoscopy of endogenous PLSCR1 and SARS-CoV-2 spike detected at single-molecule resolution in A549-ACE2 cells infected with SARS-CoV-2 (2 hpi, MOI = 25). **c**, Dynamic formation of PLSCR1-coated SARS-CoV-2-containing vesicles. Time-lapse images were obtained at 1-min intervals for around 2 h and snapshots at the indicated time points are presented. A549-ACE2 cells were infected with SARS-CoV-2-PsV-Cherry. PM, plasma membrane. **d**, Localization of wild-type (WT) PLSCR1 or mutant PLSCR1(5CA) (C[184]CCPCC[189] to AAAPAA) on SARS-CoV-2-containing vesicles. *PLSCR1*-KO A549-ACE2 cells stably expressing GFP–PLSCR1(WT) or GFP–PLSCR1(5CA) were infected with SARS-CoV-2 for 2 h (MOI = 25) (WT, *n* = 24 cells; 5CA, *n* = 28 cells). **e**, Quantification of SARS-CoV-2 infection in Huh7.5 cells expressing the indicated mutants (MOI = 1, 48 hpi, *n* = 5). **f**, Top, AlphaFold2 prediction of surrounding amino acid residues of the Phe[281] and His[262] sites. Rainbow-coloured from N terminus (blue) to C terminus

(red). Bottom, comparative SARS-CoV-2 infection in control or *PLSCR1*-KO Huh7.5 cells expressing the indicated mutants (MOI = 1, 48 hpi, *n* = 6). **g**, FACS plots showing PS externalization in NC, *PLSCR1*-KO or *TMEM16F*-KO A549-ACE2 cells in the absence or presence of 10 μM ionomycin. Ionomycin is a membrane-permeable Ca²⁺ carrier that increases intracellular Ca²⁺ levels, which triggers PS externalization in a percentage of cells (threshold, dotted line). **h**, FACS plots showing PS externalization in NC or *TMEM16F*-KO A549-ACE2 cells stably overexpressing the indicated PLSCR1 mutants in the absence or presence of 10 μM ionomycin. **i**, Relative PS externalization activity in NC A549-ACE2 cells treated with ionomycin, normalized to 1 (*n* = 3). **j**, Cell–cell fusion assay in Huh7.5 cells stably expressing the indicated PLSCR1 mutants co-cultured with 293T cells expressing SARS-CoV-2 spike (*n* = 6). **k**, Model of SARS-CoV-2 restriction by PLSCR1. All data are mean ± s.d. *P* values from two-sided Mann–Whitney test in **d** and one-way ANOVA followed by Tukey's multiple comparison test in **e,f,i,j**. Scale bars 10 μm (**a,c,d**, main), 5 μm (**b**, main and **a,c,d**, inlays) and 500 nm (**b**, inlays). All experiments performed three times, except **a** (five times).

cells, however, the nucleocapsid signal was still inside vesicles, indicating unfinished viral RNA release. Collectively, our results indicate that PLSCR1 disrupts the fusion of SARS-CoV-2 with host-cell membranes to prevent subsequent viral RNA release and protein synthesis within the cytosol.

## PLSCR1 targets SARS-CoV-2 vesicles

To delineate how PLSCR1 impedes SARS-CoV-2 fusion, we combined nanoscale imaging with protein mutagenesis to identify the membrane determinants required. First, synchronized SARS-CoV-2

infection enabled the subcellular localization of endogenous PLSCR1 to be tracked at designated time intervals in A549-ACE2 cells. PLSCR1-enriched foci were observed as early as 30 min after infection, which completely overlapped viral particles detected by anti-spike or anti-nucleocapsid antibodies (Fig. 4a and Extended Data Fig. 6a–d). PLSCR1 targeting appeared to be specific for viral entry because colocalization was lost at later times once viral replication and transcription complexes (denoted by dsRNA foci) were assembled along with new nucleocapsid or spike-protein synthesis in the cytosol (Extended Data Fig. 6a,b,e). In addition, PLSCR1 did not colocalize with human transferrin–AF488 taken up by clathrin-mediated endocytosis (Extended

Data Fig. 6f). Thus, PLSCR1 targeted the membrane entry pathway of SARS-CoV-2 rather than all endocytosed cargo.

The recruitment of PLSCR1 to SARS-CoV-2 compartments was examined in greater detail using whole-cell 4Pi single-molecule switching (W-4Pi-SMS) nanoscopy, which resolves three-dimensional (3D) structures to around 20 nm isotropically throughout entire mammalian cells[41]. W-4Pi-SMS found endogenous PLSCR1-coated SARS-CoV-2 virions in 500-nm–800-nm vesicles (Fig. 4b and Supplementary Videos 1 and 2). Some of these PLSCR1+ vesicles originated from the plasma membrane in live imaging of *PLSCR1*-KO cells expressing GFP–PLSCR1 and infected with SARS-CoV-2 pseudovirus containing mCherry-tagged Gag to mimic spike-mediated entry (Fig. 4c and Supplementary Videos 3 and 4). By 120 min after infection, such vesicles colocalized with LAMP1 (74.5%; late endosomes or lysosomes), CD63 (52.8%; late endosomes or multivesicular bodies) or IFITM3, a resident lysosomal protein and ISG that has been reported to restrict SARS-CoV-2 infection[38,42] (Extended Data Fig. 7a–d). Notably, the anti-SARS-CoV-2 activity of PLSCR1 greatly exceeded that of IFITM3, suggesting that its mechanism differs from that of other restriction factors sharing this membrane (Fig. 2j).

PLSCR1 contains a five-cysteine palmitoylation motif (C[184]CCPCC[189]) that could help anchor it to SARS-CoV-2-containing vesicles. Substitution of these cysteines is reported to cause dispersed cytosolic localization and increased nuclear import of PLSCR1 in uninfected cells[43]. Mutating all five cysteine residues to alanine (labelled 5CA) completely abolished the localization of PLSCR1 to the plasma membrane and SARS-CoV-2-containing vesicles after infection (Fig. 4d). PLSCR1(5CA) also did not protect against infection when reintroduced into *PLSCR1*-KO cells (Fig. 4e). Thus, palmitoylation and membrane localization are essential to the anti-SARS-CoV-2 function of PLSCR1.

## Structural basis of PLSCR1 activity

Palmitoylation enables PLSCR1 to target SARS-CoV-2-containing vesicles, but other protein regions might interfere with subsequent membrane fusion. Nanosecond molecular dynamics simulation (MDS) analysis showed movement in distal hydrophobic loop regions once palmitoylated PLSCR1 was docked to the plasma membrane (Supplementary Video 5). AlphaFold2 and structural homology modelling[44] predict that PLSCR1 has a flexible N-terminal domain and 12-stranded membrane β-barrel in which the C-terminal hydrophobic helix is buried (Fig. 4f). Reintroducing PLSCR1 truncations into *PLSCR1*-KO cells found that deletion of the first β-strand of the β-barrel (amino acids 86–118) or the C-terminal hydrophobic helix (amino acids 291–318) was most detrimental to antiviral activity (Extended Data Fig. 8a–d). Fine-mapping revealed that Phe[281] (a residue essential for Ca[2+] binding and phospholipid scramblase activity)[45] and the spatially adjacent residues Ile[105] and Leu[285] within the hydrophobic loops were also crucial, indicating that these regions and the β-barrel contribute to the antiviral and possibly anti-fusogenic properties of PLSCR1, as suggested from MDS modelling (Fig. 4f and Extended Data Fig. 8e).

The importance of the PLSCR1 β-barrel domain is reinforced by previous whole-genome sequencing that linked a missense single-nucleotide polymorphism in the human *PLSCR1* locus (rs343320; His[262]Tyr) with susceptibility to severe COVID-19 disease[3,4] (Fig. 4f). His[262] is located at the base of the 11th β-strand of the β-barrel. We found that the COVID-19-associated H[262]Y mutation significantly impaired anti-SARS-CoV-2 activity in human primary TEpiCs or Huh7.5 cells (Fig. 4f and Extended Data Fig. 8f–h). Neither H[262]Y nor F[281]A affected the targeting of PLSCR1 to SARS-CoV-2-containing vesicles, placing their effects downstream of membrane docking (Extended Data Fig. 8i). His[262] is also found in a non-classical nuclear localization signal that has been reported to interact with the nuclear transporter α-importin as an incoming transcription factor[46]. Introducing KKHA (Lys[258]Lys[261]His[262] to Ala) mutations did not affect antiviral activity, however, and transcriptional RNA-sequencing (RNA-seq) profiles in *PLSCR1*-KO cells

were identical to controls (Extended Data Fig. 9a,b). Indeed, PLSCR1 was not detected within the nucleus at any stage during SARS-CoV-2 infection. Thus, the COVID-19-associated H[262]Y mutation probably alters β-barrel surface or conformational properties rather than PLSCR1 nuclear translocation.

To test this possibility, we engineered substitutions at the His[262] site. Sequence comparison with mouse or bat PLSCR1 orthologues found that His[262] is naturally substituted with glutamine (Gln), which does not affect their anti-SARS-CoV-2 activities (Extended Data Fig. 9d). Engineering the COVID-19-associated Tyr mutation (bat, Q286Y; mouse, Q271Y), however, impaired such activity. Other aromatic (Phe or Trp) or basic amino acid residues (Lys or Arg) at His[262] likewise diminished the anti-SARS-CoV-2 effect of human PLSCR1, as did mutations in the spatially adjacent Asp[242] or Asp[244] (Fig. 4f and Extended Data Fig. 9d,e). His[262] therefore appears to have a role in maintaining the integrity of the β-barrel; indeed, root-mean-square fluctuation (RMSF) analysis of the H[262]Y mutation found considerable instability along with changes to the local hydrogen bond network in MDS studies (Extended Data Fig. 10a,b).

## Uncoupling lipid scramblase activity

Besides the COVID-19-associated H[262]Y mutation, a second substitution at Phe[281]Ala also impaired SARS-CoV-2 restriction (Fig. 4f). This residue has been shown to be important for Ca[2+]-dependent phosphatidylserine (PS) exposure[46]. Given that PS exposure mediated by other lipid scramblases such as TMEM16F affects SARS-CoV-2-driven syncytia formation[47], we asked whether PLSCR1 interferes with fusion by altering PS exposure. We assayed the Ca[2+]-induced externalization of PS in *PLSCR1*-KO A549 cells and found no defects in these cells compared with wild-type controls (Fig. 4g and Extended Data Fig. 10c). By contrast, *TMEM16F*-KO cells were profoundly defective, suggesting that the scramblase activity of TMEM16F could mask contributions from PLSCR1 (Fig. 4g and Extended Data Fig. 10c,d). Indeed, overexpressing PLSCR1 in the *TMEM16F*-KO background partially restored the Ca[2+]-induced externalization of PS, indicating weak scramblase activity (Fig. 4h,i and Extended Data Fig. 10e). As expected, the Ca[2+]-binding mutant PLSCR1(F281A) did not rescue scramblase activity, and neither did PLSCR1(5CA) which cannot associate with the plasma membrane. Unexpectedly, however, the COVID-19-associated H[262]Y mutation rescued PS externalization to the same levels as the wild-type protein, despite being defective for anti-SARS-CoV-2 activity (Fig. 4h,i and Extended Data Fig. 10e). Thus, the lipid scramblase activity of PLSCR1 can be uncoupled from its anti-SARS-CoV-2 function.

Subsequent cell–cell fusion assays found that the COVID-19-associated PLSCR1(H262Y) mutant did not inhibit membrane fusion despite retaining lipid scramblase activity (Fig. 4j). PS exposure therefore appears dispensable, whereas the β-barrel H[262] residue is required for anti-fusogenic activity to restrict SARS-CoV-2. Wild-type PLSCR1 rescued anti-fusogenic activity, whereas PLSCR1(5CA) and PLSCR1(F281A) were ineffective—the former because it cannot localize to the plasma membrane; and the latter probably because it resides in a β-barrel hydrophobic loop region, rather than owing to the loss of its scramblase activity (Fig. 4j). Notably, the ability of PLSCR1 to interfere with fusion did not involve changes in membrane bending rigidity. Optical tweezer assays found that the bending rigidity was similar in giant plasma-membrane vesicles (GPMVs) generated from *PLSCR1*-KO cells that expressed either wild-type or 5CA variants of GFP–PLSCR1 (Extended Data Fig. 10f–h). Thus, PLSCR1 differs from IFITM3 that alters rigidity[48], further underscoring its distinct mode of action in opposing both fusion pathways used by SARS-CoV-2 (Fig. 4k).

## Discussion

We have identified human PLSCR1 as a crucial host-defence factor against SARS-CoV-2 infection. Notably, orthologues of PLSCR1 in bats

and mice also restrict SARS-CoV-2, and human PLSCR1 can inhibit highly pathogenic bat and mouse β-coronaviruses. Thus, the anti-coronavirus activity of PLSCR1 seems to be functionally conserved across mammalian evolution. Previous work suggested that PLSCR1 can affect other human viruses by inhibiting (influenza A, hepatitis B, HIV, Epstein Barr, and cytomegaloviruses)[26–30] or promoting (hepatitis B, herpes simplex virus)[22,49] replication. In the cases in which PLSCR1 was restrictive, several possible mechanisms were described, including activating type I IFN signalling, degrading viral proteins, or blocking viral transcription and nuclear import.

We found that endogenous PLSCR1 directly targets nascent SARS-CoV-2-containing vesicles to prevent virus–membrane fusion and the release of viral RNA into the host-cell cytosol. This mechanism not only blocked endosomal entry but also interfered with TMPRSS2-dependent cell-surface fusion. PLSCR1 therefore has the potential to protect against emerging SARS-CoV-2 variants and other coronaviruses that use either route of entry. Indeed, PLSCR1 seemed to be more restrictive than IFITM3, which alters membrane rigidity, or NCOA7, which interacts with the vacuolar ATPase to promote the acidification and degradation of virus particles[36,38]. In addition, PLSCR1 did not control cathepsin activity to inhibit spike processing like CD74 (ref. 37), and its anti-SARS-CoV-2 activity was separable from that of LY6E, which blocks entry by an unknown mechanism[25]. PLSCR1 thus appears to occupy a unique position within the antiviral repertoire that is mobilized during COVID-19 infection[35].

PLSCR1 requires its β-barrel domain to disrupt the fusion of the virus with the host cell once docked to the target membrane by palmitoylation. Here it is likely to occupy plasma-membrane microdomains that are used by coronaviruses to be directed to distinct subpopulations of SARS-CoV-2-containing vesicles. Subsequent PLSCR1 β-barrel clustering could then act as a barrier to insertion of the viral fusion peptide or arrest the hemi-fusion diaphragm. IFITM3 achieves the latter objective by generating negative curvature within liquid-disordered membrane domains[48]. PLSCR1 probably partitions to the liquid-ordered phase, however, given its palmitoyl anchorage, and it seems to operate on both high-curvature (endosome) and low-curvature (plasma membrane) membranes to inhibit spike-mediated fusion. Atomistic simulations also showed that distal hydrophobic loop regions in palmitoylated PLSCR1 become exposed when docked to the membrane; these loops might offer an interactive surface to bind and sequester other fusion-competent partners—for example, tetraspannins—as accessory proteins for coronavirus entry[50].

Structural modelling showed that the short β-barrel cannot physically span the phospholipid bilayer, discounting the likelihood that PLSCR1 acts as an ion channel or a lipid transporter, as has been suggested for other scramblase family members[51]. Indeed, the central β-barrel of PLSCR1 is vastly different to the 'butterfly-fold' or rhomboidal α-helical dimers containing a permeation pore that is typical of most $Ca^{2+}$-dependent lipid scramblases[51]; this difference could explain why PLSCR1 has only weak enzymatic activity. Such activity is dispensable for the anti-SARS-CoV-2 function of PLSCR1. Our study provides an emerging mechanistic framework for PLSCR1 in blocking spike-mediated fusion by virulent coronaviruses. It highlights the PLSCR1 β-barrel as a major determinant of cell-autonomous resistance to this group of global pathogens and aids our understanding of what constitutes protective immunity within the human IFN response.

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

## Methods

### Cell lines

Huh7.5 (human hepatocellular carcinoma, a gift from C.B.W.), A549-ACE2 (human alveolar basal epithelial carcinoma cells, BEI Resources NR-53821), Vero E6 (African green monkey kidney epithelial cells, ATCC CRL-1586), HEK293T (human embryonic kidney cells, ATCC CRL-3216), HeLa (human cervical adenocarcinoma cells, ATCC CCL-2), Tonsil (human tonsillar epithelial cells, UT-SCC-60A), HaCaT (immortalized human keratinocytes, a gift from D. DiMaio) and LET1 (mouse lung epithelial type I cells, BEI Resources NR-42941) cells were cultured in Dulbecco's modified Eagle's medium (DMEM) supplemented with 10% heat-inactivated fetal bovine serum (FBS) and 1% penicillin–streptomycin (pen-strep). hTEpiCs (ScienCell 3220) were cultured in bronchial epithelial cell medium (ScienCell 3211) supplemented with 1% bronchial epithelial cell growth supplement (ScienCell 3262). Calu-3 (human lung adenocarcinoma cells, ATCC HTB-55) were cultured in Eagle's minimum essential medium (EMEM; ATCC 30-2003) with 10% FBS and 1% pen-strep. All cells, unless otherwise stated, were cultured at 37 °C and incubated with 5% $CO_2$.

HEK293T-ACE2, HeLa-ACE2, Tonsil-ACE2, HaCaT-ACE2 and LET1-ACE2 cells were generated by stably expressing human ACE2 in the aforementioned original cell lines. In brief, lentiviruses were packaged in HEK293T cells by transfecting the cells with pLV-EF1a-hACE2-Hygro, psPAX2 and VSVG. Forty-eight hours after transfection, the medium was collected, filtered through a 0.45-μm filter and added to target cells for 24 h. Cells were subsequently selected with hygromycin for 7 days before further treatments. ACE2 expression was tested by western blots as well as virus infection.

### Antibodies and reagents

Rabbit anti-GAPDH monoclonal antibody (60004-1-Ig), mouse anti-GFP tag monoclonal antibody (66002-1-Ig), rabbit anti-PLSCR1 polyclonal antibody (11582-1-AP), mouse anti-Halo tag monoclonal antibody (28a8), rabbit anti-TMEM41B polyclonal antibody (29270-1-AP) and rabbit anti-IFITM3 polyclonal antibody (11714-1-AP) were obtained from Proteintech. Rabbit anti-Na,K-ATPase polyclonal antibody (3010S), rabbit anti-Flag tag monoclonal antibody (14793S) and rabbit anti-β-tubulin monoclonal antibody (2128S) were obtained from Cell Signaling. Goat anti-ACE2 polyclonal antibody (AF933) was purchased from R&D Systems. Mouse anti-PLSCR1 monoclonal antibody (MABS483), mouse anti-dsRNA monoclonal antibody (MABE1134), rabbit anti-TMEM16F polyclonal antibody (HPA038958), sheep anti-mouse IgG horseradish-peroxidase-conjugated secondary antibody (GENXA931-1ML) and sheep anti-rabbit IgG horseradish-peroxidase-conjugated secondary antibody (GENA934-1ML) were obtained from Sigma. Rabbit anti-SARS-CoV-2 nucleocapsid monoclonal antibody (40143-R019) and rabbit anti-SARS-CoV-2 spike S2 antibody (40590-T62) were purchased from Sino Biological. Mouse anti-EEA1 monoclonal antibody (610456) was obtained from BD Biosciences. Rabbit anti-LY6E polyclonal antibody (ab300399) was purchased from Abcam. Mouse monoclonal antibody against SARS-CoV-2 spike (GTX632604) was obtained from GeneTex. Donkey anti-goat IgG horseradish-peroxidase-conjugated secondary antibody (PA1-28664), donkey anti-mouse IgG Alexa Fluoro-488 (A21202), donkey anti-rabbit IgG Alexa Fluoro-488 (A21206), donkey anti-mouse IgG Alexa Fluoro-568 (A10037), donkey anti-rabbit IgG Alexa Fluoro-568 (A10042), donkey anti-mouse IgG Alexa Fluoro-647 (A32787) and donkey anti-rabbit IgG Alexa Fluoro-647 (A31573) were purchased from Thermo Fisher Scientific. Rabbit anti-SARS-CoV-2 spike (NR-53788) monoclonal antibody, mouse anti-SARS-CoV-2 nucleocapsid monoclonal antibody (NR-53792) and rabbit anti-SARS-CoV-2 nucleocapsid monoclonal antibody (NR-53791) were obtained through BEI Resources, NIAID, NIH. Goat anti-mouse Fab AF647 (115-607-003) was obtained from Jackson ImmunoResearch. Goat anti-rabbit IgG CF660C (20813) was purchased from Biotium.

Dulbecco's phosphate-buffered saline (14190-144) and LB Miller Broth (BP1426-2) were purchased from Fisher Scientific. FBS (10438-026), DMEM (11965-092), Trypsin-EDTA (0.25%) with phenol red (25200072), pen-strep (15140122), Hanks' balanced salt solution (HBSS) without phenol red (14025092), DMEM powder with high glucose (12100046), LysoSensor Green DND-189 (L7535), PicoPure DNA Extraction Kit (KIT01013), Lipofectamine 2000 Transfection Reagent (11668019), puromycin dihydrochloride (A1113803), blasticidin S HCl (A1113903), hygromycin B (10687010), Hoechst 33342 (H3570), avidin beads (53150), carbenicillin disodium salt (10177012), Annexin V–AF647 (A23204), dithiothreitol (DTT, R0861) and DiIC18 (D7757) were obtained from Thermo Fisher Scientific. EMEM (ATCC 30-2003) was obtained from ATCC. Paraformaldehyde (sc-253236) was obtained from Santa Cruz Biotechnology. Recombinant human IFNα2a (Cyt-204), IFNβ1a (Cyt-236) and IFNλ1 (Cyt-117) proteins were obtained from Prospec Bio. Recombinant human IFNγ (285-IF-100/CF), TNF (210-TA-005/CF) and IL1β (201-LB-010) were purchased from R&D Systems. Luciferase assay reagent (E4550) and Nano-Glo assay reagent (N2011) were purchased from Promega. Camostat mesylate (SML0057), E-64d (E8640), brefeldin A (B652), hydroxychloroquine sulfate (H0915), cellulose (435244), poly-L-lysine hydrobromide (P9155) and polybrene (TR-1003) were obtained from Sigma. In-Fusion snap assembly master mix (638948) and Stellar Competent Cells (636766) for cloning were purchased from Takara Bio. Four-well chambered cover glass (C4-1.5H-N) was obtained from Cellvis. Tween-20 (AB02038-00500) and dimethyl sulfoxide (AB03091-00100) were obtained from American Bio. Sulfo-NHS-SS-Biotin (A8005) and ionomycin calcium salt (B5165) were purchased from APExBio. Paraformaldehyde (PFA; 4%, SC-281692) was obtained from ChemCruz. Glutaraldehyde (50%, 16320) was obtained from Electron Microscopy Science. Alexa Fluor 488 ChromPure Human Transferrin (009-540-050) was obtained from Jackson ImmunoResearch. cOmplete Protease Inhibitor Cocktail (11697498001) was obtained from Roche. High-Fidelity 2X PCR Master Mix (M0541L) was obtained from NEB.

### Plasmid constructs

The following constructs were obtained from Addgene: pLenti-hACE2-hygro (161758), HIV-1Gag-mCherry (85390), pLV-EF1α-IRES-Hygro (85134), pLV-EF1a-IRES-Blast (85133), pMSCV-Blasticidin (75085), Lact-C2-GFP (22852), the spike protein expression plasmid for bat CoV-WIV1 (pTwist-WIV1-CoV Δ18) (164439) and the HCV glycoprotein expression plasmid (pD603 H77 E1E2) (86983). The following reagents were obtained through BEI Resources, NIAID, NIH: SARS-Related Coronavirus 2, Wuhan-Hu-1 spike D614G-Pseudotyped Lentiviral Kit (NR-53817) including pLenti-Luc2/ZsGreen, pHDM-gag/pol, pRC-rev1b, pHDM-tat1b and pHDM-spike-D614G. Plasmids encoding the spike proteins for SARS-CoV-2 (USA-WA1/2020), SARS-CoV, MERS-CoV and HcoV-NL63 were provided by C.B.W. Spike protein expression plasmids for SARS-CoV-2 Delta (B.1.617.2) and Omicron (B.1.1.529) variants were provided by S. Chen. pMX-PH-Halo-LgBiT was provided by M. Yamamoto and Z. Matsuda. CypA-HiBiT was provided by W.M. Expression plasmids of the glycoproteins for HcoV-229E (VG40605-UT), HcoV-OC43 (VG40607-UT), HcoV-HKU1 (VG40021-UT) and EboV (VG40304-CF) were purchased from Sino Biological. pLV-EF1α-Flag-IRES-Hygro was modified from pLV-EF1α-IRES-Hygro by inserting a Flag tag between the promoter region and the multiple cloning site.

Human ACE2 was subcloned into a pLV-EF1α-IRES-Hygro vector by using pLenti-hACE2-hygro (Addgene 161758) as a template. Full-length cDNAs encoding human PLSCR1, TMEM41B, LY6E and IFITM3 were obtained by PCR using cDNA from Huh7.5 or A549 cells. Full-length cDNA encoding mouse PLSCR1 was obtained by PCR from the cDNA of mouse liver. Full-length cDNA encoding R. sinicus PLSCR1 (NCBI reference sequence: XM_019748913.1) was directly synthesized from Azenta–GENEWIZ. Full-length cDNA encoding human CD74 p41 (NCBI

reference sequence: NM_001025159) and NCOA7 isoform 4 (NCBI reference sequence: NM_001199622.1) were obtained from Origin.

pLV-Hg-PLSCR1, pLV-Hg-LY6E and pLV-Hg-TMEM41B were generated by cloning PCR fragments encoding the corresponding gene into a pLV-EF1α-IRES-Hygro vector by infusion cloning. pMSCV-PLSCR1 was generated by cloning the PCR fragment encoding PLSCR1 into pMSCV-Blasticidin. pLV-Flag-PLSCR1, pLV-Flag-mPlscr1, pLV-Flag-batPlscr1, pLV-Flag-TMEM41B, pLV-Flag-IFITM3, pLV-CD74-p41 and pLV-Flag-NCOA7 isoform 4 were generated by inserting PCR fragments encoding the corresponding gene into pLV-EF1α-Flag-IRES-Hygro by infusion cloning. pLV-GFP-PLSCR1 was generated by cloning the PCR fragments encoding PLSCR1 and EGFP into a pLV-EF1α-IRES-Hygro vector by infusion cloning. pLV-Lact-C2-GFP was generated by amplifying a Lact-C2-GFP fragment from the template plasmid purchased from Addgene and cloning it into pLV-EF1α-IRES-Hygro. pLV-PH-Halo-LgBiT was generated by amplifying a PH-Halo-LgBiT fragment from pMX-PH-Halo-LgBiT and then cloning it into pLV-EF1α-IRES-Hygro. pLV-Hg-PLSCR1-KKHA ($K^{258}K^{261}H^{262}A$); pLV-Hg-PLSCR1-H262Y, -H262Q, -H262A, -H262F, -H262W, -H262D, -H262E, -H262K, -H262R, -H262L and -H262V; pLV-Hg-PLSCR1-F281A; pLV-Hg-PLSCR1(5CA) ($C^{184}C^{185}$ $C^{186}PC^{188}C^{189}$ to AAAPAA); PLV-PLSCR1-I105A, -I108A, -I279A, -L283A, -L285A, -M293A, -D242A, -D244A, -S260A, -W263A, -T264A and -N276A; pLV-Hg-mPlscr1-Q271Y; pLV-Hg-batPlscr1-Q286Y; MSCV-PLSCR1(5CA); MSCV-PLSCR1-F281A; and MSCV-PLSCR1-H262Y were generated by PCR-based site-directed mutagenesis. PLV-Hg-Flag-PLSCR1-86-CT, pLV-Hg-Flag-PLSCR1-Δ86-118 and pLV-Hg-Flag-PLSCR1-1-290 were generated by subcloning PCR fragments encoding the corresponding PLSCR1 truncations into pLV-EF1α-Flag-IRES-Hygro.

## Virus strains

The following viruses were used in our study: SARS-CoV-2 USA-WA1/2020 (BEI Resources NR-52281), SARS-CoV-2-mNG (a gift from C.B.W.), SARS-CoV-2 Delta variant (B.1.617.2, a gift from C.B.W.), SARS-CoV-2 Omicron variant (B.1.1.529, a gift from C.B.W.), MHV-A59-GFP (BEI Resources NR-53716), Dengue (DENV-I, BEI Resources NR-82) and HSV-1 VP26-GFP (a gift from A. Iwasaki).

## Genome-wide CRISPR–Cas9 knockout screen

The genome-wide CRISPR–Cas9 knockout screen was modified from a previous report[11]. The LentiCRISPR-V2 pooled library (GeCKO v2) was amplified as described previously[52]. A total of $5 \times 10^7$ Huh7.5 or A549-ACE2 cells were transduced with lentiviruses carrying the GeCKO v2 library (MOI = 0.3) followed by puromycin selection (2 μg ml⁻¹) for 5 days. Surviving cells were split into two groups (+ or −IFNγ) and seeded into 20 T-175 flasks at a density of $5 \times 10^6$ per flask. After 24 h, IFNγ (R&D Systems) was added for an additional 20 h (10 U ml⁻¹ for Huh7.5 and 70 U ml⁻¹ for A549-ACE2). Cells were subsequently infected with icSARS-CoV-2-mNeonGreen (mNG) at MOI = 1 (for Huh7.5) or MOI = 0.3 (for A549-ACE2). At 24 hpi (A549-ACE2) or 48 hpi (Huh7.5), cells were trypsinized and fixed in 4% PFA for 30 min and analysed on a FACSAria (BD). Cells were sorted into two groups: mNG^high or mNG^low on the basis of the intensity of mNG. Cellular DNA was extracted using the Pico-Pure DNA Extraction Kit according to the manufacturer's instruction. sgRNA sequences were amplified using High-Fidelity PCR master mix (NEB) and amplicons were purified from 2.5% agarose gel. Amplicons were sequenced using an Illumina HiSeq2500 (40 million reads per sample). The enrichment of genes in mNG^high versus mNG^low was ranked by the MAGeCK algorithm and the MAGeCK P value of each gene in both the resting and the IFNγ-activated condition was calculated for comparison.

## FACS

Cells were fixed and collected in FACS buffer (1× PBS, 1% FBS, 5 mM EDTA) and filtered through a 40-μm cell strainer before FACS sorting using a BD FACSAria. For sorting of Huh7.5 and A549-ACE2 cells, gates were drawn to separate the cells into mNG^low and mNG^high populations in both IFNγ-untreated and IFNγ-treated conditions. GFP was excited by a 488-nm laser and detected with a 550-nm filter. Cells were sorted on the basis of their mNG intensity and collected into separate tubes for later processing. Data were analysed with FlowJo (BD Biosciences).

## RNA-seq

Huh7.5 cells were treated with 100 U ml⁻¹ IFNγ for 24 h. Total RNA was isolated using the Rneasy Plus Mini Kit (Qiagen). mRNA libraries for sequencing were prepared according to the standard Illumina protocol. Sequencing (100 bp, paired-end) was performed using the Illumina NovaSeq sequencing system at the Genomics Core of Yale Stem Cell Center. The RNA-seq reads were mapped to the human genome (hg38) with STAR in local mode using default settings. The uniquely mapped reads (cut-off: mapping quality score (MAPQ) > 10) were counted to ENCODE gene annotation (v.24) using FeatureCounts. Differential gene expression was analysed with the R package DESeq2. Transcripts with a $\log_2$-transformed fold change > 1 and adjusted P < 0.05 were considered as differentially expressed.

## Analysis of transcription-factor binding profiles

The promoter region of the human *PLSCR1* gene was analysed using the JASPAR website[53] (https://jaspar.genereg.net/). The sequence of the 2-kb region upstream from the transcription initiation site of *PLSCR1* was downloaded from NCBI and scanned by JASPAR using the ISRE and GAS profile. A relative score 0.85 was set as the threshold.

## Virus infection

For fluorescent reporter assays using SARS-CoV-2-mNG, P3 stocks were used for infection. Cells were seeded at 40% confluency in 96-well plates 2 days before infection. The following day, cells were either left untreated or primed with IFNγ (Huh7.5: 8 U ml⁻¹; A549-ACE2: 70 U ml⁻¹) 18 h before infection. IFNγ was kept in the medium during infection. Huh7.5 cells and A549-ACE2 cells were infected at MOI = 1 and MOI = 0.2, respectively. Subsequently, at 1 dpi (A549-ACE2) and 2 dpi (Huh7.5), the medium was removed from the wells and cells were washed once with PBS before being fixed with 4% PFA for 30 min and stained with Hoechst 33342 for an additional 20 min. Then, high-content imaging (Cytation 5, BioTek) of the cells was performed to measure mNG expression. The average intensity of mNG per cell as well as the percentage of infected cells were quantified and analysed by Gen5 software. Infection of additional cell lines was performed under the following conditions: Calu-3 (MOI = 1, 24 hpi), HeLa-ACE2 (MOI = 0.2, 24 hpi), Tonsil-ACE2 (MOI = 1, 24 hpi), HaCaT-ACE2 (MOI = 1, 24 hpi) and LET1-ACE2 (MOI = 0.1, 24 hpi).

For viral RNA experiments with SARS-CoV-2 USA-WA1/2020, the Delta variant (B.1.617.2) and the Omicron variant (BA.1), Huh7.5 cells were seeded at 40% confluency in 12-well plates 2 days before infection. On the day of infection, Huh7.5 cells were infected at MOI = 1 (2 dpi), MOI = 0.5 (1 dpi) and MOI = 0.5 (1 dpi), respectively. All infection assays above were performed in a Biosafety Level 3 (BSL-3) facility.

For infection with additional viruses, the experimental conditions are as follows: HeLa cells were infected with HSV-1 VP26-GFP at MOI = 0.2 for 48 h. LET1 cells were infected with MHV-A59-GFP at MOI = 0.1 for 48 h. Huh7.5 cells were infected with DENV-I at MOI = 0.1 for 24 h. Subsequently, at 1 dpi (A549-ACE2) and 2 dpi (Huh7.5), the medium was removed from the wells and cells were washed once with PBS before being fixed with 4% PFA for 30 min and stained with Hoechst 33342 for an additional 20 min. Then, high-content imaging (ImageXpress Pico, Molecular Devices) of the cells was performed to measure GFP expression. The average intensity of GFP per cell as well as the percentage of infected cells were quantified and analysed by the CellReporterXpress software.

For confocal imaging, A549-ACE2 cells were spinfected with SARS-CoV-2 USA-WA1/2020 at 1,000g, 37 °C for 30 min to synchronize the infection, followed by washing twice with pre-chilled PBS.

Pre-warmed DMEM was added to cells to initiate the virus entry. The cells were incubated at 37 °C for various time periods. The MOIs used for experiments are indicated in the figure legends.

## SARS-CoV-2 plaque assay

Vero E6 cells were seeded at $9 \times 10^5$ cells per well in 6-well plates for infection with SARS-CoV-2 USA-WA1/2020 the following day. First, the medium was removed, and each well was washed once with PBS. Then, 200 µl of 10-fold serial dilutions of virus was added to the corresponding wells, and cells were incubated at 37 °C for 1 h with gentle rocking every 10 min. Afterwards, 2 ml of overlay medium (DMEM, 2% FBS, 0.6% methylcellulose) was added to each well. At 2 dpi, the medium was removed, and cells were washed once with PBS. Then, cells were fixed with 4% PFA for 30 min before staining with 0.5% crystal violet solution for 15 min. Finally, cells were washed three times with PBS and then dried before counting the number of plaque-forming units (PFU).

## Measurement of viral RNA by qPCR with reverse transcription

Cells grown in 12-well plates were washed twice before total RNA was extracted using TRIzol reagent (Thermo Fisher Scientific; 13778030) and subsequently purified with the Direct-zol RNA Miniprep kit (Zymo Research; R2050). Then, the RNA was reverse-transcribed using PrimeScript RT Master Mix (Takara Bio; RR036B). The cDNA was diluted 1:5 before qPCR with reverse transcription (qRT–PCR) was performed using PowerUp SYBR Green Master Mix (Thermo Fisher Scientific; A25776). SARS-CoV-2 (US-WA1/2020, Delta variant and Omicron variant) replication was quantified by using primers specific to nucleocapsid (N) mRNA (forward 5′-GGGGAACTTCTCCTGCTAGAAT-3′; reverse 5′-CAGACATTTTGCTCTCAAGCTG-3′). DENV-I replication was quantified by using primers specific to non-structural protein 1 (NS1) mRNA (forward 5′-GCATATTGACGCTGGGAGAGAC-3′; reverse 5′-TTCTGTGCCTGGAATGATGCTG-3′). All viral mRNA levels were normalized to β-actin (forward 5′-CACCATTGGCAATGAGCGGTTC-3′; reverse 5′-AGGTCTTTGCGGATGTCCACGT-3′). Reactions were performed on the QuantStudio Real-Time PCR system (Thermo Fisher Scientific, Applied Biosystems). For relative quantification of mRNA levels, the cycle threshold (Ct) values were compared using the ΔΔCt method.

## Western blot

Cell lysates were prepared in 1.2× SDS–PAGE sample loading buffer. The cell lysates were fractionated on SDS–PAGE (12% gel) and transferred onto polyvinylidene fluoride (PVDF) membrane (Millipore; IPVH00010). Membranes were blocked with 5% milk in 1× TBST (1× Tris-buffered saline, 0.1% Tween-20) and then incubated with primary antibody at 4 °C overnight in 5% BSA. Subsequently, membranes were washed three times with 1× TBST and then incubated with horseradish-peroxidase-conjugated secondary antibodies. The membranes were exposed using Clarity Normal/Max Western ECL substrate (BioRad; 1705062), and the readout was detected using the BioRad ChemiDoc MP system.

## Production of pseudovirus particles

HIV-1-based PsV was produced in HEK293T cells plated on a 10-cm plate. Cells were transfected with lentiviral backbone (9 µg pLenti-Luc2/ZsGreen) and helper plasmids (2 µg pHDM-gag/pol, 2 µg pRC-rev1b and 2 µg pHDM-tat1b), along with an expression plasmid encoding the glycoprotein gene of the virus of interest (3 µg). After incubation for 4 h at 37 °C, the medium was replaced with fresh medium (DMEM, 10% FBS, supplemented with pen-strep). PsV particles were collected 48 h after transfection, clarified by centrifugation ($1,000g \times 5$ min), filtered through a 0.45-µm filter and then aliquoted for storage at −80 °C.

mCherry-labelled HIV-based pseudoviral particles were prepared by transfecting 293T cells plated on a 10-cm dish with 9 µg pLV-EF1a-IRES-Blast, 4.5 µg psPAX2, 1.5 µg HIV-gag-mCherry and an expression plasmid encoding spike protein from SARS-CoV-2 Omicron variant (3 µg) by Lipofectamine 2000. At 48 h after transfection, the supernatant was filtered through a 0.45-µm filter, laid onto a 20% sucrose (w/v in 1× HBSS) cushion and centrifuged using a Sorvall TH-641 rotor at 100,000g for 2 h. The supernatant was discarded and the pseudoviral particles concentrated in the pellet were resuspended with 500 µl of DMEM cell culture medium.

## Pseudovirus entry infection assay

A total of $1.2 \times 10^4$ Huh7.5 cells were seeded in each well of a clear bottom 96-well plate. Two days later, 100 µl of PsV was added to each well, and the plate was centrifuged at 1,000g for 30 min at room temperature. Infected cells were incubated for two days before being lysed with 1× passive lysis buffer (Promega; E1941) for 20 min at 4 °C. Then, 80 µl luciferase assay reagent (Promega; E4550) was added to the lysate and relative luminescence was measured using a microplate reader (BioTek).

## Protease inhibitor assay

Huh7.5 and A549-ACE2 cells were seeded at $2.5 \times 10^4$ cells per well in 96-well plates. The next day, cells were primed with protease inhibitors at the following concentrations or dilutions: mock DMSO (1:200), E-64d (25 µM), camostat (20 µM), brefeldin A (5 µg ml$^{-1}$) and HCQ (10 µM). Three hours later, Huh7.5 and A549-ACE2 cells were infected with SARS-CoV-2-mNG at MOI = 1 and MOI = 0.2, respectively. Inhibitors were kept in the medium during infection. At 1 dpi (A549-ACE2) and 2 dpi (Huh7.5), cells were washed with PBS and then fixed with 4% PFA. Nuclei were stained with Hoechst and mNG expression was measured by high-content imaging.

For inhibitor assays in Calu-3, cells were seeded at $2.75 \times 10^5$ cells per well in 12-well plates. When they reached 90–95% confluency, cells were primed with protease inhibitors at the following concentrations or dilutions: mock DMSO (1:800), E-64d (25 µM) and camostat (25 µM). Three hours later, cells were infected with SARS-CoV-2 USA-WA1/2020 at MOI = 1. Inhibitors were kept in the medium during infection, and cells were incubated at 37 °C for 1 h with gentle rocking every 10 min. Afterwards, the infection medium was removed, and cells were washed twice with PBS. Then, fresh medium with inhibitors was added to each well. At 1 dpi., cells were washed twice before total RNA was extracted using TRIzol reagent and then purified using Direct-zol RNA Miniprep kit. Subsequently, viral RNA was measured by qRT–PCR using primers specific to SARS-CoV-2 nucleocapsid mRNA.

## Biotinylated ACE2 pulldown assay

A549-ACE2 cells were seeded at $2 \times 10^6$ cells per well in 6-well plates. Two days later, cell-surface proteins were labelled with biotin. First, cells were washed twice with DBPS$^+$ solution (DPBS supplemented with 0.9 mM CaCl$_2$ and 0.49 mM MgCl$_2$, pH 7.4). Next, cells were incubated with 2.5 mg ml$^{-1}$ biotin (EZ-LinkTM Sulfo-NHS-LC-LC-Biotin, Thermo Fisher Scientific) in DBPS$^+$ solution at 4 °C for 30 min. Afterwards, cells were washed three times in 100 mM glycine for 5 min, followed by two additional wash cycles in 20 mM glycine for 5 min. Subsequently, cells were lysed with lysis buffer (1% Triton X-100, 50 mM Tris/HCl pH 7.4, 150 mM NaCl, 1 mM EDTA and Complete Protease Inhibitor Cocktail), and a portion of the whole-cell extract was aliquoted for input. The remaining extract was incubated with avidin beads on a rocker at 4 °C overnight. Finally, mixtures of the whole-cell extract and beads (biotin pulldown) were washed six times with lysis buffer and then boiled for immunoblotting.

## Immunostaining

For immunostaining, cells cultured on coverslips were washed twice with PBS, fixed with 4% PFA for 30 min and permeabilized with 0.2% Triton X-100 for 3 min. Cells were blocked with 1% BSA in PBS for 1 h and incubated with primary antibodies (1:50–1:200 diluted) in PBS

supplemented with 1% BSA overnight at 4 °C, followed by incubation with secondary antibodies (1:500 diluted) for 1 h at room temperature. Nuclei were stained with Hoechst 33342 (1:4,000 diluted) for 5 min.

## Fluorescent imaging

All confocal images were acquired using a Leica SP8 laser scanning confocal microscope with 405-nm and 488-nm lasers and a pulsed supercontinuum white light source (470 nm–670 nm). For analysing the subcellular localization of proteins, images (1,024 × 1,024) were taken under a HC PL APO 100× oil immersion objective (N.A. 1.44) with 4 frames average. For determining the number of dsRNA foci or the distribution of SARS-CoV-2 nucleocapsid, images (1,024 × 1,024) were taken under a HC PL APO CS2 63× oil immersion objective (N.A. 1.40) with 4 frames average. For determining the number of spike and nucleocapsid double-positive particles, Z-stack images (512 × 512) were acquired under a HC PL APO CS2 63× oil immersion objective (N.A. 1.40) with 0.5 µm per optical section.

For time-lapse microscopy of the formation of PLSCR1-wrapped vesicles, A549-ACE2 cells were seeded on 4-well chambered cover glass (1.5) a day before infection. Cells were spinfected with mCherry-labelled pseudoviruses containing SARS-CoV-2 spike at 1,000g for 30 min at 4 °C. Images (512 × 512) were captured on a DeltaVision OMX SR microscopy system under a 63× oil immersion objective (N.A. 1.40) with 1-min intervals for around 2 h at 37 °C, 5% $CO_2$ and 80% humidity.

## 4Pi-SMS nanoscopy

Two-colour 4Pi-SMS[41] was performed on a custom-built microscope with two opposing objectives in 4Pi configuration[54]. A549 cells were seeded on 30-mm-diameter no. 1.5H round coverslips (Thorlabs) and grown for 1 day before infection. Infected cells were fixed with 4% PFA for 30 min and permeabilized with 0.2% Triton X-100 for 3 min. Cells were blocked with 10% goat serum in PBS for 1 h and subsequently incubated with mouse anti-PLSCR1 and rabbit anti-spike antibodies overnight at 4 °C. Primary antibodies were detected by goat anti-mouse Fab AF647 (Jackson ImmunoResearch) and goat anti-rabbit IgG CF660C (Biotium) at a 1:200 dilution (2 h at room temperature). Samples were post-fixed in 3% PFA + 0.1% glutaraldehyde for 10 min and stored in PBS at 4 °C. Sample mounting, image acquisition and data processing were mostly performed as previously described[54] except that imaging speed was 200 Hz with a 642-nm laser intensity of around 12.5 kW cm$^{-2}$. Typically, 3,000 × 100-200 frames were recorded. DME were used for drift correction. All 4Pi-SMS images and videos were rendered using Point Splatting mode (20-nm particle size) with Vutara SRX 7.0.06 software (Bruker).

## Image processing and analysis

Images were processed in LAS X (Leica), SoftWoRx (v.7.0), Fiji or Imaris 9.8 (Oxford Instruments) software. Deconvolution of fluorescent images was performed in LAS X using the default settings. Fluorescence colocalization analysis was performed using Imaris software. The percentage of PLSCR1 fluorescent signal colocalized with LAMP1 or CD63 as well as the Mander's overlap coefficient were calculated by setting proper thresholds for both channels to avoid background signal. The PLSCR1-positive foci were automatically detected by surface reconstruction using Imaris. The average number or fluorescent intensity of PLSCR1-positive foci per cell was calculated by dividing the total number or fluorescent intensity of PLSCR1-positive foci by the total number of cells in each randomly selected image, respectively. Cells with large dsRNA foci were identified manually using Fiji and the percentage of cells with large dsRNA foci was calculated by dividing the number of cells with dsRNA foci by the total number of cells in each image. Cells with dispersed or endosomal nucleocapsid signal were manually identified using Fiji. The percentage of cells with the indicated nucleocapsid distribution was normalized by the total cell count in each image. Spike and nucleocapsid double-positive particles

were automatically identified using Imaris and the average number of double-positive particles per cell was normalized by the total cell number in each image. For all the image analysis, 10–13 images with 120–250 cells in total (as indicated in the figure legends) were randomly captured. The percentage of SARS-CoV-2-containing vesicles coated with PLSCR1 in each cell was quantified manually using Fiji; around 25 cells were counted in each condition.

## Virus binding and internalization assay

For the virus binding assay, A549-ACE2 cells were pre-chilled to 4 °C for 15 min followed by incubation with SARS-CoV-2 (MOI = 20) at 4 °C for 1 h. Unbound viral particles were removed by washing with pre-chilled PBS three times. The relative amount of bound virus normalized to β-actin was quantified by qPCR.

For the virus internalization assay, cells were incubated with SARS-CoV-2 using the same condition described above. Cells were then transferred to 37 °C for 30 min to allow internalization of bound virus. Uninternalized viral particles were removed by treating cells with 0.25% trypsin for 15 min at 4 °C. As a negative control, another set of cells was directly treated with trypsin after virus binding to digest all the bound but uninternalized viruses. The relative amount of internalized virus normalized to β-actin was quantified by qPCR.

## Virus–cell membrane fusion assay

The virus–cell fusion assay was performed according to the methods described previously[40,55] with modification. HIV-based pseudoviral particles containing CypA-HiBiT were prepared by transfecting 293T cells plated on a 10-cm dish with 9 µg pLenti-Luc2/ZsGreen, 4.5 µg psPAX2, 2 µg CypA-HiBiT and 3 µg pVP40-spike (encoding the spike protein from SARS-CoV-2 Delta) by Lipofectamine 2000. At 48 h after transfection, the supernatant was filtered through a 0.45-µm filter, laid onto a 20% sucrose (w/v in 1× HBSS) cushion and centrifuged using a Sorvall TH-641 rotor at 100,000g for 2 h. The supernatant was discarded and the pseudoviral particles concentrated in the pellet were resuspended with 500 µl of DMEM + 10% FBS medium.

293T-ACE2 cells were transfected with pMX-PH-Halo-LgBiT a day before infection. Huh7.5 cells were stably transduced with pLV-PH-Halo-LgBiT, followed by hygromycin (350 µg ml$^{-1}$) selection for 7 days.

Target cells expressing the LgBiT fragment were plated in a white opaque 96-well dish. One day after plating, each well of cells was spinfected with 100 µl of pseudoviruses containing CypA-HiBiT at 1,000g and 4 °C for 30 min, followed by incubation at 37 °C for 1 h (293T-ACE2) or 2 h (Huh7.5). The medium was removed and Nano-Glo assay reagent was added to the target cells. The activity of complemented Nano-Luc was measured by Spectramax i3x microplate reader (Molecular Devices).

## Cell–cell membrane fusion assay

The virus–cell fusion assay was performed according to the methods described previously[25] with modification. Huh7.5 cells (acceptor cells) of indicated genotypes were stably transduced with the construct encoding the complementary fragment of the split-NanoLuc (PH-Halo-LgBiT) as described above. 293T cells (donor cells) were transfected with plasmids encoding the spike protein from SARS-CoV-2 Omicron together with a split-NanoLuc construct encoding the HiBiT fragment. Twenty-four hours after transfection, the donor and acceptor cells (at ratio 1:1) were mixed and co-cultured in a 96-well plate at 37 °C for 18 h before assay. The medium was removed and Nano-Glo assay reagent was added to the target cells. The medium was removed and Nano-Glo assay reagent was added to the cells. The activity of complemented NanoLuc was measured by Spectramax i3x microplate reader (Molecular Devices).

For visualization of the syncytia formed after cell–cell fusion, 293T cells were transduced with plasmids encoding SARS-CoV-2 spike protein

and EGFP. Twenty-four hours after transfection, Huh7.5 cells of the indicated genotypes were co-cultured with the 293T cells at 37 °C for 18 h. The cells were washed with PBS, fixed by 4% PFA and stained with Hoechst. Images were obtained with high-content imaging (ImageXpress Pico, Molecular Devices) as described above.

## Spike protein cleavage assay

The spike cleavage assay was modified according to a previous report[56]. A549-ACE2 cells seeded on a 24-well plate were spinfected with pseudovirus particles containing SARS-CoV-2 spike at 1,000$g$ for 30 min at 4 °C. Cells were then washed twice with DPBS, incubated with pre-warmed culture medium and shifted to 37 °C. Cells were collected at different time points followed by western blot analysis. Both the intact S2 domain and the processed S2′ fragment was detected by an antibody specifically recognizing the S2 domain. Cells treated with Cathepsin inhibitor E-64d were used as a negative control.

## PS externalization assay

PS externalization was detected using Annexin V conjugated with Alexa647 (A23204, Thermo Fisher Scientific) according to the manufacturer's instruction. In brief, A549-ACE2 cells were digested by trypsin, spun down at 200$g$ and washed twice with PBS. Cell pellets were resuspended in 100 μl of 1× binding buffer (prepared from 5× stock solution provided by the manufacturer) at a density of $5 \times 10^6$ cells per ml, and treated with DMSO or 10 μM ionomycin for 10 min. Cells were subsequently incubated with 5 μl Annexin V–AF647 for 20 min at room temperature followed by the addition of 400 μl 1× binding buffer. Cells were then analysed using a Beckman CytoFLEX S flow cytometer (APC filter).

## MDS analysis

A coarse-grained simulation using the AlphaFold[57] structure of PLSCR1 with the N-terminal region truncated was assembled using the insane.py Python script[58], memembed (ref. 59) and martinize2 (ref. 60) in an asymmetric membrane comprised of phosphatidylcholine (PC):phosphatidylethanolamine (PE) (8:2) in the lower leaflet and PC:PE:PS:PtdIns(4,5)$P_2$ (2:5:2:1) in the upper leaflet. A 5-μs simulation was performed to equilibrate the system. This was then back-mapped to atomistic resolution using the cg2at tool[61]. The mutation H262Y was made at this point using PyMOL, followed by an energy minimization step using the steepest descents algorithm. Separately, the atomistic system with the palmitoyl tails was assembled using CHARMM-GUI (refs. 62,63) in the same membrane as previously described. The protein had palmitoyl chains added at residues C184, C185, C186, C188 and C189.

Three repeats were performed for each system (palmitoylated wild type and H262Y mutant). Each simulation was 50 ns using a time step of 2 fs at 310 K, using the charmm36m (ref. 64) forcefield and TIP3P water. The water bond angles and distances were constrained by SETTLE (ref. 65). Hydrogen covalent bonds were constrained using the LINCS algorithm[66]. The velocity rescale[67] and Parrinello-Rahman[68] coupling methods were used with the time constants $\tau_p = 1.0$ ps and $\tau_T = 0.1$ ps for pressure and temperature, respectively. The protein, lipids and water and ions were groups separately for temperature coupling. Simulations were run using GROMACS v.2021.3 (ref. 69). A single cut-off of 1.2 nm was used for the van der Waals interaction. The particle mesh Ewald (PME) method was used for electrostatic interactions with a cut-off of 1.2 nm.

The RMSF was calculated using the gmx rmsf tool over all repeats and visualized in PyMOL. The hydrogen bond analysis was performed with VMD.

## Generation of GPMVs

The preparation of GPMVs was performed according to the methods described previously[70,71]. In brief, A549-ACE2-*PLSCR1*-KO cells stably overexpressing GFP-PLSCR1-WT or -5CA were plated on T-25 plates coated with poly-L-lysine. After 24 h, the cells were stained with 1 μg ml$^{-1}$ DiI-C18 for 20 min, washed four times with GPMV buffer (20 mM HEPES pH 7.4, 150 mM NaCl and 2 mM CaCl$_2$) and incubated with 1 ml GPMV buffer containing 1.9 mM DTT and 27.6 mM formaldehyde. Sixteen hours after induction, GPMVs were collected from the supernatant and used freshly for analysis. The quality of GPMVs was checked using a confocal microscope.

## Measurement of membrane bending rigidity

The membrane bending modulus of a GPMVs was estimated by aspirating the GPMV using a micropipette, and pulling a thin membrane tether using a spherical bead trapped by a focused infrared laser beam. The force acting on the tether is $f = 2\pi\sqrt{2\kappa\sigma}$, where $\kappa$ is the membrane bending modulus and $\sigma$ is the membrane tension[72]. The membrane tension is varied by changing the aspiration pressure $\Delta P$, $\sigma = \Delta P R_p/[2(1 - R_p/R_v)]$, where $R_p$ and $R_v$ are the radii of the micropipette and the GPMV, respectively[73]. For every value of $\sigma$, the resulting $f$ is measured using the displacement of the trapped bead from the centre of the optical trap. The bending modulus is estimated from the slope of a line fit to the plot of $f^2$ as a function of $\sigma$. The procedure is repeated for different GPMVs.

Our home-built optical tweezers setup is combined with a spinning-disc confocal system[74] which allowed us to verify that wild-type PLSCR1–GFP was localized to the GPMV membranes whereas the 5CA mutant was distributed diffusely inside the GMPVs. The trap stiffness was calibrated using a hydrodynamic flow method[75] and was 283 pN μm$^{-1}$ for the polystyrene beads (diameter = 3.15 μm, PP-30-10, Spherotech).

For micropipette aspiration, we used a micropipette holder attached to a programmable three-axis piezo stage (100 μm range, P-611.3 NanoCube, with controller E-727 and Mikromove software, Physik Intrumente), mounted on a manual manipulator for coarse movement (Newport M-462, Newport). Both the micropipette and the glass chamber were coated with 1% BSA to minimize GPMV adhesion and facilitate the free flow of the GPMV membrane inside the micropipette. Aspiration pressure was controlled by vertically moving a water reservoir connected to the micropipette. Before aspirating a GPMV, pressure was zeroed by bringing a bead near the tip of the micropipette (3–6 μm) and adjusting the reservoir height until the bead stopped moving. Subsequently, a bead was trapped, and its zero-force position $(x_0, y_0)$ was recorded. Then, the GPMV was brought in contact with the bead briefly (around 1–3 seconds) before being pulled away to form a membrane tether. Bead positions $(x, y)$ were determined from analyses of image stacks. The presence of a tether was confirmed either visually or by releasing the trap and observing the retraction of the bead toward the GPMV. The tether force was calculated from the deviation of the bead's position from its load-free value and the trap stiffness using a custom-written MATLAB program[74]. All the plots and fitting were done using Origin 2023 (OriginLab).

## Statistics and reproducibility

Data were subjected to statistical analysis and plotted using Microsoft Excel 2010 or GraphPad Prism v.9.0. D'Agostino and Pearson omnibus normality test or Kolmogorov–Smirnov test were used to determine the normal distribution of data. For data with a normal distribution, single comparisons were performed using the two-sided Student's $t$-test for groups with equal variances, and Welch's correction was used for groups with unequal variances. Datasets that did not follow a normal distribution were analysed using a nonparametric test (Mann–Whitney test). Multiple comparisons were assessed by one-way ANOVA with Tukey's post-hoc test or two-way ANOVA with Tukey's post-hoc test, whereas Brown–Forsythe and Welch ANOVA with Dunnett's post-hoc test was used for groups with unequal variances. For all analyses, $P < 0.05$ was considered statistically significant. Results were reported as either mean ± s.e.m. or mean ± s.d. as indicated. The methods for calculating

*P* values are indicated in the figure legends. All of the *P* values obtained from statistical analysis are listed in the graphs or in the source data files. All experiments were repeated at least three times with similar results unless otherwise mentioned in the figure legends.

### Reporting summary

Further information on research design is available in the Nature Portfolio Reporting Summary linked to this article.

## Data availability

The data supporting the findings of this study are available within the paper and its Supplementary Information files. Sequences for all the sgRNAs used in this study are provided in Supplementary Table 1. Genome-wide CRISPR screening data are provided in Supplementary Tables 2 and 3. RNA-seq data are provided in Supplementary Tables 4 and 5. The raw RNA-seq data have been deposited and made publicly available in the NCBI Gene Expression Omnibus with accession number GSE233548. Full versions of all blots are provided in Supplementary Fig. 1. The gating strategies of flow cytometry are provided in Supplementary Fig. 2. Gene expression data derived from nasopharyngeal swab samples is publicly available in dbGaP (https://dbgap.ncbi.nlm.nih.gov/aa/wga.cgi?login=&page=login) with accession no. phs002433.v1.p1. The human genome reference (hg38) used in the RNA-seq analysis is available in the NCBI genome assembly with accession number GCF_000001405.39. The protein expression profiles are available in the web-based Human Protein Atlas database (https://www.proteinatlas.org/). The transcriptional-factor binding profiles are available in the web-based JASPAR database (https://jaspar.genereg.net/). The materials, reagents and other experimental data are available from the corresponding author upon request. Source data are provided with this paper.

## Code availability

The source code used for calculating membrane bending rigidity has been deposited in Zenodo (https://doi.org/10.5281/zenodo.7948870). The publicly available code used for MDS is referenced in the Methods.

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

**Acknowledgements** We thank all members of the J.D.M. laboratory for discussions; Yale Environmental Health and Safety's Biological Safety Team for the assistance with BSL-3 training; A. D. Yadavalli for the suggestion on data analysis; and M. Yamamoto for sharing plasmids and materials. This work was supported by Fast Grants (Emergent Ventures, Mercatus Institute, George Mason University) to E.F.F.; an MRC studentship (MR/N014294/1) to C.M.B.; Wellcome (208361/Z/17/Z), MRC (MR/S009213/1) and BBSRC (BB/P01948X/1, BB/R002517/1 and BB/S003339/1) grants to P.J.S.; NIH grant R01AI163395 to W.M.; and NIH grant R01 NS113236 to E.K. J.D.M. is an Investigator of the Howard Hughes Medical Institute. Schematic graphs were created with BioRender (https://biorender.com/) under the academic subscription of Yale School of Medicine—Department of Immunobiology's plan.

**Author contributions** D.X., W.J. and J.D.M. conceived the project, designed experiments and wrote the manuscript. D.X. and W.J. performed most of the experiments with the help of all other authors. L.W. performed the FACS analysis for PS externalization and analysed the RNA-seq data. R.G.G. helped with establishing genome-wide CRISPR–Cas9 screens. E.-S.P. performed the biotin pulldown experiment. M.S. performed the 4Pi-SMS imaging with supervision from J.B. P.K. helped with the analysis of protein structures. S.K.C. and E.K. performed and analysed the membrane bending rigidity experiments. N.R.C. and E.F.F. provided gene expression data from patient nasal swabs. P.D.U., J.R.G. and W.M. provided help and unpublished reagents for membrane fusion assays. C.M.B. and P.J.S. performed the MDS analysis. C.B.W. provided key virus strains and scientific feedback. J.D.M. supervised the entire project and was responsible for finalizing and submitting the manuscript.

**Competing interests** The authors declare no competing interests.

**Additional information**
**Correspondence and requests for materials** should be addressed to John D. MacMicking.

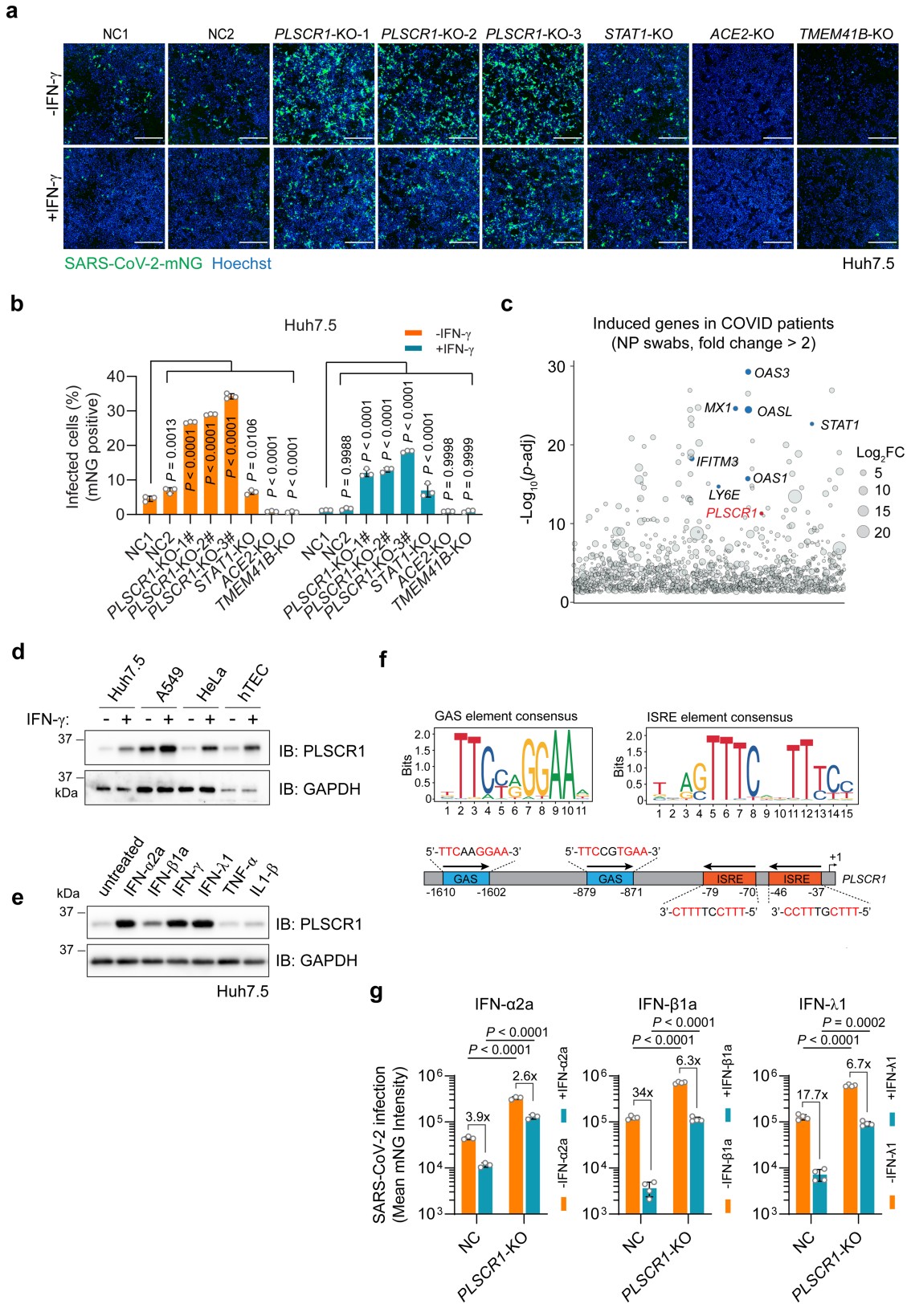

**Extended Data Fig. 1** | See next page for caption.

**Extended Data Fig. 1 | PLSCR1 expression profiles in patients with COVID-19 and in various cell types, together with antiviral responses to different immune stimuli. a**, Representative images of resting or IFNγ-activated Huh7.5 cells infected with SARS-CoV-2-mNG at an MOI of 1 for 48 h. Genotypes of Huh7.5 cells are indicated. Related to Fig. 1f. **b**, Quantification of % infected cells in **a**. ($n = 3$). **c**, Comparison of the expression level of mRNAs extracted from nasopharyngeal swab between patients with COVID-19 ($n = 30$) and control individuals who were negative for SARS-CoV-2 ($n = 8$). Only transcripts with $\log_2$-transformed fold change > 1 and adjusted $P$ value < 0.05 are presented on the plot. Several upregulated ISGs with known antiviral activities were highlighted. **d**, Western blot showing the IFNγ-induced upregulation of PLSCR1 across cell lines. hTEC: human primary tracheal epithelial cells. **e**, Western blot showing the protein expression of PLSCR1 in cells treated with the indicated cytokines for 20 h. Concentrations used: IFNα2a/β1a (500 U ml⁻¹), IFNγ (500 U ml⁻¹), IFNλ1 (1 ng ml⁻¹), TNF (100 ng ml⁻¹), IL1β (25 ng ml⁻¹). **f**, Schema depicting the presence and position of GAS and ISRE elements in the promoter region (2 kb upstream from the transcription initiation site) of human *PLSCR1* gene. **g**, Effect of *PLSCR1* deficiency on IFNα2a ($n = 3$), IFNβ1a ($n = 4$) or IFNλ1 ($n = 4$) mediated restriction of SARS-CoV-2-mNG infection in Huh7.5 cells (MOI = 1, 48 hpi). Concentrations used: IFNα2a (50 U ml⁻¹), IFNβ1a (2,000 U ml⁻¹) and IFNλ1 (1 ng ml⁻¹). Data are mean ± s.d. $P$ values were calculated using one-way ANOVA followed by Tukey's multiple comparison test (**b**) or two-way ANOVA, followed by Tukey's test (**g**). Scale bar in **a**: 500 µm. Experiments in this figure were performed three times.

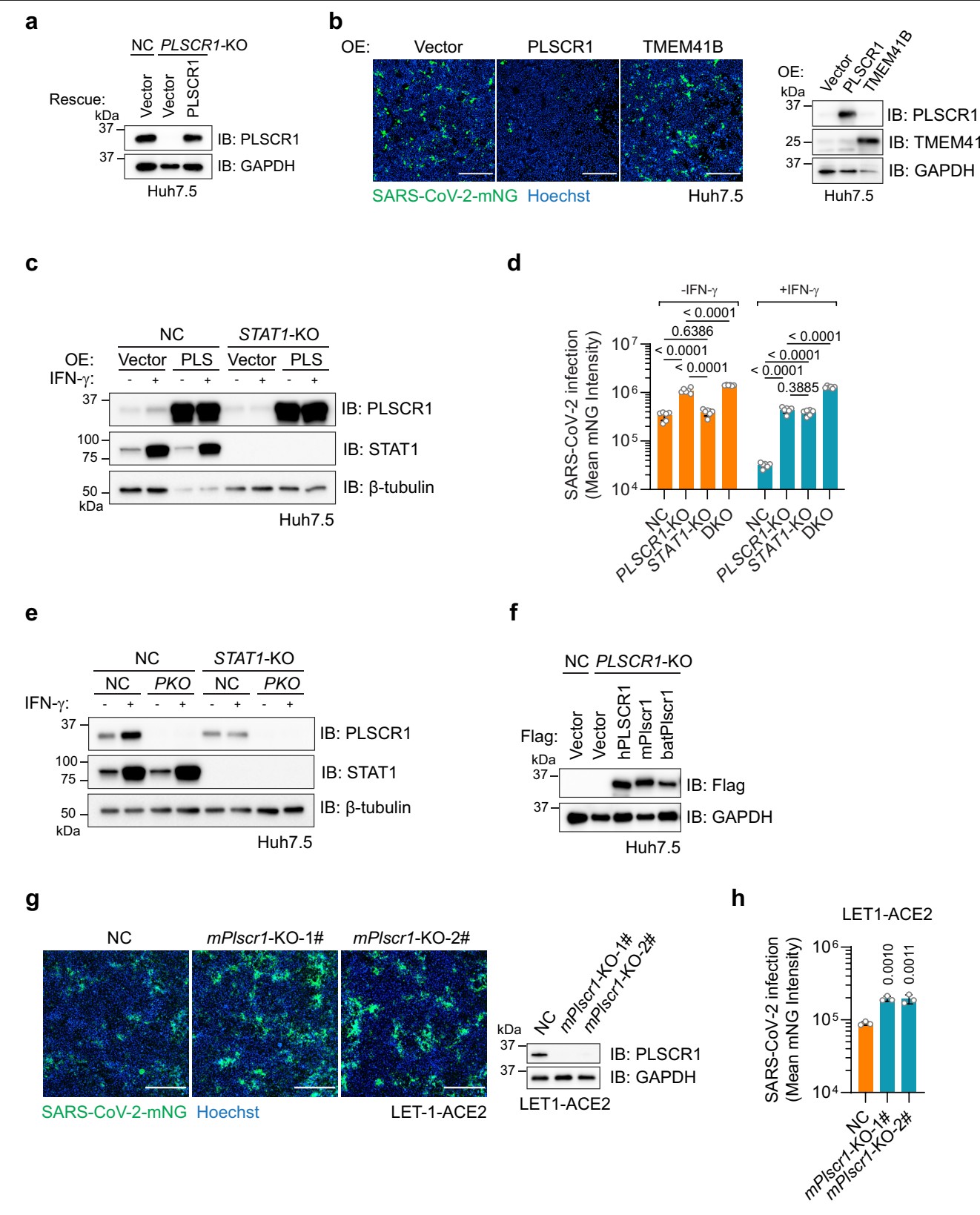

**Extended Data Fig. 2 |** See next page for caption.

**Extended Data Fig. 2 | The anti-SARS-CoV-2 activity of PLSCR1 in basal conditions is independent of STAT1 and conserved across species.**
**a**, Western blot showing the expression level of the indicated proteins in Huh7.5 cells. Related to Fig. 2b. **b**, Left, representative images showing the infectivity of SARS-CoV-2-mNG in Huh7.5 cells overexpressing the indicated proteins. Right, western blot showing the expression level of the indicated proteins. Related to Fig. 2d. **c**, Western blot showing the expression level of the indicated proteins. Related to Fig. 2e. **d**, Quantification of SARS-CoV-2 infection in Huh7.5 cells of the indicated genotypes in the presence or absence of IFNγ (10 U ml⁻¹) (MOI = 1, 48 hpi). DKO: *PLSCR1/STAT1* double-KO. (n = 6) **e**, Western blot

showing the expression level of the indicated proteins in **d**. **f**, The expression level of PLSCR1 orthologues in Huh7.5 cells. hPLSCR1: *H. sapiens*, mPlscr1: *M. musculus*, batPlscr1: *R. sinicus*. Related to Fig. 2g. **g**, Left, representative images showing the infectivity of SARS-CoV-2-mNG in NC or m*Plscr1*-KO LET1-ACE2 cells. Right, western blot showing the expression level of the indicated proteins. **h**, Quantification of SARS-CoV-2-mNG infection in LET1-ACE2 cells in **g** (MOI = 0.1, 24 hpi) (*n* = 3). Data are mean ± s.d. *P* values were calculated using one-way ANOVA followed by Tukey's multiple comparison test in **d**, **h**. Scale bar in **b**: 500 µm. Experiments in this figure were performed three times, except **a** (five times).

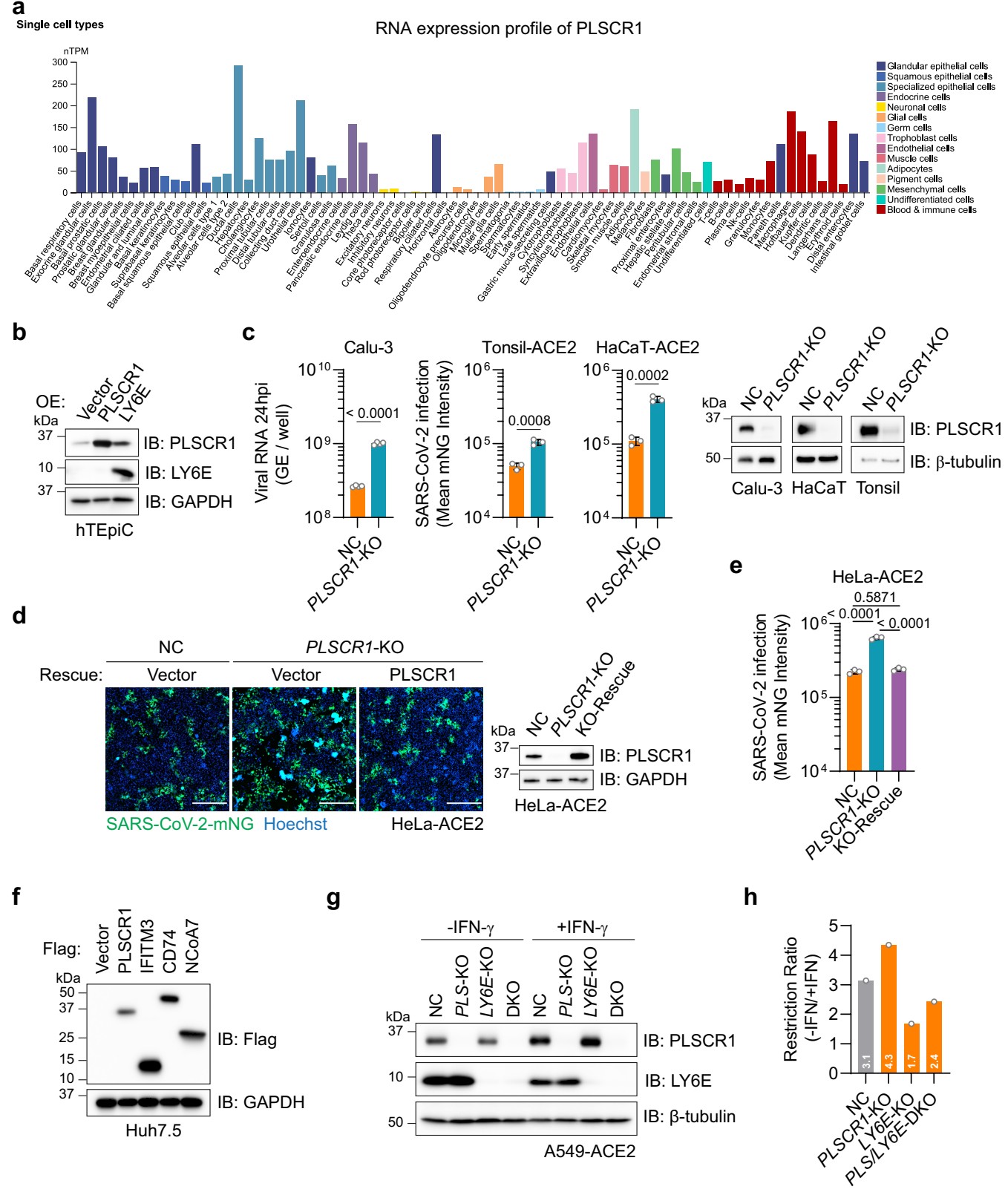

**Extended Data Fig. 3** | See next page for caption.

**Extended Data Fig. 3 | Anti-SARS-CoV-2 activity across different cell types and ISGs. a**, Expression profile of PLSCR1 across cell types. Data were extracted from the Human Protein Atlas database. **b**, Western blot showing the protein expression level in hTEpiCs overexpressing the indicated proteins. Related to Fig. 2h. **c**, Left, quantification of intracellular SARS-CoV-2 RNA in Calu-3 cells (MOI = 1, 24 hpi, *n* = 4) or SARS-CoV-2-mNG infection in HaCaT-ACE2 (MOI = 1, 24 hpi, *n* = 3), and Tonsil-ACE2 (MOI = 1, 24 hpi, *n* = 3) cells. Right, western blot showing the expression level of the indicated proteins in Calu-3, HaCaT-ACE2 or Tonsil-ACE2 cells. **d**, Left, representative images showing the infectivity of SARS-CoV-2-mNG in HeLa-ACE2 of indicated genotypes. Right, western blot showing the expression level of the indicated proteins in HeLa-ACE2. **e**, Quantification of intracellular SARS-CoV-2 RNA in heLa-ACE2 in **d** (MOI = 0.2, 24 hpi, *n* = 3) cells. **f**, Western blot showing the expression level in Huh7.5 cells stably overexpressing the indicated ISGs. Related to Fig. 2i. **g**, Western blot showing the endogenous expression levels of PLSCR1 and LY6E in A549-ACE2 single- or double-KO cells. Related to Fig. 2j. **h**, Bar graph showing the average restriction ratio (−IFNγ/+ IFNγ) in A549-ACE2 cells after SARS-CoV-2 infection of the indicated genotypes in the presence or absence of IFNγ (100 U ml$^{-1}$) 48 hpi (MOI = 0.2) (*n* = 6). Related to Fig. 2j. Data are mean ± s.d. *P* values were calculated using two-sided Student's *t*-test in **c** (middle and right), two-sided Student's *t*-test with Welch's correction in **c** (left) or one-way ANOVA followed by Tukey's multiple comparison test in **e**. Scale bar in **b**, **f**: 500 μm. Experiments in this figure were performed three times.

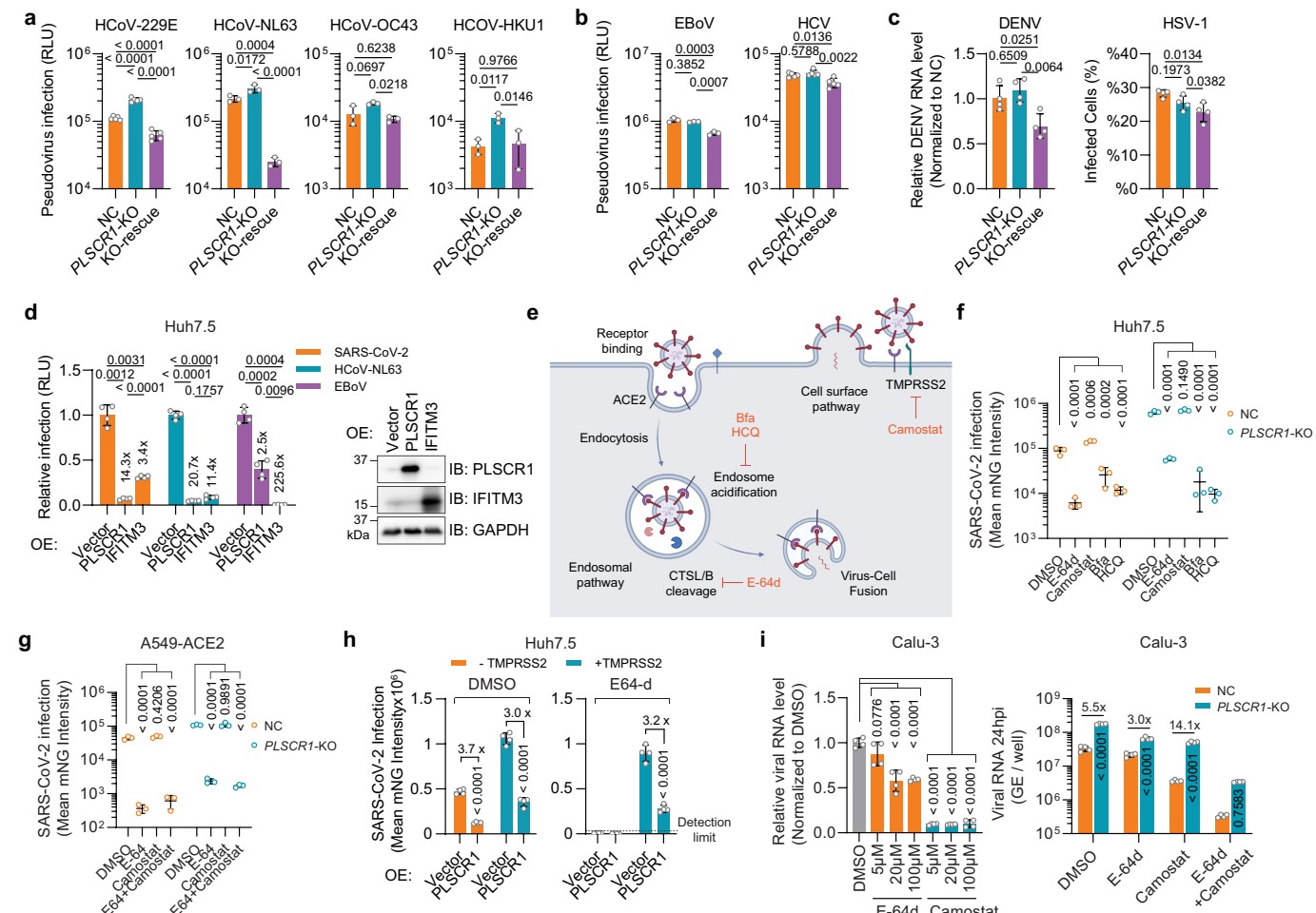

**Extended Data Fig. 4 | PLSCR1 engages both the endosomal pathway and the cell-surface pathway of SARS-CoV-2 entry. a,b,** Quantification of viral entry efficiency in Huh7.5 cells inoculated with pseudoviruses bearing fusion proteins from HCoV-229E ($n = 5$), HCoV-OC43 ($n = 3$), hCoV-NL63 ($n = 3$), hCoV-HKU1 ($n = 3$), EBoV (n = 3) and HCV ($n = 5$). EBoV: Ebola virus, HCV: Hepatitis C virus. **c,** Left, relative amount of intracellular viral RNA in Huh7.5 cells infected with DENV (Dengue virus type I) at an MOI = 0.5 for 24 h. The amount of viral RNA in NC cells was normalized to 1. ($n = 4$) Right: Quantification of % infected cells in HeLa cells infected with HSV-1 VP26-GFP at an MOI of 0.1 for 48 h. **d,** Quantification of the relative entry efficiency of the indicated pseudovirus in Huh7.5 cells overexpressing (OE) PLSCR1 or IFITM3. The luminescence intensity in vector group was normalized to 1. $n = 4$. **e,** Schematic showing the dissection of the cell entry route of SARS-CoV-2. **f,g,** Effect of the indicated compounds on SARS-CoV-2 entry in Huh7.5 cells (MOI = 1, 48 hpi, $n = 3$) (**f**) and A549-ACE2 cells (MOI = 0.2, 24 hpi, $n = 3$) (**g**). E-64d: 20 μM, Camostat: 30 μM, Bfa (Brefeldin a): 10 μM, HCQ: 10 μM. Cells were treated with indicated compounds 2 h before infection. **h,** Quantification of SARS-CoV-2 infection in E-64d (20 μM)

treated or untreated Huh7.5 cells overexpressing vector or PLSCR1 with or without ectopic expression of TMPRSS2 (MOI = 1, 48 hpi). ($n = 4$) **i,** Left, dose response of indicated compounds on SARS-CoV-2 infection in Calu-3 (MOI = 1, 24 hpi, $n = 4$). The amount of viral RNA in DMSO group was normalized to 1. E-64-d and Camostat groups share the same DMSO control group. Right, quantification of SARS-CoV-2 infection in Control or *PLSCR1*-KO Calu-3 (MOI = 1, 24 hpi, $n = 4$) treated with indicated compounds (E-64-d: 20 μM, Camostat: 20 μM). Cells were treated with the indicated compounds 2 h before infection. The amount of viral RNA in NC-DMSO group was normalized to 1. Data are mean ± s.d. $P$ values were calculated using one-way ANOVA followed by Tukey's multiple comparison test in **a–c,d** (HCoV-NL63 group), **f,g,** Brown–Forsythe and Welch ANOVA with Dunnett's post-hoc test in **d** (SARS-CoV-2 and EBoV group), two-sided Student's $t$-test in **h,** two-way ANOVA followed by Tukey's multiple comparison test in **i** (left) or two-way ANOVA followed by Šídák's multiple comparisons test in **i** (right). Experiments in this figure were performed three times.

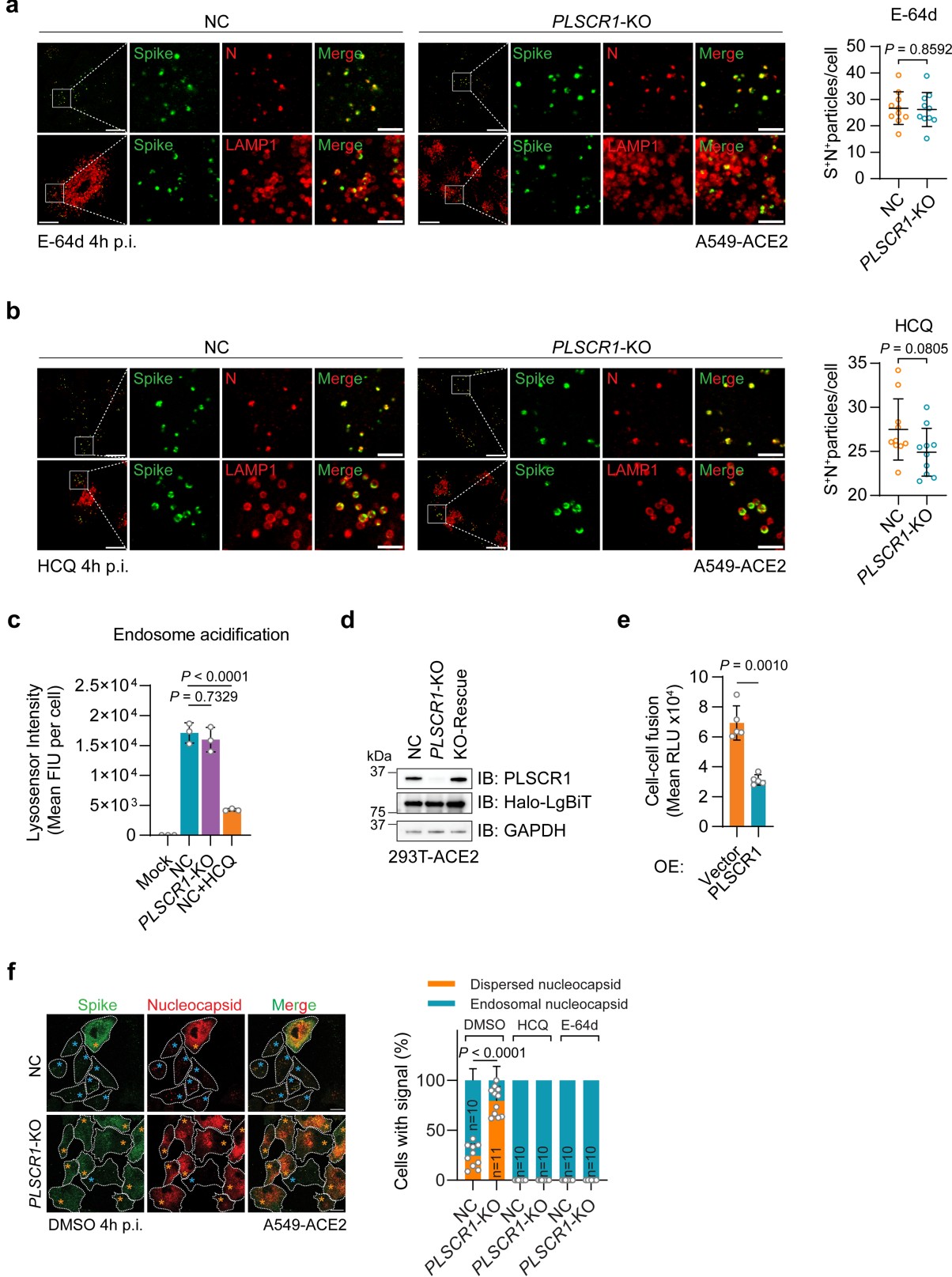

**Extended Data Fig. 5 |** See next page for caption.

**Extended Data Fig. 5 | PLSCR1 blocks virus–cell membrane fusion and the subsequent release of viral content. a**, Representative images showing the localization of indicated proteins in uninfected or infected WT A549-ACE2 cells treated with 20 μM E-64d at MOI = 20 for 4 h (left). Quantification of the average amount of spike and nucleocapsid double-positive particles per cell (right). $n = 10$ image fields (NC: 146 cells, KO: 131 cells). **b**, Representative images showing the localization of indicated proteins in uninfected or infected WT A549-ACE2 cells treated with 20 μM HCQ at MOI = 20 for 4 h (left). Quantification of the average amount of spike and nucleocapsid double-positive particles per cell (right). $n = 10$ image fields (NC: 141 cells, KO: 156 cells). **c**, Quantification of fluorescence intensities of Lysosensor in control or *PLSCR1*-KO A549-ACE2 cells in the presence or absence of HCQ (20 μM). ($n = 3$). **d**, Western blot showing the protein expression levels in 293T-ACE2 cells. Related to Fig. 3h. **e**, Quantification of cell–cell fusion by co-culture of Huh7.5 cells overexpressing vector or PLSCR1 and 293T cells expressing SARS-CoV-2 spike. ($n = 6$). **f**, Left, representative images showing the of distribution of SARS-CoV-2 spike and nucleocapsid protein in control or *PLSCR1*-KO A549-ACE2 cells (MOI = 10, 4 hpi). Orange stars represent cells with dispersed and bright nucleocapsid signal. Blue stars represent cells with endosomal nucleocapsid signal. Right, quantification of the percentage of cells with dispersed or endosomal nucleocapsid signal. Number of cells analysed within each of 10–11 fields (left to right):142, 181, 146, 131, 141 and 156. *n* values are labelled on graph. Data are mean ± s.d. *P* values were calculated using two-sided Student's *t*-test in **a**,**b**,**f**, two-sided Student's *t*-test with Welch's correction in **e** or one-way ANOVA followed by Tukey's multiple comparison test in **c**. Scale bar in **a**, **b**, **f**: 20 μm, inlays: 5 μm. Experiments in this figure were performed three times.

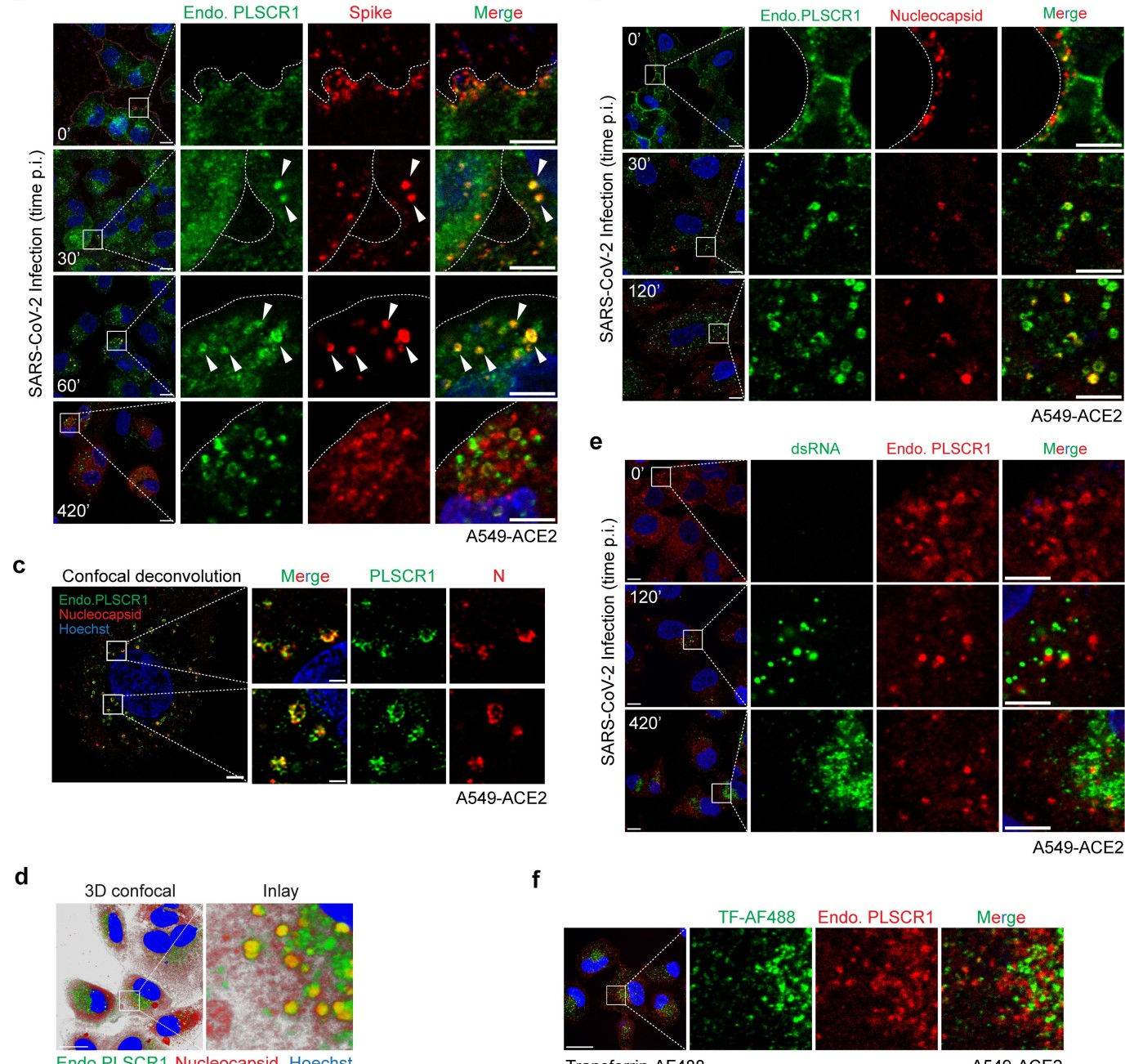

**Extended Data Fig. 6 | Cell biology of PLSCR1 in anti-SARS-CoV-2 activity.**
**a**, Representative images showing the localization of endogenous PLSCR1 and SARS-CoV-2 spike in WT A549-ACE2 cells during the early stage of virus entry. Time post-infection is indicated. (MOI = 25). **b**, Representative images showing the localization of endogenous PLSCR1 and SARS-CoV-2 nucleocapsid in WT A549-ACE2 cells during the early stage of virus entry. Time post-infection is indicated. (MOI = 25). **c**, Deconvolution confocal images showing the colocalization of endogenous PLSCR1 and SARS-CoV-2 nucleocapsid. (MOI = 25, 2 hpi). **d**, Three-dimensional confocal microscopy images showing the colocalization of endogenous PLSCR1 and SARS-CoV-2 nucleocapsid in A549-ACE2 cells at 2 hpi. (MOI = 25). **e**, Representative images of the localization of endogenous PLSCR1 and viral replication centre (indicated by dsRNA) in A549-ACE2 cells during virus infection. Time post-infection is indicated. (MOI = 25). **f**, Representative images showing the localization of endogenous PLSCR1 and human transferrin–Alexa Fluor 488 30 min after treatment. Scale bar in **a**,**b**,**e**: 10 μm, inlays: 5 μm. Scale bar in **c**: 5 μm, inlays: 1μm. Scale bar in **d**,**f**: 20 μm, inlays: 5 μm. Experiments in this figure were performed three times, except **a**–**c** (five times).

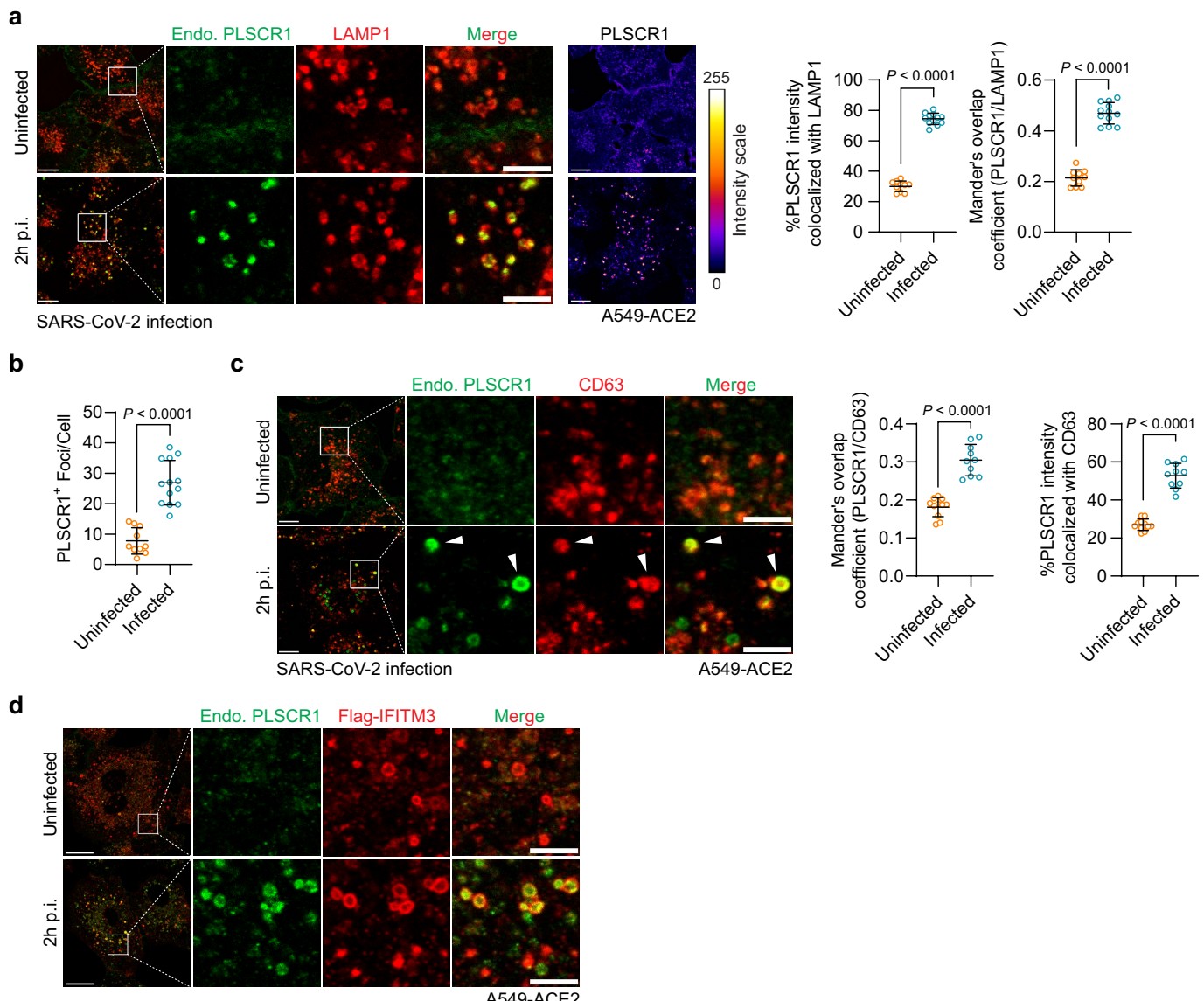

**Extended Data Fig. 7 | Comparison of PLSCR1 with other endolysosomal membrane proteins during SARS-CoV-2 infection. a**, Left, confocal images showing the localization of endogenous PLSCR1 and LAMP1 in uninfected or infected WT A549-ACE2 cells (MOI = 20, 2 hpi). Middle and right, quantification of the percentage of PLSCR1 intensity colocalized with LAMP1 (middle) or the Mander's colocalization coefficient of PLSCR1 with LAMP1 (right). Uninfected group: *n* = 10 image fields (63 cells), infected group: *n* = 13 image fields (96 cells). **b**, Quantification of the average number of PLSCR1-positive foci per cell. Uninfected group: *n* = 10 image fields (63 cells), infected group: *n* = 13 image fields (96 cells). **c**, Left, representative images showing the localization of endogenous PLSCR1 and CD63 in uninfected or infected WT A549-ACE2 cells

(MOI = 20, 2 hpi). Middle and right, quantification of the Mander's colocalization coefficient of PLSCR1 with CD63 (middle) or the percentage of PLSCR1 intensity colocalized with CD63 (right). Uninfected group: *n* = 11 image fields (51 cells), infected group: *n* = 10 image fields (54 cells). **d**, Representative images showing the localization of endogenous PLSCR1 and Flag–IFITM3 in uninfected or infected WT A549-ACE2 cells (MOI = 20, 2 hpi). Data are mean ± s.d. *P* values were calculated using two-sided Student's *t*-test in **a**,**b**,**c** (middle) or two-sided Student's *t*-test with Welch's correction in **c** (right). Scale bar in **a** and **c**: 10 μm, inlays: 5 μm. Scale bar in **d**: 20 μm, inlays: 5 μm. Experiments in this figure were performed three times.

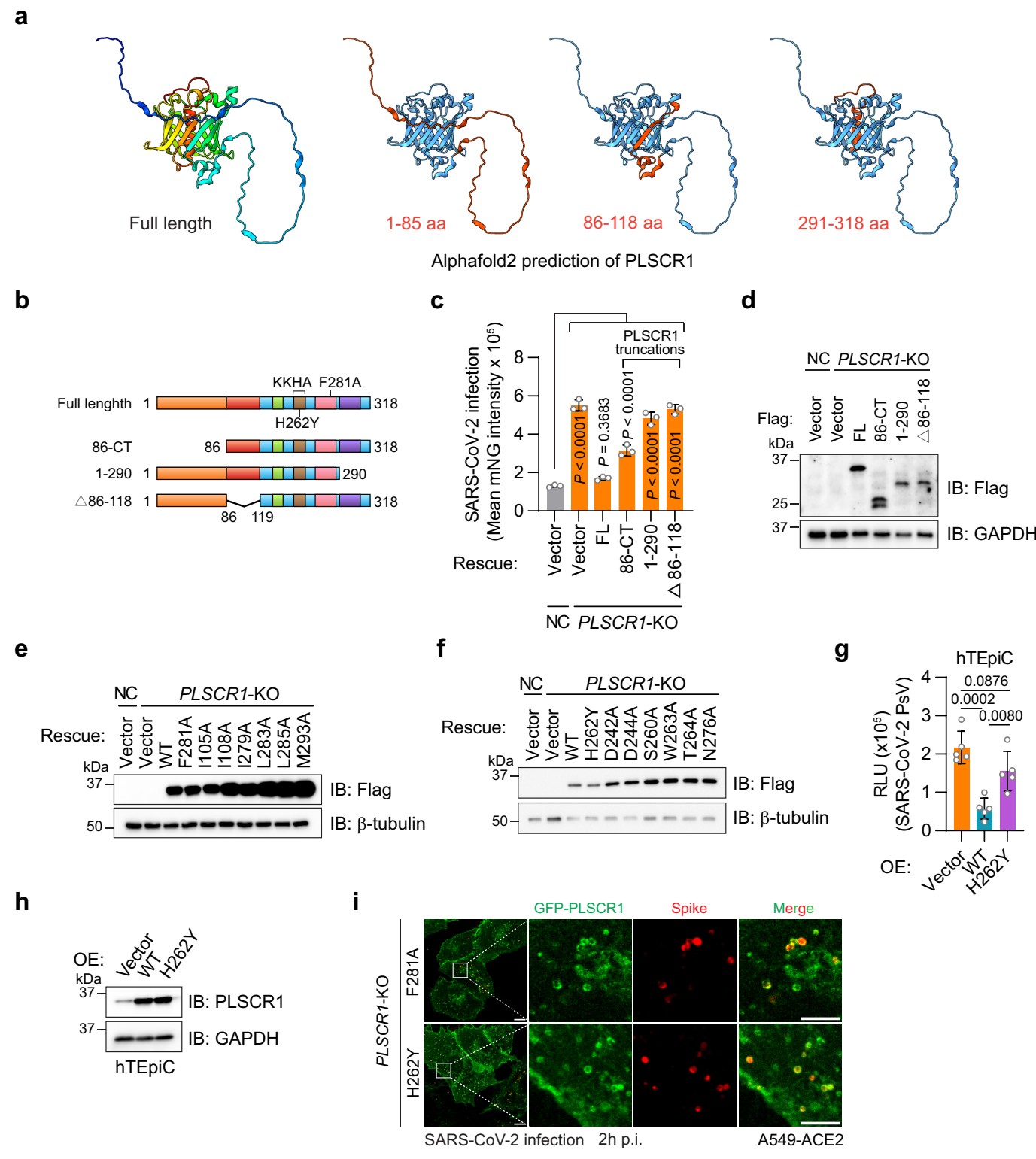

**Extended Data Fig. 8 | Structure–function determinants of PLSCR1 anti-SARS-COV-2 activity. a**, AlphaFold2 structure prediction of sequences deleted from PLSCR1. **b**, Domain map depicting the generation of truncations and mutations of PLSCR1. **c**, Quantification of SARS-CoV-2 infection in control or *PLSCR1*-KO Huh7.5 cells expressing the indicated truncations (MOI = 1, 48 hpi). (n = 3). **d**, Western blot showing the expression level of the indicated truncations or mutations of PLSCR1 in Huh7.5 cells in **c**. **e**,**f**, Western blots showing the expression level of the indicated truncations or mutations of PLSCR1 in Huh7.5 cells. Related to Fig. 4f. **g**, Quantification of SARS-CoV-2 pseudovirus infection in hTEpiCs stably expressing the indicated PLSCR1 mutants. Luciferase activity was measured at 48 hpi (n = 5). **h**, Western blot showing the expression level of the indicated mutations of PLSCR1 in hTEpiCs in **g**. **i**, Representative images showing the localization of GFP–PLSCR1 F281A and H262Y mutants on SARS-CoV-2-containing vesicles in A549-ACE2 *PLSCR1*-KO cells infected with SARS-CoV-2 for 2 h (MOI = 25). Data are mean ± s.d. *P* values were calculated using one-way ANOVA followed by Tukey's multiple comparison test in **c**,**g**. Scale bar in **i**: 10 μm, inlays: 5 μm. Experiments in this figure were performed three times.

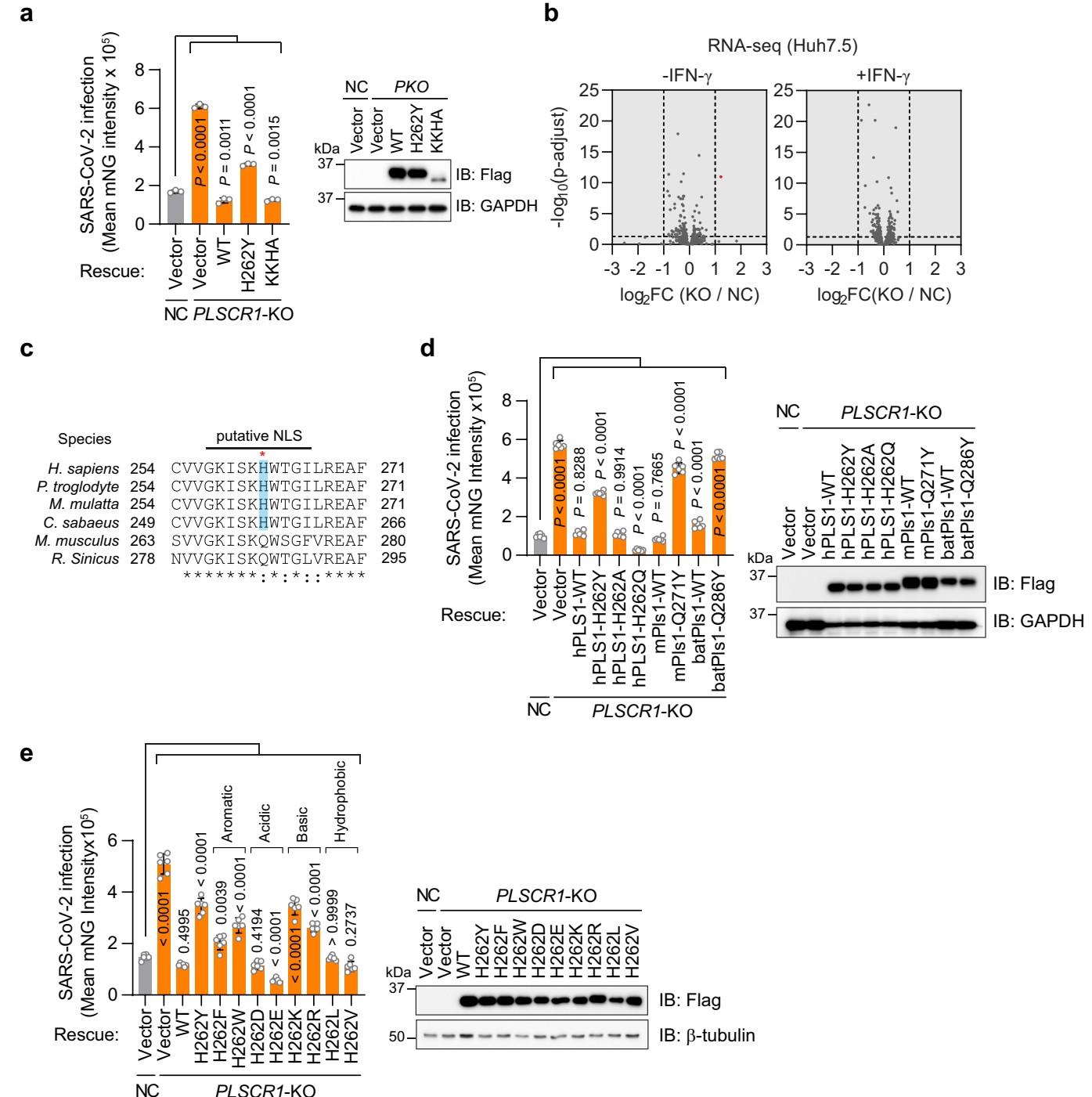

**Extended Data Fig. 9 | The anti-SARS-CoV-2 activity of PLSCR1 is independent of its nuclear localization signal. a**, Left, quantification of SARS-CoV-2 infection in Huh7.5 cells expressing the indicated mutants of PLSCR1 (MOI = 1, 48 hpi). KKHA: $K^{258}K^{261}H^{262}$ to Ala. Right, western blot showing the protein expression. ($n$ = 3). **b**, Volcano plot comparing the mRNA expression level between NC and *PLSCR1*-KO Huh7.5 in the absence or presence of IFNγ. Transcripts with $Log_2FC > 1$ and adjusted $P$ value < 0.05 are highlighted in red. ($n$ = 3). **c**, Sequence homology alignment of sequences flanking the H262 residue in PLSCR1 orthologues. **d**, Left, quantification of SARS-CoV-2 infection

in NC or *PLSCR1*-KO Huh7.5 cells expressing the indicated mutants of human, mouse or bat PLSCR1. (MOI = 1, 48 hpi). Right, western blot showing the protein expression. ($n$ = 6). **e**, Left, quantification of SARS-CoV-2 infection in NC or *PLSCR1*-KO Huh7.5 cells expressing the indicated single-point substitutions of the H262 residue. (MOI = 1, 48 hpi). Right, western blot showing the protein expression. ($n$ = 6). Data are mean ± s.d. $P$ values were calculated using one-way ANOVA followed by Tukey's multiple comparison test in **a**,**d**,**e** or DESeq2 algorithm in **b**. Experiments in this figure were performed three times.

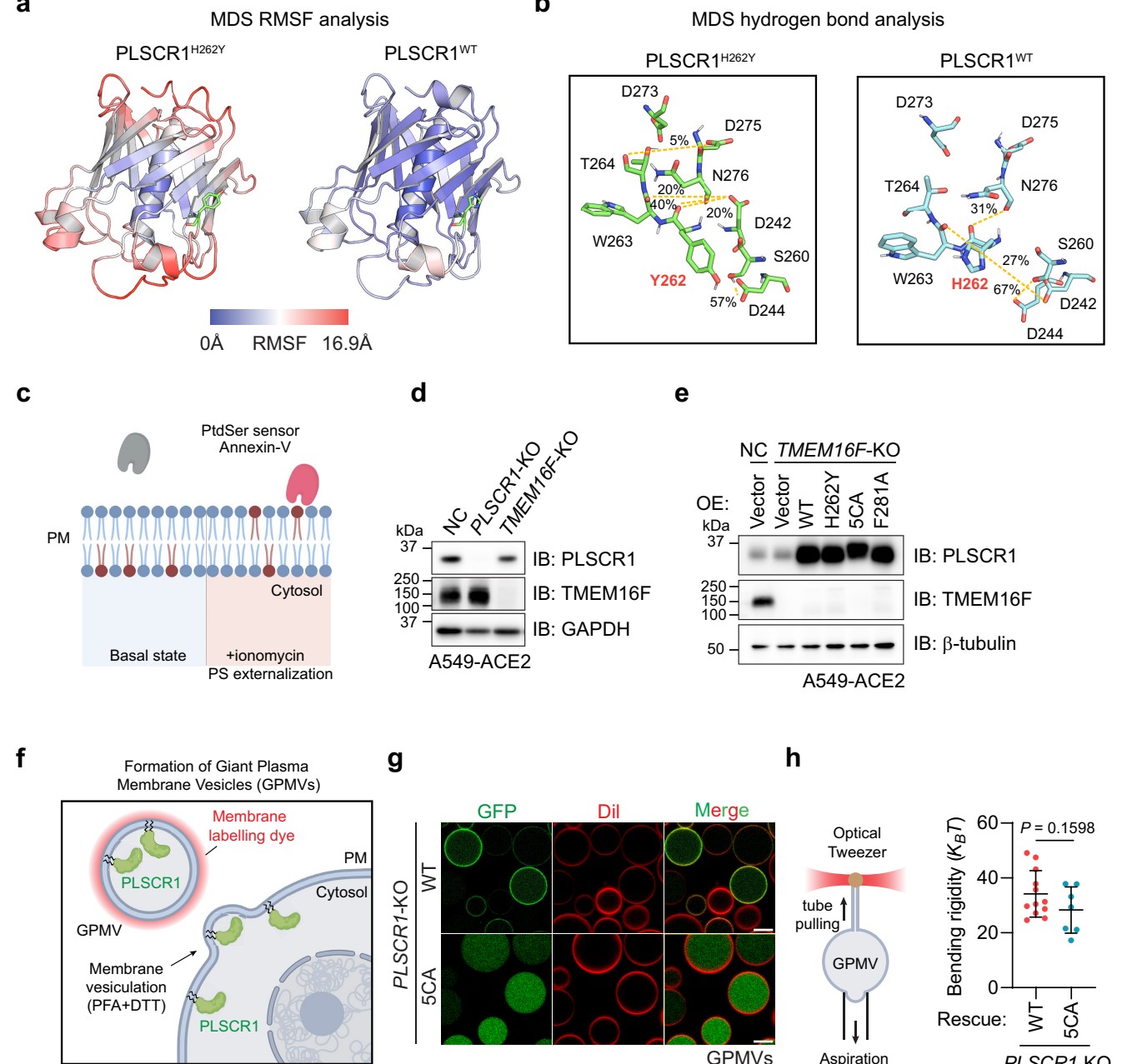

**Extended Data Fig. 10 | The anti-SARS-CoV-2 activity of PLSCR1 is uncoupled from its phospholipid scramblase activity. a**, MDS analysis of RMSF for the β-barrel domain of PLSCR1 WT and the H262Y mutant. **b**, MDS analysis of the hydrogen bond network surrounding the amino acid 262 residue in WT and the H262Y mutant. **c**, Schema depicting the PS externalization assay. Externalized PS was detected by membrane-impermeable Annexin V conjugated with Alexa647. **d**, Western blot showing the protein expression level. Related to

Fig. 4g. **e**, Western blot showing the protein expression level. Related to Fig. 4h. **f**, Schema depicting the preparation of GPMVs. **g**, Representative images showing GPMVs generated from A549-ACE2-*PLSCR1*-KO cells stably expressing GFP–PLSCR1 WT or 5CA protein. (WT: *n* = 12; 5CA: *n* = 7). **h**, GPMV bending rigidity values. Data are mean ± s.d. *P* values were calculated using two-sided Student's *t*-test in **h**. Scale bar in **g**: 10 μm. Experiments in this figure were performed three times, except **h** (two times).

| | |
|---|---|

# Reporting Summary

## Statistics

For all statistical analyses, confirm that the following items are present in the figure legend, table legend, main text, or Methods section.

| n/a | Confirmed | |
|---|---|---|
| ☐ | ☒ | The exact sample size (*n*) for each experimental group/condition, given as a discrete number and unit of measurement |
| ☐ | ☒ | A statement on whether measurements were taken from distinct samples or whether the same sample was measured repeatedly |
| ☐ | ☒ | The statistical test(s) used AND whether they are one- or two-sided *Only common tests should be described solely by name; describe more complex techniques in the Methods section.* |
| ☒ | ☐ | A description of all covariates tested |
| ☐ | ☒ | A description of any assumptions or corrections, such as tests of normality and adjustment for multiple comparisons |
| ☐ | ☒ | A full description of the statistical parameters including central tendency (e.g. means) or other basic estimates (e.g. regression coefficient) AND variation (e.g. standard deviation) or associated estimates of uncertainty (e.g. confidence intervals) |
| ☐ | ☒ | For null hypothesis testing, the test statistic (e.g. *F*, *t*, *r*) with confidence intervals, effect sizes, degrees of freedom and *P* value noted *Give P values as exact values whenever suitable.* |
| ☒ | ☐ | For Bayesian analysis, information on the choice of priors and Markov chain Monte Carlo settings |
| ☒ | ☐ | For hierarchical and complex designs, identification of the appropriate level for tests and full reporting of outcomes |
| ☒ | ☐ | Estimates of effect sizes (e.g. Cohen's *d*, Pearson's *r*), indicating how they were calculated |

*Our web collection on statistics for biologists contains articles on many of the points above.*

## Software and code

Policy information about availability of computer code

| Data collection | SoftMax® Pro Software (v.7) was used to collect data from microplate reader. BD FACSDiva (v.8) was used to collect flow cytometry data. Bio-Rad Image Lab Touch Software (v.2.2.0.08) was used to collect images of protein and DNA gels. Microscopy images were obtained using LAS X (v 3.7.4). Gen5 (BioTek) and CellReporterXpress (Molecular devices) were used to collect fluorescence images of 96-well plates. |
|---|---|
| Data analysis | GraphPad Prism 9 was used to generate all graphs and perform statistical analysis.<br>LAS X (v 3.7.4), FUJI/ImageJ or Imaris 9.8 were used to process confocal images.<br>Gen5 (BioTek) and CellReporterXpress (Molecular devices) were used to process high-content images of 96-well plates.<br>FlowJo 10.8.1 was used to process and visualize FACS data.<br>SoftWoRx (v7.0) was used to process live cell images.<br>Vutara SRX 7.0.06 software was used to process 4Pi single-molecule switching (4Pi-SMS) images.<br>Whole-genome CRISPR/Cas9 screening data was analyzed using MAGeCK (v.0.5.6).<br>RNA seq was analyzed using STAR (version 2.8), DESeq2 (version 3.9) and R version 3.5.0.<br>Microsoft Excel 2010 was used to record and store data.<br>ChimeraX 1.4 (https://www.rbvi.ucsf.edu/chimerax/) was used to analyze protein structures.<br>GROMACS version 2021.3 was used for molecular dynamic simulation and RMSF analysis.<br>Coarse-grained simulation was analyzed using Memembed 1.15, Insane 3, Martinize 2 and Python version 3.7.<br>PyMOL (2.3.0) was used for RMSF and H-bond visualization<br>VMD (1.9.4) was used for the hydrogen bond analysis and movie creation<br>The membrane bending rigidity data was analyzed using MATLAB 9.13 and Origin 2023. |

For manuscripts utilizing custom algorithms or software that are central to the research but not yet described in published literature, software must be made available to editors and reviewers. We strongly encourage code deposition in a community repository (e.g. GitHub). See the Nature Portfolio guidelines for submitting code & software for further information.

## Data

Policy information about availability of data

All manuscripts must include a data availability statement. This statement should provide the following information, where applicable:
- Accession codes, unique identifiers, or web links for publicly available datasets
- A description of any restrictions on data availability
- For clinical datasets or third party data, please ensure that the statement adheres to our policy

The data supporting the findings of this study are available within the paper and its Supplementary Information files. Genome-wide CRISPR screening data are provided in Supplementary Tables 2 and 3. RNA-seq data are provided in Supplementary Tables 4 and 5. Full versions of all blots are provided in Supplementary Fig. 1. The gating strategies of flow cytometry are provided in Supplementary Fig. 2. The human genome reference (hg38) used in RNA-seq analysis is available in the NCBI genome assembly with accession number GCF_000001405.39. The protein expression profiles are available in the web-based Human Protein Atlas database (https://www.proteinatlas.org/). The transcriptional factor binding profiles are available in the web-based JASPAR database (https://jaspar.genereg.net/). The Materials, reagents or other experimental data are available from the corresponding author upon request. Source data are provided with this paper.

## Human research participants

Policy information about studies involving human research participants and Sex and Gender in Research.

| | |
|---|---|
| Reporting on sex and gender | N/A |
| Population characteristics | N/A |
| Recruitment | N/A |
| Ethics oversight | N/A |

Note that full information on the approval of the study protocol must also be provided in the manuscript.

# Field-specific reporting

Please select the one below that is the best fit for your research. If you are not sure, read the appropriate sections before making your selection.

☒ Life sciences          ☐ Behavioural & social sciences          ☐ Ecological, evolutionary & environmental sciences

For a reference copy of the document with all sections, see nature.com/documents/nr-reporting-summary-flat.pdf

# Life sciences study design

All studies must disclose on these points even when the disclosure is negative.

| | |
|---|---|
| Sample size | The sample size and the results of statistical analyses are described in the relevant figure legends. Sample size was determined based on experimental trials and with consideration of previous publications on similar experiments (ref. 22, 23, 24, 25) to allow for confident statistical analyses. No statistical methods were used to predetermine sample sizes. |
| Data exclusions | No data was excluded. |
| Replication | All experimental findings were replicated at least three times unless otherwise mentioned in figure legends. |
| Randomization | For validation of whole-genome screening candidates and the experiments measuring the function of PLSCR1 mutants, cells of each genotype were seeded on culture plate in random orders or randomly assigned to different treatments. Otherwise, randomization was not performed. Data variability was controlled by multiple biological replicates and multiple technical replicates within an experiment. Cells were grown under the same conditions in dishes or plates to minimize unexpected environmental variations. |
| Blinding | For validation of whole-genome screening candidates, samples were labeled as code and were blind to the individual who performed the experiment. For microscopy data collection, the samples of different treatments or genotypes were randomly selected for imaging. The fields of view were chosen on a random basis. For other experiments, blinding was not performed because different genotypes or wells require different treatments or conditions. |

# Reporting for specific materials, systems and methods

We require information from authors about some types of materials, experimental systems and methods used in many studies. Here, indicate whether each material, system or method listed is relevant to your study. If you are not sure if a list item applies to your research, read the appropriate section before selecting a response.

## Materials & experimental systems

| n/a | Involved in the study |
|---|---|
| ☐ | ☒ Antibodies |
| ☐ | ☒ Eukaryotic cell lines |
| ☒ | ☐ Palaeontology and archaeology |
| ☒ | ☐ Animals and other organisms |
| ☒ | ☐ Clinical data |
| ☒ | ☐ Dual use research of concern |

## Methods

| n/a | Involved in the study |
|---|---|
| ☒ | ☐ ChIP-seq |
| ☐ | ☒ Flow cytometry |
| ☒ | ☐ MRI-based neuroimaging |

## Antibodies

Antibodies used

The following antibodies were purchased from Proteintech:
Rabbit anti-GAPDH monoclonal antibody (Clone 1E6D9) (60004-1-Ig) (WB: 1:2000)
Mouse anti-GFP tag monoclonal antibody (Clone 1E10H7) (66002-1-Ig) (WB: 1:2000)
Rabbit anti-PLSCR1 polyclonal antibody (11582-1-AP) (WB: 1:2000)
Mouse anti-Halo tag monoclonal antibody (Clone 28a8) (WB: 1:500)
Rabbit anti-TMEM41B polyclonal antibody (29270-1-AP) (WB: 1:2000)
Rabbit anti-IFITM3 polyclonal antibody (11714-1-AP) (WB: 1:2000)

The following antibodies were purchased from Cell Signaling:
Rabbit anti-Na,K-ATPase polyclonal antibody (#3010S) (WB:1:1000)
Rabbit anti-Flag tag monoclonal antibody (Clone D6W58) (14793S) (WB: 1:2000)
Rabbit anti-β-Tubulin monoclonal antibody (Clone 9F3) (#2128S) (WB: 1:1000)

The following antibody was purchased from R&D Systems:
Goat anti-ACE2 polyclonal antibody (AF933) (WB: 1:4000)

The following antibodies were purchased from Sigma-Aldrich:
Mouse anti-PLSCR1 monoclonal antibody (Clone 4D2) (MABS483) (IFA: 1:200)
Mouse anti-dsRNA monoclonal antibody (Clone rJ2) (MABE1134) (IFA: 1:200)
Rabbit anti-TMEM16F polyclonal antibody (HPA038958) (WB: 1:2000)
Sheep anti-mouse IgG horse-radish peroxidase-conjugated secondary antibody (GENXA931-1ML) (WB:1:5000)
Sheep anti-rabbit IgG horse-radish peroxidase-conjugated secondary antibody (GENA934-1ML) (WB:1:5000)

The following antibody was purchased from Sino Biological:
Rabbit anti-SARS-CoV-2 nucleocapsid monoclonal antibody (Clone 019) (40143-R019) (IFA: 1:200)
Rabbit anti-SARS-CoV-2 Spike S2 antibody (40590-T62) (WB: 1:2000)

The following antibody was purchased from BD Biosciences:
Mouse anti-EEA1 monoclonal antibody (Clone 14) (610456) (IFA: 1:100)

The following antibody was purchased from Abcam:
Rabbit anti-LY6E monoclonal antibody (Clone EPR26038-105)( ab300399) (WB: 1:1000)

The following antibody was purchased from GeneTex:
Mouse monoclonal antibody against SARS-CoV-2 Spike (Clone 1A9) (GTX632604) (IFA: 1:200)

The following antibodies were purchased from ThermoFisher:
Donkey anti-mouse IgG Alexa Fluoro-488 (A21202) (IFA: 1:500)
Donkey anti-rabbit IgG Alexa Fluoro-488 (A21206) (IFA: 1:500)
Donkey anti-mouse IgG Alexa Fluoro-568 (A10037) (IFA: 1:500)
Donkey anti-rabbit IgG Alexa Fluoro-568 (A10042) (IFA: 1:500)
Donkey anti-mouse IgG Alexa Fluoro-647 (A32787) (IFA: 1:500)
Donkey anti-rabbit IgG Alexa Fluoro-647 (A31573) (IFA: 1:500)
Donkey anti-Goat IgG horse-radish peroxidase-conjugated secondary antibody (PA1-28664) (WB: 1:5000)

The following antibodies were obtained from BEI Resources, NIAID, NIH:
Rabbit anti-SARS-CoV-2 spike monoclonal antibody (Clone number not available) (NR-53788) (IFA: 1:200)
Mouse anti-SARS-CoV-2 nucleocapsid monoclonal antibody (Clone 05) (NR-53792) (IFA: 1:200)
Rabbit anti-SARS-CoV-2 nucleocapsid monoclonal antibody (Clone 001) (NR-53791) (IFA: 1:200)

The following antibody was purchased from Jackson ImmunoResearch:
Goat anti-mouse Fab AF647 (115-607-003) (IFA: 1:200)

The following antibody was purchased from Biotium:
Goat anti-rabbit IgG CF660C (Cat# 20813) (IFA: 1:200)

| Validation | Antibodies were validated by the manufacturers/providers as well as RRID database: |
|---|---|

Rabbit anti-GAPDH monoclonal antibody (60004-1-Ig): https://www.ptglab.com/products/GAPDH-Antibody-60004-1-Ig.htm. RRID: AB-2107436.
Mouse anti-GFP tag monoclonal antibody (66002-1-Ig): https://www.ptglab.com/products/eGFP-Antibody-66002-1-Ig.htm. RRID: AB_11182611.
Rabbit anti-PLSCR1 polyclonal antibody (11582-1-AP): https://www.ptglab.com/products/PLSCR1-Antibody-11582-1-AP.htm. RRID: AB_2165659.
Mouse anti-Halo tag monoclonal antibody (28a8): https://www.ptglab.com/products/Halo-antibody-28A8.htm. RRID: AB_2827565.
Rabbit anti-TMEM41B polyclonal antibody (29270-1-AP): https://www.ptglab.com/products/TMEM41B-Antibody-29270-1-AP.htm. RRID:AB_2918264.
Rabbit anti-IFITM3 polyclonal antibody (11714-1-AP): https://www.ptglab.com/products/IFITM3-Antibody-11714-1-AP.htm. RRID:AB_2295684.
Rabbit anti-Na,K-ATPase polyclonal antibody (#3010S): https://www.cellsignal.com/products/primary-antibodies/na-k-atpase-antibody/3010. RRID:AB_2060983.
Rabbit anti-Flag tag monoclonal antibody (14793S): https://www.cellsignal.com/products/primary-antibodies/dykddddk-tag-d6w5b-rabbit-mab-binds-to-same-epitope-as-sigma-s-anti-flag-m2-antibody/14793. RRID:AB_2572291.

Goat anti-ACE2 polyclonal antibody (AF933): https://www.rndsystems.com/products/human-mouse-rat-hamster-ace-2-antibody_af933. RRID:AB_355722.

Mouse anti-PLSCR1 monoclonal antibody (MABS483): https://www.sigmaaldrich.com/US/en/product/mm/mabs483.
Mouse anti-dsRNA monoclonal antibody (MABE1134): https://www.sigmaaldrich.com/US/en/product/mm/mabe1134. RRID:AB_2819101.
Rabbit anti-SARS-CoV-2 nucleocapsid monoclonal antibody (40143-R019): https://www.sinobiological.com/antibodies/cov-nucleocapsid-40143-r019. RRID:AB_2827973.
Rabbit anti-SARS-CoV-2 Spike S2 antibody (40590-T62):https://www.sinobiological.com/antibodies/cov-spike-40590-t62
Mouse anti-EEA1 monoclonal antibody (610456): https://www.bdbiosciences.com/en-nz/products/reagents/microscopy-imaging-reagents/immunofluorescence-reagents/purified-mouse-anti-eea1.610456. RRID:AB_397829.
Rabbit anti-LY6E polyclonal antibody (ab300399): https://www.abcam.com/products/primary-antibodies/ly6esca-2-antibody-epr26038-105-ab300399.html
Mouse monoclonal antibody against SARS-CoV-2 Spike (GTX632604): https://www.genetex.com/Product/Detail/SARS-CoV-SARS-CoV-2-COVID-19-spike-antibody-1A9/GTX632604. RRID:AB_2864418.
Donkey anti-mouse IgG Alexa Fluoro-488 (A21202): https://www.thermofisher.com/antibody/product/Donkey-anti-Mouse-IgG-H-L-Highly-Cross-Adsorbed-Secondary-Antibody-Polyclonal/A-21202. RRID:AB_141607.
Donkey anti-rabbit IgG Alexa Fluoro-488 (A21206): https://www.thermofisher.com/antibody/product/Donkey-anti-Rabbit-IgG-H-L-Highly-Cross-Adsorbed-Secondary-Antibody-Polyclonal/A-21206. RRID:AB_2535792.
Donkey anti-mouse IgG Alexa Fluoro-568 (A10037): https://www.thermofisher.com/antibody/product/Donkey-anti-Mouse-IgG-H-L-Highly-Cross-Adsorbed-Secondary-Antibody-Polyclonal/A10037. RRID:AB_2534013.
Donkey anti-rabbit IgG Alexa Fluoro-568 (A10042): https://www.thermofisher.com/antibody/product/Donkey-anti-Rabbit-IgG-H-L-Highly-Cross-Adsorbed-Secondary-Antibody-Polyclonal/A10042. RRID:AB_2534017.
Donkey anti-mouse IgG Alexa Fluoro-647 (A32787): https://www.thermofisher.com/antibody/product/Donkey-anti-Mouse-IgG-H-L-Highly-Cross-Adsorbed-Secondary-Antibody-Polyclonal/A32787. RRID:AB_2762830.
Donkey anti-rabbit IgG Alexa Fluoro-647 (A31573): https://www.thermofisher.com/antibody/product/Donkey-anti-Rabbit-IgG-H-L-Highly-Cross-Adsorbed-Secondary-Antibody-Polyclonal/A-31573. RRID:AB_2536183.
Rabbit anti-SARS-CoV-2 spike monoclonal antibody (NR-53788): https://www.beiresources.org/Catalog/BEIMonoclonalAntibodies/NR-53788.aspx.
Mouse anti-SARS-CoV-2 nucleocapsid monoclonal antibody (NR-53792): https://www.beiresources.org/Catalog/BEIMonoclonalAntibodies/NR-53792.aspx.
Rabbit anti-SARS-CoV-2 nucleocapsid monoclonal antibody (NR-53791): https://www.beiresources.org/Catalog/BEIMonoclonalAntibodies/NR-53791.aspx.
Goat anti-mouse Fab AF647 (115-607-003): https://www.jacksonimmuno.com/catalog/products/115-607-003. RRID:AB_2338931.
Goat anti-rabbit IgG CF660C (Cat# 20813): https://biotium.com/product/goat-anti-rabbit-igg-hl-highly-cross-absorbed-cf-dye-storm/?attribute_pa_conjugation=cf660c.
Rabbit anti-TMEM16F polyclonal antibody (HPA038958): https://www.sigmaaldrich.com/US/en/product/sigma/hpa038958. RRID:AB_10672835
Sheep anti-mouse IgG horse-radish peroxidase-conjugated secondary antibody (GENXA931-1ML): https://www.sigmaaldrich.com/US/en/product/sigma/genxa9311ml. RRID:AB_772209.
Sheep anti-rabbit IgG horse-radish peroxidase-conjugated secondary antibody (GENA934-1ML): https://www.sigmaaldrich.com/US/en/product/sigma/gena9341ml. RRID:AB_772206.
Donkey anti-Goat IgG horse-radish peroxidase-conjugated secondary antibody (PA1-28664): https://www.thermofisher.com/antibody/product/Donkey-anti-Goat-IgG-H-L-Secondary-Antibody-Polyclonal/PA1-28664. RRID:AB_10990162.

# Eukaryotic cell lines

Policy information about cell lines and Sex and Gender in Research

| Cell line source(s) | Huh7.5 (kind gift from C. Wilen, commercially available in Apath LLC); A549-ACE2 (BEI Resources #NR-53821); Vero E6 (ATCC CRL-1586); HEK293T (ATCC CRL-3216); HeLa (ATCC CCL-2); Tonsil (UT-SCC-60A); HaCaT (kind gift from D. DiMaio, commercially available in AddexBio Technologies; #T0020001); LET1 (BEI Resources; NR-42941); hTEpiC (ScienCell #3220); Calu-3 (ATCC HTB-55).<br><br>Cell lines stably expressing human ACE2 (HeLa-ACE2, 293T-ACE2, Tonsil-ACE2 and HaCaT-ACE2) were generated from the aforementioned cell lines. For more information, please refer to the Methods part. |
|---|---|

| | |
|---|---|
| Authentication | Vero-E6, HEK293T, HeLa and Calu-3 cells were obtained from and pre-authenticated by ATCC and used at low passages. hTEpiC was obtained and pre-authenticated by ScienCell and used at low passages . A549-ACE2 and LET1 cells were obtained and pre-authenticated by BEI resource and used at low passages. CRISPR-Cas9 knockout cells constructed by us were validated by absence of protein expression by western blot. Cell morphology was used as authentication. |
| Mycoplasma contamination | All cell lines were tested routinely for mycoplasma contamination either by PCR analysis. If mycoplasma was detected, the cell line was discarded and any results acquired from the contaminated cell lines were reconfirmed in mycoplasma negative cells. |
| Commonly misidentified lines<br>(See ICLAC register) | None. |

# Flow Cytometry

## Plots

Confirm that:

☒ The axis labels state the marker and fluorochrome used (e.g. CD4-FITC).

☒ The axis scales are clearly visible. Include numbers along axes only for bottom left plot of group (a 'group' is an analysis of identical markers).

☒ All plots are contour plots with outliers or pseudocolor plots.

☒ A numerical value for number of cells or percentage (with statistics) is provided.

## Methodology

| | |
|---|---|
| Sample preparation | For CRISPR screening: Cells infected with SARS-CoV-2-mNeonGreen were fixed by 4% PFA for 30 min. Cells were washed 3 times by PBS. Cells were resuspended in FACS buffer (1xPBS, 1% FBS, 5mM EDTA) and filtered through a 40-μm cell strainer prior to FACS sorting using a BD FACSAria.<br><br>For PS externalization assay: Cells were digested by trypsin, spun down at 200 × g and washed twice with PBS. Cell pellets were resuspended in 100 μL of 1 × binding buffer (obtained from Thermo) at a density of 5×106 cells /mL and treated with DMSO or 10 μM ionomycin for 10 min. Cells were subsequently incubated with 5 μL Annexin V AF647 (Thermo) and DAPI for 20 min at room temperature followed by the addition of 400 μL 1× binding buffer. Cells were then analyzed using Beckman CytoFLEX S flow cytometer (APC filter). |
| Instrument | BD FACSAria was used to collect all data and perform cell sorting. Beckman CytoFLEX S flow cytometer was used to analyze flow cytometry data for PS externalization. |
| Software | BD FACSDiva (v.8) was used to collect flow cytometry data. FlowJo (v.10.2) was used to analyze data. |
| Cell population abundance | Populations sorted on mNeonGreen levels were estimated to be >95% pure based on analysis of post-sorting populations in pilot studies. |
| Gating strategy | The boundary for the mNeonGreen-High population was set based on maximal difference observed in wildtype cells between the interferon-γ treated and inteferon-γ non-treated samples.<br><br>For PS externalization assay, DAPI negative cells were selected. The cut-off of Annexin V AF647 positive cells was set based on the peak position in WT A549-ACE2 cells. |

☒ Tick this box to confirm that a figure exemplifying the gating strategy is provided in the Supplementary Information.

