## [Peer Review File · Nature]

Manuscript Title: PLSCR1 is a cell-autonomous defense factor against SARS-CoV-2 infection

Reviewer Comments & Author Rebuttals

Reviewer Reports on the Initial Version:

Referees' comments:

Referee #1 (Remarks to the Author):

In this submission, Xu et al. describe the antiviral properties of the lipid scramblase PLSCR1, particularly in the context of SARS-CoV-2 infection. PLSCR1 was a prioritized hit from a whole-genome CRISPR/Cas9 screen performed in hepatocytes and lung epithelial cells, and performs constitutive antiviral function as is further upregulated by IFN-gamma (as well as type I and type III IFNs). Through a lot of comprehensive experiments that rely heavily on confocal immunofluorescence microscopy, the authors propose that PLSCR1 inhibits entry of SARS-CoV-2 and other coronaviruses that use type I fusion machinery. The authors make a solid case that PLSCR1 localization to virus-containing vesicles and endolysosomes is an essential component of its antiviral activity, but what it does exactly to prevent virus fusion remains unclear. PLSCR1 does not seem to influence endolysosomal acidification or cathepsin activity. Therefore, it is tempting to speculate that the influence of PLSCR1 on the externalization of lipids (like phosphatidylserine) may disfavor virus-cell membrane fusion in endolysosomes. Given that several lipid scramblases have been described recently to influence the function of virus fusion machinery (TMEM41B, TMEM16F, and others), this is an important contribution to a budding field in viral immunology. Overall, this is very interesting description of a novel virus restriction activity that adds PLSCR1 to a short list of host proteins known to directly interfere with the fusion process of pathogenic viruses, including SARS-CoV-2. However, the manuscript could be improved in the following ways.

Major:

1. The authors mostly take advantage of a recombinant SARS-CoV-2 expressing mNeonGreen to characterize the antiviral function of PLSCR1. It would be useful to assess the effect of PLSCR1 KO on wild type SARS-CoV-2 growth as measured by plaque assay.
2. The authors use E64d and camostat to inhibit cathepsins or TMPRSS2, respectively, to make the claim that PLSCR1 inhibits endocytic entry of SARS-CoV-2. However, interpretation of those experiments largely depends on the effects of those drugs on the cell lines tested. The authors should assess how ectopic expression of TMPRSS2 influences sensitivity of SARS-CoV-2 to PLSCR1, especially since other entry inhibitors like IFITM and NCOA7 have been shown to be less effective when TMPRSS2 is highly expressed. Can the authors be sure that PLSCR specifically antagonizes some aspect of type I fusion protein-mediated entry, rather than PLSCR being active against viruses undergoing fusion at particular membrane sites?
3. The promoter region of PLSCR1 should be described. Does it have an ISRE and/or GAS elements that can explain the degree to which it is upregulated by type I, type II, and type III IFNs?
4. Can the authors confirm that PLSCR1 still inhibits SARS-CoV-2 entry in STAT1 KO cells? This would

tease apart the direct antiviral activity of PLSCR1 and any potential indirect effects it has on other ISGs (in cells that have a competent RIG-I pathway).

5. Since the authors identify that the beta barrel of PLSCR1 seems to be important for its antiviral activity, can they do more to address whether mutations in the beta barrel that cause loss of antiviral function also influence its localization to virus-containing vesicles/endosomes? This would reinforce their conclusion that trafficking of PLSCR1 to virus-containing vesicles is truly essential for its antiviral function.

6. Figure 2E should be redrawn as column bar graphs, since it appears that data from only a single time point is shown (24 hours post infection). The "0" time point is not very useful here, and the "12" time point, which is represented by a tick on the X-axis, was not examined. Thus, the only data that can be commented on is the 24 h time point. The authors may need to do more here to prove that virus replication has occurred between 0 and 24 hours, and that this PCR assay adequately reflects that virus replication has occurred.

7. Since the authors state the possibility that local scramblase activity of PLSCR1 in virus-containing vesicles may be important for its restriction activity, the authors should isolate vesicles/endolysosomes from infected cells and check the status of PS there. Otherwise, readers will be left not knowing whether the enzymatic activity of PLSCR1 is required for this novel antiviral activity and knowing this would significantly raise the impact of the study. Also, the authors should discuss studies that have previously shown that calcium-driven PS exposure influences the entry/fusion of SARS-CoV-2 and other viruses, like Influenza A.

8. The authors don't do enough to put this study in the context of other known inhibitors of SARS-CoV-2 entry into cells. I would suggest referencing Majdoul et al. NRI, 2021 as it reviews the screens that helped identify a number of restriction factors against SARS-CoV-2 entry, including LY6E, IFITM, and CH25H. In addition, other factors that are known to influence cathepsin function (and thereby inhibit Spike-mediated entry) are discussed. I think a brief discussion of how these proteins are believed to inhibit the fusion process would be helpful, and it would give the authors another chance to discuss the possible mechanisms by which PLSCR1 inhibits fusion. For example, the authors show some evidence that the mechanisms of PLSCR1 and IFITM3 may be different, since IFITM3 strongly inhibits EBOV and modestly inhibits VSV, while PLSCR1 has little to no effect on EBOV and VSV. What can be said about the effect of PLSCR1 on membrane fluidity/curvature?

Minor:

1. Line 105: I don't agree that IFNs inhibited infection "in a PLSCR1-dependent manner" because some degree of sensitivity to IFNs is still observed in PLSCR1 KO cells. The fold inhibition by IFNs is reduced but inhibition is still apparent in PLSCR1 KO cells. This suggests that PLSCR1 contributes to protection mediated by IFNs but it is not the only factor that does so.

2. Line 128: Reference 30 might be inappropriately used here. It seems that reference 29 should be here instead (the one speaking about animal reservoirs of SARS-CoV-2).

3. Line 227: Should read "appears"

4. I don't think it is appropriate to refer to HeLa cells as "fibroblasts."

Signed: Alex Compton

Referee #2 (Remarks to the Author):

In this original study, the authors employed a CRISPR-cas9 screening strategy to probe the IFN-gamma-induced antiviral mechanism against SARS-CoV-2. These efforts identified PLSCR1 as an IFN induced protein active against SARS-CoV-2 and other coronavirus infections in a variety of different cell types and species. Interestingly, PLSCR1, which localizes to the plasma membrane, suppressed the endosome entry pathway controlling release of viral RNA into the cytoplasm. The results are convincing, significant and contribute to a better understanding of innate immune responses against SARS-CoV-2. However, there are several points that should be addressed with revised data analysis and/or presentation and text modifications:

1. IFN-gamma or IFN-alpha2a treatment of Huh7.5 cells seemed to completely inhibit SARS-CoV-2 infection (Fig. 1a). That graph should be in a log₁₀ scale instead of a linear “relative %” scale to better show how much protection the IFNs provided. This should be in pfu/ml instead of percent infected cells.

2. It is stated that Huh7.5 are defective for RIG-I signaling “thus ensuring antiviral activities were only elicited by exogenous IFNs”, however, dsRNA of SARS-CoV-2 and other coronaviruses signal through MDA5. This statement should be clarified, and the interpretation of these results modified.

3. PLSCR1 KO in both Huh7.5 and A549 cells resulted in higher levels of SARS-CoV-2 infection, in the presence or absence of IFN-gamma pre-treatment. In PLSCR1 KO cells, there was a 5- to 7-fold increase in infected cells, thus confirming an effect PLSCR1 against SARS-CoV-2. However, extending these results to concluding that hPLSCR1 “is obligate to combat SARS-CoV-2 infection” is an over interpretation for obvious reasons (e.g., no evidence was presented that PLSCR1 is the only ISG that inhibits SARS-CoV-2 infections). In fact, the results show lesser anti-SARS-CoV-2 effects of IFITM3 (extended data figure 5G). What these results do show is that PLSCR1 contributes to basal and IFN-gamma induced innate immunity against SARS-CoV-2.

4. Fig. 2c should be presented as a log₁₀ scale, not a linear scale. There seems to be a 10-fold inhibition by PLSCR1 over-expression in the absence of IFN-gamma (about 2x10⁵ pfu/ml to about 2x10⁴). What is the extent of the inhibition in the presence of IFN-gamma? You can't tell from the graph.

5. Some readers of this paper may be left with the impression that this is the first report that PLSCR1 is an antiviral ISG because prior literature on the subject was not mentioned or cited in the paper. The antiviral effects of PLSCR1 were first reported 18 years ago (PMID: 15308695). A range of different types of viruses are inhibited by PLSCR1, with different antiviral mechanisms. Including among PLSCR1-susceptible viruses are HBV, HIV, HTLV1, IAV, EBV, and HCMV through PLSCR1 interactions with viral proteins. A recent review on the subject provides background for these various antiviral studies on PLSCR1 [Col, J.D. et al. (2022); PMID: 35650588]. The prior literature should be cited and discussed in relation to the study. This paper is nonetheless significant because it presents a newly discovered antiviral molecular mechanism of PLSCR1, one that may be specific for coronaviruses.

Referee #3 (Remarks to the Author):

In the paper a "Human PLSCR1 is a cell-autonomous defense factor against SARS-CoV-2 infection" the authors did multiple CRISPR screens in parallel: Screens of SARS-CoV-2 infected A549 and Huh7.5 cells with and without IFN gamma stimulation. They found that IFN gamma stimulated PLSCR1 inhibited SARS-CoV-2 USA-WA1/2020, as well as Delta B.1.617.2 and Omicron BA.1 lineages via targeting virus-containing endolysosomal vesicles.

First it was shown that IFN gamma restricted SARS-CoV-2 in Huh7.5 cells in a STAT1-dependent manner.

As mentioned CRISPR screens were performed in parallel. GeCKO v2 library transfected +/- IFN treated cells were infected with SARS-CoV-2 mNeonGreen, to allow FACS sorting into permissive mNeonGreen high and non-permissive mNeonGreen low populations. These populations were then analyzed for sgRNA abundance by NGS. The mNG high group represents host defense factors, the mNG low group represent host dependency factors.

As expected the KO of key IFN gamma pathway genes or ISGs made cells more susceptible in the IFN gamma stimulated condition but did not have an effect in the non-stimulated condition. PLSCR1-KO significantly enhanced SARS-CoV-2 infection in +/- IFN treated cells in Huh7.5 and A549 cells. The authors thus conclude that PLSCR1 is obligate to combat SARS-CoV-2 USA-WA1/2020, Delta B.1.617.2 and Omicron B.1.1.529 infection. The antiviral effect of PLSCR1 was rescued by genetic complementation. It was also rescued by cross species complementation with horseshoe bat or mouse PLSCR1. Then they also show that PLSCR1-KO is also proviral in Let1, Calu-3, Tonsil-ACE2, HaCaT-ACE2 and Hela-ACE2 cell lines.

Using a pseudovirus system it was shown that PLSCR1 expression blocked MERS-CoV, SARS-CoV(-2, VOC) and BatCoV-WIV-1 spike-mediated entry. Endemic CoV spike-mediated entry was inhibited to a lesser extend. Thus they conclude that PLSCR1 inhibits the type I fusion mechanism.

To gain mechanistic insight, the authors specifically inhibit cathepsin-dependent and TMPRSS2-dependent virus entry using E-64d and camostat in control and PLSCR1-KO cells. PLSCR1-KO cells showed increased entry, whereas E-64d treatment reversed that phenotype and blocked entry. Camostat treatment in PLSCR1-KO did not alter viral entry. This shows that PLSCR1 plays a role in endosomal fusion pathway. They moreover show with a split luciferase assay that PLSCR1 interferes with Sars-CoV-2 infection after receptor-mediated internalization.

Further, stainings of PLSCR1, as well as viral proteins and microscopy analysis reveal that PLSCR1 localizes to SARS-CoV-2 vesicles and inhibits viral RNA release. The authors also assessed which region or domain of PLSCR1 may crucial for the observed phenotype and identified that deletion of a beta strand and beta barrel (aa 86-118) or a C-terminal hydrophobic helix (aa 291-318) to be detrimental for PLSCR1 antiviral activity. Notably, His262 which has been identified to be associated to severe COVID19 (in case of His/Tyr change) is close to these regions. A corresponding His262Tyr

PLSCR1 mutant were showed impaired anti-SARSR-CoV-2 activity.

Overall, this is an interesting work and identifies PLSCR1 as an antiviral factor for SARS-CoV-2. The mechanistic work in the second part of the paper is well done. However, there're major concerns about the first part of the study and about novelty when it comes to a role of PLSCR1 during virus infections.

Specific concerns:

major:

1. a very general but major concern is that a role of PLSCR1 for virus replication is not novel. PLSCR1 has been studied already in HCV and herpesvirus infections (and others). Although it is acknowledged that the current manuscript goes very deep into mechanistic studies on entry, it is disturbing that neither in the introduction nor in the discussion anything is mentioned about previous work on PLSCR1 in the context of virus infection.

2. The actual impact of the antiviral activity of PLSCR1 appears to be overstated throughout all experiments that deal with measurements of viral replication. This includes experiments shown in Figure 2 -4. The authors show a linear scale that should actually be logarithmic. For example if the linear scale is shown and indicated that values have to multiplied by 10E6 the differences look much greater than they actually are (e.g. 2x10E6 compared to 4x10E6 is actually not very much). Therefore, graphs have to be changed to a log scale. A similar concern is the measurement of "relative viral RNA level; normalised to NC 1h". It would be much clearer to determine and show genome equivalents. Most likely the criticism raised above would also apply since differences may not look so great anymore.

3. The claim that PLSCR1 may have antiviral activity against viruses that enter cell via type-I fusion proteins is not fully supported by the provided data. A much wider panel of viruses should be used. Notably, the use of HCV in this context would require to mention and discuss previous work on PLSCR1/HCV (see comment point 1)

4. The authors show that mainly, or even exclusively, the endosomal route of entry is targeted by PLSCR1. This raises questions about the overall impact of this mechanism, since inhibition of the endosomal entry pathway by antiviral drugs have not been very successful in in vivo experiments and in humans.

minor:

please explain the initial screen better. In particular figure 1e. It looks like PLSCR1 is appearing as one the top hits in Huh7 cell with or without IFNg treatment, while it appears as top hit in A549-ACE2 cells only under conditions of IFNg treatment.

Author Rebuttals to Initial Comments:

Referees' comments:

Referee #1 (Remarks to the Author):

In this submission, Xu et al. describe the antiviral properties of the lipid scramblase PLSCR1, particularly in the context of SARS-CoV-2 infection. PLSCR1 was a prioritized hit from a whole-genome CRISPR/Cas9 screen performed in hepatocytes and lung epithelial cells, and performs constitutive antiviral function as is further upregulated by IFN-gamma (as well as type I and type III IFNs). Through a lot of comprehensive experiments that rely heavily on confocal immunofluorescence microscopy, the authors propose that PLSCR1 inhibits entry of SARS-CoV-2 and other coronaviruses that use type I fusion machinery. The authors make a solid case that PLSCR1 localization to virus-containing vesicles and endolysosomes is an essential component of its antiviral activity, but what it does exactly to prevent virus fusion remains unclear. PLSCR1 does not seem to influence endolysosomal acidification or cathepsin activity. Therefore, it is tempting to speculate that the influence of PLSCR1 on the externalization of lipids (like phosphatidylserine) may disfavor virus-cell membrane fusion in endolysosomes. Given that several lipid scramblases have been described recently to influence the function of virus fusion machinery (TMEM41B, TMEM16F, and others), this is an important contribution to a budding field in viral immunology. Overall, this is very interesting description of a novel virus restriction activity that adds PLSCR1 to a short list of host proteins known to directly interfere with the fusion process of pathogenic viruses, including SARS-CoV-2. However, the manuscript could be improved in the following ways.

Major:

1. The authors mostly take advantage of a recombinant SARS-CoV-2 expressing mNeonGreen to characterize the antiviral function of PLSCR1. It would be useful to assess the effect of PLSCR1 KO on wild type SARS-CoV-2 growth as measured by plaque assay.

We had performed plaque assays to validate the effect of PLSCR1 KO on wild-type SARS-CoV-2 infection in the original manuscript. As seen with SARS-CoV-2-mNG, knocking out PLSCR1 led to a significant increase in the production of infectious SARS-CoV-2 virions. This antiviral defect was fully restored by reintroducing PLSCR1 cDNA into the KO cells. Please refer to the data in Manuscript Fig. 2c.

2. The authors use E64d and camostat to inhibit cathepsins or TMPRSS2, respectively, to make the claim that PLSCR1 inhibits endocytic entry of SARS-CoV-2. However, interpretation of those experiments largely depends on the effects of those drugs on the cell lines tested. The authors should assess how ectopic expression of TMPRSS2 influences sensitivity of SARS-CoV-2 to PLSCR1, especially since other entry inhibitors like IFITM and NCOA7 have been shown to be less effective when TMPRSS2 is highly

expressed. Can the authors be sure that PLSCR1 specifically antagonizes some aspect of type I fusion protein-mediated entry, rather than PLSCR1 being active against viruses undergoing fusion at particular membrane sites?

Thanks for this thoughtful comment.

We have found ectopic expression of TMPRSS2 can influence the antiviral effect of PLSCR1 on SARS-CoV-2 although it appears less pronounced than for the endosomal pathway. Consistent with previous reports, ectopic expression of TMPRSS2 in Huh7.5 cells (which are TMPRSS2-negative) dramatically enhanced virus-plasma membrane fusion (increased formation of syncytia) and increased the infectivity of SARS-CoV-2 (inserted Fig.1 and Manuscript Extended Data Fig. 4h). When we overexpressed PLSCR1 in wild-type Huh7.5 cells, we observed a 3.7-fold inhibition in SARS-CoV-2 infection. The antiviral effects of PLSCR1 were slightly attenuated (dropped from 3.7-fold to 3.0-fold) in Huh7.5 cells ectopically expressing TMPRSS2.

Inserted Fig.1. Quantification of SARS-CoV-2 infection in E-64d (20 μ M) treated or untreated Huh7.5 cells overexpressing vector or PLSCR1 with or without ectopic expression of TMPRSS2 (MOI=1, 48 h p.i.).

TMPRSS2 overexpression has been reported to attenuate the anti-SARS-CoV-2 effects of NCOA7 and IFITM3, which stems from bypassing NCOA7 and IFITM3-mediated restriction of endosomal virus entry (Shi et al, EMBO J, 2021; Khan et al, PLOS Pathogens, 2021). For PLSCR1, however, the attenuation by TMPRSS2 over-expression was milder, suggesting PLSCR1 can also antagonize the cell-surface entry pathway in addition to its restriction of the endosomal pathway. This is because PLSCR1 associates with the plasma membrane under basal conditions (Manuscript Fig. 4c and Extended Data Fig. 6a,b). As infection proceeds, PLSCR1 becomes enriched in endolysosomes harboring SARS-CoV-2 which probably result in higher local concentrations that exert more effective antiviral control for this pathway at later times.

This interpretation is supported using E-64d treatment to decouple the two viral entry pathways. E-64d strongly inhibited SARS-CoV-2 infection in WT Huh7.5 cells naturally lacking TMPRSS2 which validates viral entry relies on the endosomal pathway in these

cells. Here PLSCR1 restricts viral entry by the endosomal pathway. In TMPRSS2-expressing Huh7.5 cells, however, E-64d had little effect, indicating that SARS-CoV-2 uses the cell surface pathway under these conditions. PLSCR1 also showed some antiviral activity in TMPRSS2-expressing cells, with or without E-64d (Inserted Fig.1). Thus, PLSCR1 impacts both viral entry pathways.

To further validate these observations, we tested the antiviral activity of PLSCR1 in Calu-3 cells. Calu-3 cells predominantly support cell-surface fusion due to their high endogenous expression of TMPRSS2. We generated *PLSCR1* KO Calu-3 cells which significantly increased the overall susceptibility of these cells (5.5-fold increase in susceptibility). Using a saturating dose of E-64d to block the endosomal pathway and force entry only via the TMPRSS2 pathway, that susceptibility was less pronounced but still detectable (dropped to 3-fold). Conversely, adding camostat to block the TMPRSS2 pathway and force entry only via the endosomal pathway led to a more pronounced susceptibility (increased to 14-fold) (Inserted Fig.2 and Manuscript Extended Data Fig. 4i).

Inserted Fig.2. Left: Dose response of indicated compounds on SARS-CoV-2 infection in Calu-3 (MOI=1, 24h p.i., n=4). The amount of viral RNA in DMSO group was normalized to 1. Right: Quantification of SARS-CoV-2 infection in Control or *PLSCR1*-KO Calu-3 (MOI=1, 24h p.i., n=4) treated with indicated compounds (E64d: 20 μM, Camostat: 20 μM). Cells were treated with indicated compounds 2h before infection.

Together these results show that PLSCR1 can inhibit SARS-CoV-2 through both the endosomal and cell-surface entry pathways while being more effective in the former. Importantly, PLSCR1 has the capacity to protect multiple cell types regardless of how SARS-CoV-2 infection is initiated. This may be due to the strong enrichment of plasma membrane-resident PLSCR1 and the localization to endosomes upon engaging engulfed viruses (Fig 4a, Extended Data Fig. 6b and Extended Data Fig. 7a). Whether this operates at certain membrane sites or microdomains is currently unknown although we do know palmitoylated PLSCR1 produced in human cells strongly binds phosphatidylinositides (PIPs) normally present in both the PM and endosomes (data not shown). Whether this

binding has functional consequences is a question for future work. Regarding viruses that use type I fusion, it appears PLSCR1 is particularly effective in restricting coronaviruses, so it may be influenced by host factors such as tetraspannins (eg. CD9) used as accessory proteins for entry by this group rather than the class I fusion mechanism *per se*. We have now rewritten this section to remove any emphasis on the type of fusion class employed by different viruses. Thanks for bringing it to our attention.

3. The promoter region of PLSCR1 should be described. Does it have an ISRE and/or GAS elements that can explain the degree to which it is upregulated by type I, type II, and type III IFNs?

Following your suggestion, we have analyzed the promoter region of the human *PLSCR1* locus. Two GAS elements and two ISRE elements exist within 2Kb upstream of the transcription initiation site, offering a clear explanation for why *PLSCR1* is upregulated by type I, type II, and type III IFNs (**Manuscript Extended Data Fig. 1f**).

Inserted Fig.3. The presence and position of GAS (Gamma interferon activation site) and ISRE (Interferon-sensitive response element) elements within the promoter region (2 Kb upstream from the transcription initiation site) of the human *PLSCR1* gene.

4. Can the authors confirm that *PLSCR1* still inhibits SARS-CoV-2 entry in *STAT1* KO cells? This would tease apart the direct antiviral activity of *PLSCR1* and any potential indirect effects it has on other ISGs (in cells that have a competent RIG-I pathway).

We confirmed that *PLSCR1* still restricts SARS-CoV-2 entry in *STAT1* KO Huh7.5 cells (**Inserted Fig. 4; Manuscript Fig. 2e and Extended Data Fig. 2c**). When we ectopically expressed *PLSCR1* in a *STAT1*-KO background, we observed significant restriction of SARS-CoV-2 infection comparable with *PLSCR1* overexpression in non-targeting control (NC) cells. This result suggests that the anti-SARS-CoV-2 activity of *PLSCR1* does not require other ISGs to confer its function nor do other ISGs require *PLSCR1* for their activity. It reinforces the direct antiviral action of *PLSCR1* towards SARS-CoV-2.

Inserted Fig.4. Left: Quantification of SARS-CoV-2-mNG infection in control or STAT1-KO Huh7.5 cells overexpressing PLSCR1 in the presence or absence of IFN- γ . S1KO: STAT11-KO. (n=5) Right: Western blot showing the expression level of indicated proteins.

In parallel, we generated STAT1 and PLSCR1 double mutant KO Huh7.5 cells (Inserted Fig.5 and Manuscript Extended Data Fig. 2d,e). Under basal conditions, PLSCR1 KO versus PLSCR1/STAT1 double KO led to similar levels of increase in SARS-CoV-2 infection, suggesting that PLSCR1 is a primary restriction factor under basal conditions. Under IFN- γ -stimulated conditions, PLSCR1/STAT1 double KO led to a further increase in SARS-CoV-2 infection compared with the loss of either PLSCR1 or STAT1 alone, indicating that other ISGs can act alongside PLSCR1 under IFN- γ -treated conditions.

Inserted Fig.5. Left: Quantification of SARS-CoV-2 infection in Huh7.5 cells of indicated genotypes in the presence or absence of IFN- γ (10 U.ml⁻¹) (MOI=1, 48h p.i.). (n=6) Right: Western blot showing the expression level of indicated proteins.

5. Since the authors identify that the beta barrel of PLSCR1 seems to be important for its antiviral activity, can they do more to address whether mutations in the beta barrel that cause loss of antiviral function also influence its localization to virus-containing

vesicles/endosomes? This would reinforce their conclusion that trafficking of PLSCR1 to virus-containing vesicles is truly essential for its antiviral function.

Thanks for the constructive advice since it is critical to establish if PLSCR1 localization to virus-containing vesicles is necessary for its antiviral function. Initially we mutated all the residues in the PLSCR1 palmitoylation site (5 Cys to Ala) which abolished membrane association in accordance with earlier work (Wiedmer et al, Biochemistry, 2003). The 5CA mutation completely ablated PLSCR1's ability to target to SARS-CoV-2-containing vesicles (Inserted Fig. 6 and Manuscript Fig. 4d). It also completely ablated restriction of SARS-CoV-2 infection, unlike WT PLSCR1. Because the 5CA mutation falls outside the β -barrel, loss of antiviral activity is not due to disruption of the barrel structure itself, but rather to the loss of membrane anchorage to virus-containing vesicles or endosomes through palmitoylation. Localization is therefore essential for PLSCR1 to confer its antiviral function (Inserted Fig.7 and Manuscript Fig. 4e).

Inserted Fig.6. Comparison of the localization of PLSCR1 on SARS-CoV-2 containing vesicles between WT and 5CA (C184CCPCC189 to AAAPAA) mutant. A549-ACE2 PLSCR1-KO cells stably expressing GFP-PLSCR1-WT or GFP-PLSCR1-5CA were infected with SARS-CoV-2 for 2h (MOI=25). (WT: 24 cells. 5CA: 28 cells). Scale bar: 10 μ m, inlays: 5 μ m.

Inserted Fig.7. Quantification of SARS-CoV-2 infection in Huh7.5 cells expressing indicated mutants (MOI=1, 48 h p.i.). PKO: PLSCR1-KO (n=5).

We also undertook the opposite experiment, examining whether loss-of-function mutations in the β -barrel itself affect the targeting of PLSCR1 to virus-containing vesicles. PLSCR1 localization was unaffected by F281A or H262Y mutations, indicating that these

mutations affect downstream events such as conformation changes and/or functional stability of PLSCR1 once it has assembled on the membrane (Inserted Fig. 8 and Extended Data Fig. 8i).

Inserted Fig.8. Representative images showing the localization of GFP-PLSCR1 F281A and H262Y mutants on SARS-CoV-2 containing vesicles in A549-ACE2 PLSCR1-KO cells infected with SARS-CoV-2 for 2h (MOI=25). Scale bar: 10 μ m, inlays: 5 μ m.

Lastly, new mutagenesis experiments along with molecular dynamic simulation have further highlighted these regions of the PLSCR1 protein as being important for its anti-SARS-CoV-2 function (Manuscript Fig. 4f-i, Extended Data Fig. 8e-i, Fig. 9a,c-e and Fig. 10f).

Overall, PLSCR1 targeting to virus-containing vesicles appears essential for anti-SARS-CoV-2 activity. Specific regions within the beta-barrel also contribute to antiviral activity after PLSCR1 has docked to the membrane.

6. Figure 2E should be redrawn as column bar graphs, since it appears that data from only a single time point is shown (24 hours post infection). The “0” time point is not very useful here, and the “12” time point, which is represented by a tick on the X-axis, was not examined. Thus, the only data that can be commented on is the 24 h time point. The authors may need to do more here to prove that virus replication has occurred between 0 and 24 hours, and that this PCR assay adequately reflects that virus replication has occurred.

We have revised all figures as requested. The original Fig. 2E has now been redrawn as column bar graphs that reflect SARS-CoV-2 viral RNA levels at the 24 hpi time point, (Manuscript Fig. 2f). Relative viral RNA levels have also been converted to absolute values expressed in SARS-CoV-2 genome equivalents per well, and the y-axis has been changed to a Log₁₀ scale at the request of other reviewers. In addition, an 8-hour time point between the 0 and 24-hour time points is added to the figure below to show that virus replication has indeed occurred (Inserted Fig.9).

Inserted Fig.9. Quantification of intracellular SARS-CoV-2 RNA level in Huh7.5. Primers detecting Nucleocapsid were used to amplify viral RNA. GE: Genome equivalents. Cells were infected with SARS-CoV-2 USA-WA1 strain at a MOI of 0.5 and harvested at 1 h, 8h and 24h post infection. (n=4)

7. Since the authors state the possibility that local scramblase activity of PLSCR1 in virus-containing vesicles may be important for its restriction activity, the authors should isolate vesicles/endolysosomes from infected cells and check the status of PS there. Otherwise, readers will be left not knowing whether the enzymatic activity of PLSCR1 is required for this novel antiviral activity and knowing this would significantly raise the impact of the study. Also, the authors should discuss studies that have previously shown that calcium-driven PS exposure influences the entry/fusion of SARS-CoV-2 and other viruses, like Influenza A.

Thanks for the suggestion. Unfortunately, we are restricted from purifying SARS-CoV-2-infected endosomes due to the homogenization process producing large quantities of infectious aerosol particles that may be hazardous to others sharing our BL3 facility space at Yale. We therefore devised alternative experiments to address the question of whether the enzymatic activity of PLSCR1 is required for this novel antiviral activity.

PLSCR1 was originally identified as a Ca²⁺-activated phospholipid scramblase (Basse et al, JBC, 1996; Zhou et al, JBC, 1997). However, this putative function remains controversial. Blood cells isolated from *Plscr1*^{-/-} mice are still able to mobilize phosphatidylserine (PS) to the outer leaflet upon Ca²⁺ stimulation (Zhou et al, Blood, 2002) and increasing the amount of PLSCR1 (via interferon-alpha induction) does not increase PS externalization (Zhou et al, Blood, 2000). We likewise found Ca²⁺-activated PS externalization was unaltered in *PLSCR1*-KO versus wild-type A549-ACE2 cells (Manuscript Fig. 4i and Extended Data Fig. 10a).

This result may be due to compensation by other Ca²⁺-activated lipid scramblases which, if removed, could allow assessment of PLSCR1 scramblase activity *in situ*. Such an assay would also enable PLSCR1 mutants lacking anti-SARS-CoV-2 activity to be tested for scramblase activity, thereby establishing a shared functional relationship, or uncoupling these two activities. Measuring plasma membrane scramblase activity was relevant

because the PM directly gives rise to SARS-CoV-2-containing vesicles in the endosomal entry pathway of ACE2-expressing A549 cells (Manuscript Fig. 4c and Supplementary Video 3 ,4).

With this approach, we tested whether TMEM16F, a potent Ca^{2+} -activated scramblase that operates at the PM and is required for SARS-CoV-2 entry (Suzuki et al, Nature, 2010; Braga et al., Nature 2021; Sim et al., Cell Reports 2022), could potentially mask the *PLSCR1*-KO phenotype because it has stronger scramblase activity.

Stable deletion of TMEM16F in A549-ACE2 cells greatly diminished Ca^{2+} -activated PS externalization, in contrast to *PLSCR1*-KO cells which resembled A549-ACE2 controls. Thus, TMEM16F is a dominant PS scramblase in these cells with the potential to mask any weaker scramblase activity of PLSCR1 (Inserted Fig. 10, Manuscript Fig. 4i and Extended Data Fig. 10a,b). Removing TMEM16F could therefore provide a sufficiently sensitive background to test PLSCR1 and its loss-of-function antiviral mutants.

Inserted Fig.10. Left: Schema depicting the phosphatidylserine (PS) externalization assay. Externalized PS was detected by membrane impermeable Annexin V conjugated with Alexa647. Middle: FACS plots showing the level of PS externalization in NC, *PLSCR1*-KO or *TMEM16F*-KO A549-ACE2 cells in the absence or presence of 10 μM ionomycin. Ionomycin acts as a membrane permeable Ca^{2+} carrier which elevates intracellular Ca^{2+} level and triggers PS externalization. Right: Western blot showing the protein expression level

We therefore stably over-expressed these proteins in *TMEM16F*-KO cells. Wild-type *PLSCR1* partially restored the deficiency in PS externalization caused by *TMEM16F* deficiency, indicating that *PLSCR1* does indeed have weak scramblase activity on the plasma membrane (Inserted Fig.11 and Manuscript Fig. 4j).

Each of the mutants (F281A, 5CA, H262Y) expressed at the same levels in *TMEM16F*-KO cells enabled us to test if scramblase activity coincides with the antiviral function of *PLSCR1*. Consistent with a previous study showing that F281A within the Ca^{2+} -binding EF-hand motif inactivates *PLSCR1* scramblase activity (Zhou et al, Biochemistry, 1998), we found the F281A mutant lost its ability to enhance PS externalization. Unsurprisingly, membrane association was also obligate for scramblase activity, as the *PLSCR1* 5CA mutation blocking palmitoylation also showed no PS externalization. Unexpectedly,

however, the COVID-19-associated H262Y mutation (which significantly impaired PLSCR1 antiviral activity), showed wild-type levels of PS externalization (Inserted Fig.11 and Manuscript Fig. 4j). Thus, the weak PS scramblase activity of PLSCR1 can be uncoupled from its novel antiviral activity.

Inserted Fig.11. Left: FACS plots showing the level of PS externalization in NC or TMEM16F-KO A549-ACE2 cells stably overexpressing indicated PLSCR1 mutants in the absence or presence of 10 μ M ionomycin. Right: Relative quantification of PS externalization activity. PS exposure in NC A549-ACE2 cells treated with ionomycin was normalized to 1. (n=3)

This result is reinforced by the fact that other Ca^{2+} -dependent lipid scramblases like TMEM16F and TMEM41B promote rather than inhibit SARS-CoV-2 entry and the formation of virus-containing vesicles (Braga et al, Nature, 2021; Hoffman et al., Cell 2021; Sim et al, Cell Reports, 2022). Indeed, because the scramblase activity of PLSCR1 is so much weaker than that of TMEM16F on the PM, it seems unlikely that PLSCR1 could antagonize TMEM16F by altering the net direction of PS to block viral entry.

We therefore conclude that the antiviral function of PLSCR1 is essentially independent of its scramblase activity. This is a novel finding and helps redirect the field.

8. The authors don't do enough to put this study in the context of other known inhibitors of SARS-CoV-2 entry into cells. I would suggest referencing Majdoul et al. NRI, 2021 as it reviews the screens that helped identify a number of restriction factors against SARS-CoV-2 entry, including LY6E, IFITM, and CH25H. In addition, other factors that are known to influence cathepsin function (and thereby inhibit Spike-mediated entry) are discussed. I think a brief discussion of how these proteins are believed to inhibit the fusion process would be helpful, and it would give the authors another chance to discuss the possible mechanisms by which PLSCR1 inhibits fusion. For example, the authors show some

evidence that the mechanisms of PLSCR1 and IFITM3 may be different, since IFITM3 strongly inhibits EBOV and modestly inhibits VSV, while PLSCR1 has little to no effect on EBOV and VSV. What can be said about the effect of PLSCR1 on membrane fluidity/curvature?

As per your suggestion, we provide a summary of the proposed mechanisms by which previously reported restriction factors may inhibit the entry of SARS-CoV-2 in the discussion of our revised manuscript. Thanks for the input. They are also outlined in more detail below as part of the reviewer response because in the manuscript we must conform with word limits:

- LY6E was shown to inhibit syncytia formation as well as spike protein-mediated cell-cell fusion but did not perturb the proteolytic processing of the spike protein, which means that LY6E likely interferes with the fusion process itself and not spike protein cleavage (Pfaender et al, Nat Microbiol, 2020; Zhao et al, J Virol, 2020; Majdoul et al, Nat Microbiol, 2020).
- IFITM3 has been reported to be both pro- and antiviral in different contexts of SARS-CoV-2 infection (Shi et al, EMBO J, 2021; Majdoul et al, Nat Microbiol, 2020). Overexpression of the serine protease TMRPSS2, which promotes SARS-CoV-2 cell-surface fusogenic entry rather than endosomal entry, attenuated the antiviral effects of IFITM3 and interestingly converted it to an enhancer of SARS-CoV-2 infection. Meanwhile, IFITM3 restricted SARS-CoV-2 infection during endosomal entry, and this inhibitory effect is likely mediated by its amphipathic helix, which is proposed to increase local membrane rigidity and curvature to block viral fusion (Shi et al, EMBO J, 2021; Guo et al, ACS Nano, 2021; Majdoul et al, Nat Microbiol, 2020).
- CH25H catalyzes an enzymatic product 25HC to regulate membrane cholesterol content in different cellular compartments (Abrams et al, Nat Microbiol, 2020; Majdoul et al, Nat Microbiol, 2020). CH25H and 25HC were demonstrated to inhibit SARS-CoV-2 spike-mediated fusion, likely by interfering with cholesterol export, since addition of soluble cholesterol rescued SARS-CoV-2 spike-mediated entry in cells that were pre-treated with 25HC (Zang et al, PNAS, 2020; Zu, Cell Research, 2020; Majdoul et al, Nat Microbiol, 2020).
- CIITA (MHC class II transactivator) can activate the expression of the p41 isoform of CD74, which was reported to alter cathepsin-mediated processing of the SARS-CoV-2 spike protein to block viral entry (Bruchez et al, Science, 2020; Majdoul et al, Nat Microbiol, 2020).
- The short interferon-inducible isoform of NCOA7 was reported to inhibit SARS-CoV-2 entry (Khan et al, PLoS Pathogens, 2021; Majdoul et al, Nat Microbiol, 2020). Based on previous findings that NCOA7 can promote elevated cathepsin activity in endolysosomes (Doyle et al, Nat Microbiol, 2018), NCOA7 may prevent viral entry by enhancing cathepsin-mediated proteolysis to promote degradation of virion particles in endolysosomes before fusogenic entry occurs.

The mechanism by which PLSCR1 restricts SARS-CoV-2 differs from most of the restriction factors above. It does not interfere with cathepsin-mediated cleavage of the spike protein or endolysosomal proteolysis (CIITA/CD74, NCOA7), nor does it alter membrane rigidity (IFITM3). It does not appear to use its weak PS scramblase activity for blocking viral-endosome or viral-cell surface fusion, since mutations uncouple these activities. We do know that within the endosomal pathway, PLSCR1 operates after spike proteolysis but before virus-endosome fusion takes place to allow SARS-CoV-2 ssRNA to be deposited within the host cytosol to initiate translation and formation of dsRNA replication intermediates. In this regard, PLSCR1 acts at a similar stage to LY6E but appears independent since deficiencies in both loci (PLSCR1/LY6E DKO mutants) exceeded the defects for either single KO mutant alone (Manuscript Extended Data Figure 3h). This would point to separate activities rather than belonging to the same convergent linear pathway; in the latter case the magnitude of the phenotype would remain unaltered. Furthermore, the β -barrel domain used by PLSCR1 to disrupt viral-membrane fusion is structurally very different from the GPI-anchored LY6E protein, reinforcing that they likely enlist different molecular mechanisms.

To test if PLSCR1 had any effects on intrinsic membrane properties such as bending rigidity like IFITM3, we isolated giant plasma membrane vesicles (GPMVs) from PLSCR1-KO A549-ACE2 cells stably expressing either EGFP-PLSCR1 WT or the 5CA mutant. These were tested in a membrane tether pulling assay (Manuscript Extended Data Fig. 10f-g). The 5CA mutant served as a negative control, since it cannot not bind the GPMV membrane as shown in Manuscript Extended Data Fig. 10g. We found no significant differences between these two groups, indicating that the inhibition of membrane fusion by PLSCR1 does not rely on altering the membrane bending rigidity, a mechanism partially attributed to IFITM3 function (Guo et al., ACS Nano, 2021). It is also unlikely that membrane curvature plays a prominent role in how PLSCR1 inhibits beta-coronavirus fusion, because PLSCR1 was effective in inhibiting fusion between both high (virus-endosome) and low curvature membranes (cell-cell fusion).

Minor:

1. Line 105: I don't agree that IFNs inhibited infection "in a PLSCR1-dependent manner" because some degree of sensitivity to IFNs is still observed in PLSCR1 KO cells. The fold inhibition by IFNs is reduced but inhibition is still apparent in PLSCR1 KO cells. This suggests that PLSCR1 contributes to protection mediated by IFNs but it is not the only factor that does so.

Agreed. This sentence has been modified to simply state that PLSCR1 is required for the activity of type I and III IFNs against SARS-CoV-2.

2. Line 128: Reference 30 might be inappropriately used here. It seems that reference 29 should be here instead (the one speaking about animal reservoirs of SARS-CoV-2).

Corrected.

3. Line 227: Should read “appears”

Corrected. Thanks again for bringing these typographical errors to our attention.

4. I don't think it is appropriate to refer to HeLa cells as “fibroblasts.”

Our apologies for the confusion. We corrected it to “cervical epithelial cells”.

Signed: Alex Compton

Referee #2 (Remarks to the Author):

In this original study, the authors employed a CRISPR-cas9 screening strategy to probe the IFN-gamma-induced antiviral mechanism against SARS-CoV-2. These efforts identified PLSCR1 as an IFN induced protein active against SARS-CoV-2 and other coronavirus infections in a variety of different cell types and species. Interestingly, PLSCR1, which localizes to the plasma membrane, suppressed the endosome entry pathway controlling release of viral RNA into the cytoplasm. The results are convincing, significant and contribute to a better understanding of innate immune responses against SARS-CoV-2. However, there are several points that should be addressed with revised data analysis and/or presentation and text modifications:

1. IFN-gamma or IFN-alpha2a treatment of Huh7.5 cells seemed to completely inhibit SARS-CoV-2 infection (Fig. 1a). That graph should be in a log₁₀ scale instead of a linear “relative %” scale to better show how much protection the IFNs provided. This should be in pfu/ml instead of percent infected cells.

Thank you for the suggestion. Figure 1a has now been updated to reflect a log₁₀ scale and plaque assays were performed to convert SARS-CoV-2 titers to p.f.u./mL. Our results show that IFN-alpha2a and IFN-gamma have comparable antiviral effects against SARS-CoV-2, with their respective IC₅₀ in the same range (7.138 pM vs 3.245 pM). Please see inserted Fig.12 and Manuscript Fig. 1a) below.

Inserted Fig.12. Huh7.5 cells were treated with different concentrations of interferons (IFNs) and then infected with SARS-CoV-2 (isolate USA-WA1/2020) at a MOI of 1. Virus production (p.f.u./mL) was quantified by plaque assay 2 days post-infection (p.i.). (n=3).

2. It is stated that Huh7.5 are defective for RIG-I signaling “thus ensuring antiviral activities were only elicited by exogenous IFNs”, however, dsRNA of SARS-CoV-2 and other coronaviruses signal through MDA5. This statement should be clarified, and the interpretation of these results modified.

Our apologies for the confusion. It was based on a previous study showing that defective RIG-I signaling in Huh7.5 cells failed to induce type I IFNs when infected with RNA viruses like HCV, SenV or VSV (Sumpter et al, J Virol, 2005). However, the above study lacks direct evidence to extend this conclusion to RNA beta-coronaviruses like SARS-CoV-2 in Huh7.5 cells so we have removed it from the text. Which RNA sensor dominates will depend on the type of virus used and the type of host cell being examined. For example, MDA5 appears essential for sensing SARS-CoV-2 in Calu-3 cells (Yin et al, Cell Reports, 2021). Hence we thank Reviewer #2 for bringing this clarification to our attention.

3. PLSCR1 KO in both Huh7.5 and A549 cells resulted in higher levels of SARS-CoV-2 infection, in the presence or absence of IFN-gamma pre-treatment. In PLSCR1 KO cells, there was a 5- to 7-fold increase in infected cells, thus confirming an effect PLSCR1 against SARS-CoV-2. However, extending these results to concluding that hPLSCR1 “is obligate to combat SARS-CoV-2 infection” is an over interpretation for obvious reasons (e.g., no evidence was presented that PLSCR1 is the only ISG that inhibits SARS-CoV-2 infections). In fact, the results show lesser anti-SARS-CoV-2 effects of IFITM3 (extended data figure 5G). What these results do show is that PLSCR1 contributes to basal and IFN-gamma induced innate immunity against SARS-CoV-2.

Agreed. We revised the statement and reaffirmed its basal contributions throughout as well. We have also included new data showing comparisons with other ISGs that confirm PLSCR1 is indeed an important new anti-SARS-CoV-2 restriction factor.

4. Fig.2c should be presented as a log₁₀ scale, not a linear scale. There seems to be a 10-fold inhibition by PLSCR1 over-expression in the absence of IFN-gamma (about 2x10⁵ pfu/ml to about 2x10⁴). What is the extent of the inhibition in the presence of IFN-gamma? You can't tell from the graph.

Figure 2c has now been updated to present the data on a log₁₀ scale. Under IFN-gamma-stimulated conditions, there was also ~10-fold reduction (from 5.1x10⁴ to 4x10³ pfu/mL) in SARS-CoV-2 infection after PLSCR1-deficient Huh7.5 cells were genetically complemented with wild-type PLSCR1. Hence PLSCR1 confers anti-SARS-CoV-2 activity under both basal and cytokine-stimulated conditions.

5. Some readers of this paper may be left with the impression that this is the first report that PLSCR1 is an antiviral ISG because prior literature on the subject was not mentioned or cited in the paper. The antiviral effects of PLSCR1 were first reported 18 years ago (PMID: 15308695). A range of different types of viruses are inhibited by PLSCR1, with different antiviral mechanisms. Including among PLSCR1-susceptible viruses are HBV, HIV, HTLV1, IAV, EBV, and HCMV through PLSCR1 interactions with viral proteins. A recent review on the subject provides background for these various antiviral studies on PLSCR1 [Col, J.D. et al. (2022); PMID: 35650588]. The prior literature should be cited and discussed in relation to the study. This paper is nonetheless significant because it presents a newly discovered antiviral molecular mechanism of PLSCR1, one that may be specific for coronaviruses.

Thank you for the constructive feedback. We apologize for not elaborating on prior PLSCR1 studies in the original version since the manuscript was principally focused on SARS-CoV-2. However, we have now introduced these findings within the discussion section of the revised manuscript but had to keep the details brief to remain under the word limit.

Referee #3 (Remarks to the Author):

In the paper a "Human PLSCR1 is a cell-autonomous defense factor against SARS-CoV-2 infection" the authors did multiple CRISPR screens in parallel: Screens of SARS-CoV-2 infected A549 and Huh7.5 cells with and without IFN gamma stimulation. They found that IFN gamma stimulated PLSCR1 inhibited SARS-CoV-2 USA-WA1/2020, as well as Delta B.1.617.2 and Omicron BA.1 lineages via targeting virus-containing endolysosomal vesicles.

First it was shown that IFN gamma restricted SARS-CoV-2 in Huh7.5 cells in a STAT1-dependent manner.

As mentioned CRISPR screens were performed in parallel. GeCKO v2 library transfected

+/- IFN treated cells were infected with SARS-CoV-2 mNeonGreen, to allow FACS sorting into permissive mNeonGreen high and non-permissive mNeonGreen low populations. These populations were then analyzed for sgRNA abundance by NGS. The mNG high group represents host defense factors, the mNG low group represent host dependency factors.

As expected the KO of key IFN gamma pathway genes or ISGs made cells more susceptible in the IFN gamma stimulated condition but did not have an effect in the non-stimulated condition. PLSCR1-KO significantly enhanced SARS-CoV-2 infection in +/- IFN treated cells in Huh7.5 and A549 cells. The authors thus conclude that PLSCR1 is obligate to combat SARS-CoV-2 USA-WA1/2020, Delta B.1.617.2 and Omicron B.1.1.529 infection. The antiviral effect of PLSCR1 was rescued by genetic complementation. It was also rescued by cross species complementation with horseshoe bat or mouse PLSCR1. Then they also show that PLSCR1-KO is also proviral in Let1, Calu-3, Tonsil-ACE2, HaCaT-ACE2 and Hela-ACE2 cell lines.

Using a pseudovirus system it was shown that PLSCR1 expression blocked MERS-CoV, SARS-CoV(-2, VOC) and BatCoV-WIV-1 spike-mediated entry. Endemic CoV spike-mediated entry was inhibited to a lesser extend. Thus they conclude that PLSCR1 inhibits the type I fusion mechanism.

To gain mechanistic insight, the authors specifically inhibit cathepsin-dependent and TMPRSS2-dependent virus entry using E-64d and camostat in control and PLSCR1-KO cells. PLSCR1-KO cells showed increased entry, whereas E-64d treatment reversed that phenotype and blocked entry. Camostat treatment in PLSCR1-KO did not alter viral entry. This shows that PLSCR1 plays a role in endosomal fusion pathway. They moreover show with a split luciferase assay that PLSCR1 interferes with Sars-CoV-2 infection after receptor-mediated internalization.

Further, stainings of PLSCR1, as well as viral proteins and microscopy analysis reveal that PLSCR1 localizes to SARS-CoV-2 vesicles and inhibits viral RNA release. The authors also assessed which region or domain of PLSCR1 may crucial for the observed phenotype and identified that deletion of a beta strand and beta barrel (aa 86-118) or a C-terminal hydrophobic helix (aa 291-318) to be detrimental for PLSCR1 antiviral activity. Notably, His262 which has been identified to be associated to severe COVID19 (in case of His/Tyr change) is close to these regions. A corresponding His262Tyr PLSCR1 mutant were showed impaired anti-SARSR-CoV-2 activity.

Overall, this is an interesting work and identifies PLSCR1 as an antiviral factor for SARS-CoV-2. The mechanistic work in the second part of the paper is well done. However, there're major concerns about the first part of the study and about novelty when it comes to a role of PLSCR1 during virus infections.

Specific concerns:

major:

1. a very general but major concern is that a role of PLSCR1 for virus replication is not novel. PLSCR1 has been studied already in HCV and herpesvirus infections (and others). Although it is acknowledged that the current manuscript goes very deep into mechanistic studies on entry, it is disturbing that neither in the introduction nor in the discussion anything is mentioned about previous work on PLSCR1 in the context of virus infection.

As stated earlier, we apologize for not elaborating on these studies given word limits in the submission guidelines and the fact that the manuscript was heavily focused on SARS-CoV-2 rather than other viruses. We have now highlighted these papers in our revised manuscript, specifically in the discussion section. Thanks for the feedback.

The mechanistic studies conducted in our paper found a novel anti-SARS-CoV-2 function of PLSCR1, which directly antagonizes the viral fusion process both at the plasma membrane and within virus-containing vesicles to inhibit viral dsRNA entry into the cytosol. This mechanism is different from any previously reported activity of PLSCR1 during viral infection, including activation of type I IFN signaling (Dong et al, J Viol, 2004), degradation of viral proteins (Yuan et al, J Proteome Res, 2015), and interaction with viral proteins to block viral transcription or nuclear import (Kusano et al, Biochem Biophys Res Commun, 2013; Kusano et al, J Biol Chem, 2019; Luo et al, PLOS Pathog, 2018). In one case, the interaction of PLSCR1 with viral proteins during the viral fusion process was reported to promote infection (Gong et al, FEBS Lett, 2011).

Our mutagenic analysis (25 separate mutations in human PLSCR1 alone) has helped in dissecting which protein activities confer its anti-SARS-CoV-2 profile. Such mutants have been used in many of the new assay systems in this revised version. These include *in situ* lipid scramblase assays in human TMEM16F KO cells; viral spike-plasma membrane fusion assays for syncytium formation; plasma membrane rigidity assays using optical tweezers on PLSCR1 KO cells expressing wild-type or palmitoylation mutants; and real-time nanosecond molecular dynamic simulations of palmitoylated human PLSCR1 protein bound to the lipid membrane. Most importantly, uncoupling PLSCR1 scramblase activity from its anti-SARS-CoV-2 function enables the field to move in a new direction and look for fusion partners or membrane activities that are not reliant on PS externalization. This is a major step forward and especially exciting for understanding immune defense to coronaviruses.

2. The actual impact of the antiviral activity of PLSCR1 appears to be overstated throughout all experiments that deal with measurements of viral replication. This includes experiments shown in Figure 2 -4. The authors show a linear scale that should actually

be logarithmic. For example if the linear scale is shown and indicated that values have to be multiplied by 10E6 the differences look much greater than they actually are (e.g. 2x10E6 compared to 4x10E6 is actually not very much). Therefore, graphs have to be changed to a log scale. A similar concern is the measurement of "relative viral RNA level; normalised to NC 1h". It would be much clearer to determine and show genome equivalents. Most likely the criticism raised above would also apply since differences may not look so great anymore.

We agree that the \log_{10} scale is suitable for visualizing data distributed across a wide range (e.g. Fig 2b, c). We have therefore converted all the viral replication data from Fig. 2 and 3 to a \log_{10} scale but have kept the comparative mutagenic analysis in Fig. 4e and 4i as a linear scale because it is easier to discern which point mutations genuinely impact anti-SARS-CoV-2 activity. A similar approach is applied to PLSCR1 mutants in Extended Data Fig. 8c and 9d,e. For viral RNA we have now converted all measurements to genome equivalents as requested.

Regardless of the metric being measured (p.f.u., fluorescence, genome equivalents), the fold change remains the same irrespective of the type of scale used (\log_{10} scale, linear scale, or relative RNA level). Indeed, PLSCR1 compares favorably with other anti-SARS-CoV-2 restriction factors which have already been established (e.g. IFITM3, NCoA7 isoform4, LY6E and CD74 p41 isoform) (Majdoul et al, Nat. Rev. Immunol., 2021). New data added to the revised manuscript now directly compares these ISGs head-to-head with PLSCR1. Each was stably expressed in Huh7.5 cells and carried a Flag tag enabling simultaneous detection by the same anti-Flag antibody in immunoblots. All were expressed at similar or higher levels than PLSCR1. As shown in **inserted Fig.13 (Manuscript Fig. 2j and Extended Data Fig. 3f)**, the anti-SARS-CoV-2 function of PLSCR1 was stronger than that of both IFITM3 and NCoA7 (10.5-fold inhibition vs. ~1.5-2.0 fold). Ectopic expression of the CD74 p41 isoform exhibited the strongest activity (55-fold inhibition), however, its expression is principally confined to professional immune cells. This may limit its ability to provide widespread protection, whereas PLSCR1 is distributed across numerous human cell types (**Manuscript Extended Data Fig. 3a**), especially airway epithelial cells which serve as the frontline barrier engaging SARS-CoV-2. Indeed, we also compared the anti-SARS-CoV-2 function of PLSCR1 directly with LY6E in primary human trachea epithelial cells (hTEpiC) as part of the airway system. Here, LY6E served as a positive antiviral control. We found that their anti-SARS-CoV-2 activities are comparable (7.9-fold vs. 7.2-fold) (**Inserted Fig.14, Manuscript Fig. 2h and Extended Data Fig. 3b**). Thus, PLSCR1 exhibits potencies commensurate with or even exceeding established anti-SARS-CoV-2 proteins.

Inserted Fig.13. Left: Quantification of SARS-CoV-2 infection in Huh7.5 cells stably overexpressing indicated interferon stimulating genes (ISGs) (MOI=1, 48h p.i.). (n=6) Right: Protein expression level.

Inserted Fig.14. Quantification of pseudovirus (PsV) infection in human trachea epithelial cells (hTEpiC) stably expressing indicated proteins. Cells were challenged with an HIV-1-based luciferase-expressing vector pseudotyped with SARS-CoV-2-Spike (Omicron). PsV infection was quantified by luciferase activity 48 h post-infection. RLU, relative light units. (n=6)

Interferon mobilizes hundreds of ISGs to combat invading pathogens. Sometimes their anti-viral effects are cumulative or redundant which can result in either a modest or no observable phenotype. At other times these compound effects may arise from parallel pathways to yield a more pronounced phenotype. For example, the anti-IAV effect of NCOA7 and IFITM3 are cumulative and independent (Doyle et al, Nature Microbiology, 2018). We also found PLSCR1 co-operates with LY6E in A549-ACE2 cells to produce an additive phenotype which is more marked than either pathway alone. Human lung A549 epithelia express endogenous LY6E that is otherwise absent in human Huh7.5 hepatoma cells (Pfaender et al, Nature Microbiology, 2020). This allowed direct loss-of-function comparison of LY6E with PLSCR1. Individual deletions found PLSCR1 was more important than LY6E for anti-SARS-CoV-2 immunity under basal conditions, while each exhibited similar susceptibility under IFN- γ -induced conditions (Inserted Fig.15 below, Manuscript Fig. 2 and Extended Data Fig. 3g,h). Most notably, however, dual deletion of

both ISGs led to heightened defects exceeding each of the single knockouts alone. Hence both ISGs appear to act in concert as independent pathways which helps exert greater control of SARS-CoV-2 infection.

Inserted Fig.15. Left: Quantification of SARS-CoV-2 infection in A549-ACE2 cells of indicated genotypes in the presence or absence of IFN-γ (100 U.ml⁻¹) 48 h p.i. (MOI=0.2). (n=6) Right: Bar graph showing the average restriction ratio (-IFN-γ/+IFN-γ).

3. The claim that PLSCR1 may have antiviral activity against viruses that enter cell via type-I fusion proteins is not fully supported by the provided data. A much wider panel of viruses should be used. Notably, the use of HCV in this context would require to mention and discuss previous work on PLSCR1/HCV (see comment point 1)

Thank you for the sound advice and we have avoided classifying virus targets of PLSCR1 based on fusion properties alone. This is because our experiments spanning 14 different intact or pseudotyped viruses (now including HCV) found the effectiveness of PLSCR1 is not strictly related to the type of fusion protein used for viral entry. For example, PLSCR1 is highly active against most beta coronaviruses tested (which use a type I fusion mechanism) but it is largely dispensable for Ebola virus which also uses type I fusion for entry. Conversely, PLSCR1 overexpression in Huh7.5 cells exhibited modest activity against Dengue virus (type II fusion)(Manuscript Extended Data Fig. 4b,c). Hence it appears other factors besides the type of fusion mechanism used for entry can influence how PLSCR1 impacts viral infection. The identity of any additional factors is a focus of future study.

As you suggested, we have also now cited previous work about other viruses including HCV in the discussion section.

4. The authors show that mainly, or even exclusively, the endosomal route of entry is targeted by PLSCR1. This raises questions about the overall impact of this mechanism, since inhibition of the endosomal entry pathway by antiviral drugs have not been very successful in in vivo experiments and in humans.

New experiments conducted on the TMPRSS2 pathway during the interim period have shown PLSCR1 can also interfere with the cell surface route (Manuscript Extended Data Fig. 4h,i). This is because PLSCR1 also directly inhibits cell surface fusion as revealed in new acceptor-donor syncytium assays added to the manuscript (Manuscript Fig. 3i and Fig. 4j). This was an exciting discovery and suggests it has broad activity against SARS-CoV-2 variants irrespective of the entry route or cell type.

SARS-CoV-2 typically utilizes two different fusion routes to invade host cells: a cathepsin-dependent endosomal fusion pathway and a TMPRSS2-dependent cell-surface fusion pathway (Jackson et al, Nat. Rev. Mol. Cell. Biol., 2022). The preference towards either pathway depends on the host cell type and the SARS-CoV-2 variant examined. The expression of TMPRSS2 is relatively high in the lower airway, while cathepsin-L (required for the endosomal pathway) is more dominant in upper airway cells, like nasal epithelial cells. The recent Omicron variant altered its preference to the endosomal route and therefore prefers to replicate in the upper airway (Willett et al, Nature Microbiology, 2022).

Inhibitors of the endosomal pathway are widely used in basic research and demonstrate great potency against SARS-CoV-2 infection in experimental cell models. However, the reason why such inhibitors are less successful in clinical trials is complicated. Factors including tissue distribution and bioavailability as well as toxicity all need to be considered before Phase 3 trials or clinical use. Given the endolysosomal pathway is crucial for survival of all cell types, its nonspecific inhibition by E-64d (a pan-cathepsin inhibitor) or HCQ (alkalinizes the pH of late endolysosomes) will likely cause severe side effects. Hence the reason to search for more specific viral targets or defense factors like PLSCR1 that could be manipulated through small molecules.

minor:

please explain the initial screen better. In particular figure 1e. It looks like PLSCR1 is appearing as one the top hits in Huh7 cell with or without IFN γ treatment, while it appears as top hit in A549-ACE2 cells only under conditions of IFN γ treatment.

This outcome arose from how the screen was designed and analyzed. As shown in Fig. 1d, under resting conditions, 77.9% of A549-ACE2 cells were infected with SARS-CoV-2-mNG. Since genes were ranked by the MAGeCK algorithm according to their relative enrichment in the highly infected population (mNG^{High}) versus low infection population (mNG^{Low}), it means that 77.9% of sgRNA-integrated cells in the resting condition are enriched in mNG^{High}. Consequently, despite the significant enrichment of PLSCR1 sgRNAs in mNG^{High}, other sgRNAs will also be highly enriched in mNG^{High} and therefore mask the true ranking and antiviral phenotype of PLSCR1 in the resting condition. It is why PLSCR1 is not as conspicuous under resting conditions in A549-ACE2 cells.

Conversely, due to the relatively low expression level of endogenous ACE2 in Huh7.5 cells, only 8.86% cells were infected under resting conditions. This makes the enrichment

of PLSCR1 sgRNAs in the mNG^{high} population much more obvious and significant. In either case, the important result is that PLSCR1 was discovered as a new anti-SARS-CoV-2 restriction factor across different human cell types and activation conditions within genome-wide screens.

Reviewer Reports on the First Revision:

Referees' comments:

Referee #1 (Remarks to the Author):

The authors have adequately responded to my concerns following the first round of peer review. Moreover, the added experimental results have resulted in a greater understanding of the molecular mechanisms by which PLSCR1 inhibits SARS-CoV-2 entry.

Alex Compton

Referee #2 (Remarks to the Author):

None

Referee #3 (Remarks to the Author):

The authors did a great job in addressing all of the reviewers' comments. As new data were included that addressed these comments, the overall quality and impact is improved. The role of PLSCR1 as a defense factor against SARS-CoV-2 infection is now very comprehensively described in this work and this will certainly stimulate the field. As noted by the authors, there's still a considerable amount of work needed to fully understand the exact mechanism, but it is agreed that elucidating this is beyond the scope of the current work.